# On the Variance of the Fisher Information for Deep Learning

**Alexander Soen**
The Australian National University
alexander.soen@anu.edu.au

**Ke Sun**
CSIRO's Data61, Australia
The Australian National University
sunk@ieee.org

## Abstract

In the realm of deep learning, the Fisher information matrix (FIM) gives novel insights and useful tools to characterize the loss landscape, perform second-order optimization, and build geometric learning theories. The exact FIM is either unavailable in closed form or too expensive to compute. In practice, it is almost always estimated based on empirical samples. We investigate two such estimators based on two equivalent representations of the FIM — both unbiased and consistent. Their estimation quality is naturally gauged by their variance given in closed form. We analyze how the parametric structure of a deep neural network can affect the variance. The meaning of this variance measure and its upper bounds are then discussed in the context of deep learning.

## 1  Introduction

The Fisher information is one of the most fundamental concepts in statistical machine learning. Intuitively, it measures the amount of information carried by a single random observation when the underlying model varies along certain directions in the parameter space: if such a variation does not change the underlying model, then a corresponding observation contains zero (Fisher) information and is non-informative regarding the varied parameter. Parameter estimation is impossible in this case. Otherwise, if the variation significantly changes the model and has large information, then an observation is informative and the parameter estimation can be more efficient as compared to parameters with small Fisher information. In machine learning, this basic concept is useful for defining intrinsic structures of the parameter space, measuring model complexity, and performing gradient-based optimization.

Given a statistical model that is specified by a parametric form $p(z \,|\, \theta)$ and a continuous domain $\theta \in \mathcal{M}$, the Fisher information matrix (FIM) is a 2D tensor varying with $\theta \in \mathcal{M}$, given by

$$\mathcal{I}(\theta) = \mathbb{E}_{p(z \,|\, \theta)} \left( \frac{\partial \ell}{\partial \theta} \frac{\partial \ell}{\partial \theta^\top} \right), \tag{1}$$

where $\mathbb{E}_{p(z \,|\, \theta)}(\cdot)$, or simply $\mathbb{E}_p(\cdot)$ if the model $p$ is clear from the context, denotes the expectation w.r.t. $p(z \,|\, \theta)$, and $\ell := \log p(z \,|\, \theta)$ is the log-likelihood function. All vectors are column vectors throughout this paper. Under weak conditions (see Lemma 5.3 in Lehmann and Casella [16] for the univariate case), the FIM has the equivalent expression $\mathcal{I}(\theta) = \mathbb{E}_{p(z \,|\, \theta)} \left( -\partial^2 \ell / \partial \theta \partial \theta^\top \right)$. Given $N$ i.i.d. observations $z_1, \ldots, z_N$, these two equivalent expressions of the FIM lead to two different estimators

$$\hat{\mathcal{I}}_1(\theta) = \frac{1}{N} \sum_{i=1}^{N} \left( \frac{\partial \ell_i}{\partial \theta} \frac{\partial \ell_i}{\partial \theta^\top} \right) \quad \text{and} \quad \hat{\mathcal{I}}_2(\theta) = \frac{1}{N} \sum_{i=1}^{N} \left( -\frac{\partial^2 \ell_i}{\partial \theta \partial \theta^\top} \right), \tag{2}$$

35th Conference on Neural Information Processing Systems (NeurIPS 2021).

where $\ell_i := \log p(\boldsymbol{z}_i \,|\, \boldsymbol{\theta})$ is the log-likelihood of the $i$'th observation $\boldsymbol{z}_i$. The notations $\hat{\mathcal{I}}_1(\boldsymbol{\theta})$ and $\hat{\mathcal{I}}_2(\boldsymbol{\theta})$ are abused for simplicity as they depend on both $\boldsymbol{\theta}$ and the random observations $\boldsymbol{z}_i$.

These estimators are universal and independent to the parametric form $p(\boldsymbol{z} \,|\, \boldsymbol{\theta})$. They are expressed in terms of the 1st- or 2nd-order derivatives of the log-likelihood. Usually, we already have these derivatives to perform gradient-based learning. Therefore, we can save computational cost and reuse these derivatives to estimate the Fisher information, which in turn can be useful, *e.g.*, to perform natural gradient optimization [1, 25]. Estimating the FIM is especially meaningful for deep learning, where the computational overhead of the exact FIM can be significant.

It is straightforward from the law of large numbers and the central limit theorem that both estimators in Eq. (2) are unbiased and consistent. This is formally stated as follows.

**Proposition 1.**

$$\mathbb{E}_{p(\boldsymbol{z} \,|\, \boldsymbol{\theta})}\left(\hat{\mathcal{I}}_1(\boldsymbol{\theta})\right) = \mathbb{E}_{p(\boldsymbol{z} \,|\, \boldsymbol{\theta})}\left(\hat{\mathcal{I}}_2(\boldsymbol{\theta})\right) = \mathcal{I}(\boldsymbol{\theta}).$$

$$\forall \epsilon > 0, \ \lim_{N \to \infty} \text{Prob}\left(\left\|\hat{\mathcal{I}}_1(\boldsymbol{\theta}) - \mathcal{I}(\boldsymbol{\theta})\right\|_F + \left\|\hat{\mathcal{I}}_2(\boldsymbol{\theta}) - \mathcal{I}(\boldsymbol{\theta})\right\|_F > \epsilon\right) = 0,$$

*where* $\text{Prob}(\cdot)$ *denotes the probability of the parameter statement being true and* $\|\cdot\|_F$ *is the Frobenius norm of a tensor (with* $\|\cdot\|_2$ *as the regular vector $L_2$-norm)*

The Fisher information can be zero for non-regular models or infinite [7]. However, these properties may not be preserved by the empirical estimators.

How far can $\hat{\mathcal{I}}_1(\boldsymbol{\theta})$ and $\hat{\mathcal{I}}_2(\boldsymbol{\theta})$ deviate from the "true FIM" $\mathcal{I}(\boldsymbol{\theta})$, and how fast can they converge to $\mathcal{I}(\boldsymbol{\theta})$ as the number of observations increases? To answer these questions, it is natural to think of the variance of $\hat{\mathcal{I}}_1(\boldsymbol{\theta})$ and $\hat{\mathcal{I}}_2(\boldsymbol{\theta})$. For example, an estimator with a large variance means the estimation does not accurately reflect $\mathcal{I}(\boldsymbol{\theta})$; and any procedure depending on the FIM consequently suffers from the estimation error. Through studying the variance, we can control the estimation quality and reliably perform subsequent measurements or algorithms based on the FIM.

Towards this direction, we made the following contributions that will unfold in the following Sections 2 to 4:

- We review and rediscover two equivalent expression of the FIM in the context of deep feed-forward networks (Section 2);

- We give in closed form the variance (extending to meaningful upper bounds) and discuss the convergence rate of the estimators $\hat{\mathcal{I}}_1(\boldsymbol{\theta})$ and $\hat{\mathcal{I}}_2(\boldsymbol{\theta})$ (Section 3);

- We analyze how the 1st- and 2nd-order derivatives of the neural network can affect the estimation of the FIM (Section 4).

We discuss related work in Section 5 and conclude in Section 6.

## 2 Feed-forward Networks with Exponential Family Output

This section realizes the concept of Fisher information in a feed-forward network with exponentially family output and explains why its estimators are useful in theory and practice.

Consider supervised learning with a neural network. The underlying statistical model is $p(\boldsymbol{z} \,|\, \boldsymbol{\theta}) = p(\boldsymbol{x})p(\boldsymbol{y} \,|\, \boldsymbol{x}, \boldsymbol{\theta})$, where $\boldsymbol{z} = (\boldsymbol{x}, \boldsymbol{y})$, the random variable $\boldsymbol{x}$ represents features, and $\boldsymbol{y}$ is the target variable. The marginal distribution $p(\boldsymbol{x})$ is parameter-free, usually fixed as the empirical distribution $p(\boldsymbol{x}) = \frac{1}{M}\sum_{i=1}^M \delta(\boldsymbol{x} - \boldsymbol{x}_i)$ w.r.t. a set of observations $\{\boldsymbol{x}_i\}_{i=1}^M$, where $\delta(\cdot)$ is the Dirac delta. In this paper, we consider w.l.o.g. $M = 1$ as the FIM w.r.t. observations $\{\boldsymbol{x}_i\}_{i=1}^M$ is simply the average over FIMs of each individual observation. All results generalize to multiple observations by taking the empirical average. The predictor $p(\boldsymbol{y} \,|\, \boldsymbol{x}, \boldsymbol{\theta})$ is a neural network with parameters $\boldsymbol{\theta} = \{\boldsymbol{W}_{l-1}\}_{l=1}^L$

and exponential family output units, given by

$$p(\boldsymbol{y}\,|\,\boldsymbol{x}) = \exp\left(\boldsymbol{t}^\top(\boldsymbol{y})\boldsymbol{h}_L - F(\boldsymbol{h}_L)\right),$$
$$\boldsymbol{h}_L = \boldsymbol{W}_{L-1}\bar{\boldsymbol{h}}_{L-1},$$
$$\boldsymbol{h}_l = \sigma(\boldsymbol{W}_{l-1}\bar{\boldsymbol{h}}_{l-1}), \quad (l = 1, \ldots, L-1)$$
$$\bar{\boldsymbol{h}}_l = (\boldsymbol{h}_l^\top, 1)^\top,$$
$$\boldsymbol{h}_0 = \boldsymbol{x}, \tag{3}$$

where $\boldsymbol{t}(\boldsymbol{y})$ is the sufficient statistics of the prediction model, $F(\boldsymbol{h}) = \log \int \exp(\boldsymbol{t}^\top(\boldsymbol{y})\boldsymbol{h})d\boldsymbol{y}$ is the log-partition function, and $\sigma : \Re \to \Re$ is an element-wise non-linear activation function. Moreover, $\boldsymbol{W}_l$ is a $n_{l+1} \times (n_l + 1)$ matrix, representing the neural network parameters (weights and biases) in the $l$'th layer, where $n_l := \dim(\boldsymbol{h}_l)$ denotes the size of layer $l$. We use $\boldsymbol{W}_l^-$ for the $n_{l+1} \times n_l$ weight matrix without the bias terms, obtained by removing the last column of $\boldsymbol{W}_l$. $\boldsymbol{h}_l$ is a learned representation of the input $\boldsymbol{x}$. All intermediate variables $\boldsymbol{h}_l$ are extended to include a constant scalar 1 in $\bar{\boldsymbol{h}}_l$, so that a linear layer can simply be expressed as $\boldsymbol{W}_l\bar{\boldsymbol{h}}_l$. The last layer's output $\boldsymbol{h}_L$ with dimensionality $n_L$ specifies the natural parameter of the exponential family.

We need the following Lemma which gives the FIM w.r.t. $\boldsymbol{h}_L$, which is a $n_L \times n_L$ matrix in simple closed form for commonly used probability distributions.

**Lemma 2.** *For the neural network model specified in Eq. (3),*

$$\mathcal{I}(\boldsymbol{h}_L) = \mathrm{Cov}(\boldsymbol{t}(\boldsymbol{y})) = \frac{\partial\boldsymbol{\eta}}{\partial\boldsymbol{h}_L},$$

*where* $\mathrm{Cov}(\cdot)$ *denotes the covariance matrix w.r.t.* $p(\boldsymbol{y}\,|\,\boldsymbol{x},\boldsymbol{\theta})$, $\boldsymbol{\eta} := \boldsymbol{\eta}(\boldsymbol{h}_L) := \partial F/\partial\boldsymbol{h}_L$ *is the expectation parameters, and the vector-vector-derivative* $\partial\boldsymbol{\eta}/\partial\boldsymbol{h}_L$ *denotes the Jacobian matrix of the mapping* $\boldsymbol{h}_L \to \boldsymbol{\eta}$.

The derivatives of the log-likelihood $\ell(\boldsymbol{\theta}) := \log p(\boldsymbol{x}, \boldsymbol{y}\,|\,\boldsymbol{\theta})$ characterize its landscape and are essential to compute the FIM. By Eq. (3), the score function (gradient of $\ell$) is

$$\frac{\partial\ell}{\partial\boldsymbol{\theta}} = \left(\frac{\partial\boldsymbol{h}_L}{\partial\boldsymbol{\theta}}\right)^\top (\boldsymbol{t}(\boldsymbol{y}) - \boldsymbol{\eta}(\boldsymbol{h}_L)) = \frac{\partial\boldsymbol{h}_L^a}{\partial\boldsymbol{\theta}}(\boldsymbol{t}_a - \boldsymbol{\eta}_a). \tag{4}$$

In this paper, we mix the usual $\Sigma$-notation of summation with the Einstein notation: in the same term, an index appearing in both upper- and lower-positions indicates a sum over this index. For example, $t_a h^a = \sum_a t_a h_a$. Hence, in our equations, upper- and lower-indexes have the same meaning: both $h^a$ and $h_a$ mean the $a$'th element of $\boldsymbol{h}$. For convenience and consistency, we take quantities w.r.t. $\boldsymbol{\theta}$ as upper indexed and other quantities as lower indexed, *i.e.*, $\mathcal{I}^{ij}(\boldsymbol{\theta})$ versus $\mathcal{I}_{ij}(\boldsymbol{h}_L)$. This mixed representation of sums helps to simplify our expressions without causing confusion. From Eq. (4) and Lemma 2, the Hessian of $\ell$ is given by

$$\frac{\partial^2\ell}{\partial\boldsymbol{\theta}\partial\boldsymbol{\theta}^\top} = (\boldsymbol{t}_a - \boldsymbol{\eta}_a)\frac{\partial^2\boldsymbol{h}_L^a}{\partial\boldsymbol{\theta}\partial\boldsymbol{\theta}^\top} - \frac{\partial\boldsymbol{h}_L^a}{\partial\boldsymbol{\theta}}\frac{\partial\boldsymbol{\eta}_a}{\partial\boldsymbol{\theta}^\top} = (\boldsymbol{t}_a - \boldsymbol{\eta}_a)\frac{\partial^2\boldsymbol{h}_L^a}{\partial\boldsymbol{\theta}\partial\boldsymbol{\theta}^\top} - \frac{\partial\boldsymbol{h}_L^a}{\partial\boldsymbol{\theta}}\mathcal{I}_{ab}(\boldsymbol{h}_L)\frac{\partial\boldsymbol{h}_L^b}{\partial\boldsymbol{\theta}^\top}. \tag{5}$$

Similar to the case of a general statistical model, the FIM is equivalent to the expectation of the Hessian of $-\ell$ as long as the activation function is smooth enough.

**Theorem 3.** *Consider the neural network model in Eq. (3). For any activation function $\sigma \in C^2(\Re)$ (both $\sigma'(z)$ and $\sigma''(z)$ exist and are continuous), we have $\mathcal{I}(\boldsymbol{\theta}) = \mathbb{E}_p\left(-\frac{\partial^2\ell}{\partial\boldsymbol{\theta}\partial\boldsymbol{\theta}^\top}\right)$.*

**Remark 3.1.** *ReLU networks do not have this equivalent expression as* $\mathrm{ReLU}(z)$ *is not differentiable at $z = 0$.*

Through the definition of the FIM, or alternatively its equivalent formula in Theorem 3, we arrive at the same expression

$$\mathcal{I}(\boldsymbol{\theta}) = \left(\frac{\partial\boldsymbol{h}_L}{\partial\boldsymbol{\theta}}\right)^\top \mathcal{I}(\boldsymbol{h}_L)\frac{\partial\boldsymbol{h}_L}{\partial\boldsymbol{\theta}^\top} = \frac{\partial\boldsymbol{h}_L^a}{\partial\boldsymbol{\theta}}\mathcal{I}_{ab}(\boldsymbol{h}_L)\frac{\partial\boldsymbol{h}_L^b}{\partial\boldsymbol{\theta}^\top}. \tag{6}$$

Equation (6) takes the form of a generalized Gauss-Newton matrix [18]. This general expression of the FIM has been known in the literature [24, 25]. Under weak conditions, $\mathcal{I}(\boldsymbol{\theta})$ is a pullback

metric [30] of $\mathcal{I}(\boldsymbol{h}_L)$ in Lemma 2 associated with the mapping $\boldsymbol{\theta} \to \boldsymbol{h}_L$. To compute $\mathcal{I}(\boldsymbol{\theta})$ in closed form, one need to first compute the Jacobian matrix of size $n_L \times \dim(\boldsymbol{\theta})$ then perform the matrix multiplication in Eq. (6). The naive algorithm to evaluate Eq. (6) has a computational complexity of $\mathcal{O}(n_L^2 \dim(\boldsymbol{\theta}) + n_L \dim^2(\boldsymbol{\theta}))$, where the term $\mathcal{O}(n_L \dim^2(\boldsymbol{\theta}))$ is dominant as $\dim(\boldsymbol{\theta}) \gg n_L$ in deep architectures. Once the parameter $\boldsymbol{\theta}$ is updated, the FIM has to be recomputed. This is infeasible in practice for large networks where $\dim(\boldsymbol{\theta})$ can be millions or billions.

The two estimators $\hat{\mathcal{I}}_1(\boldsymbol{\theta})$ and $\hat{\mathcal{I}}_2(\boldsymbol{\theta})$ in Eq. (2) provide a computationally inexpensive way to estimate the FIM. Given $\boldsymbol{\theta}$ and $\boldsymbol{x}$, one can draw i.i.d. samples $\boldsymbol{y}_1, \ldots, \boldsymbol{y}_N \sim p(\boldsymbol{y} \mid \boldsymbol{x}, \boldsymbol{\theta})$. Both $\partial \ell_i / \partial \boldsymbol{\theta}$ and $\partial^2 \ell_i / \partial \boldsymbol{\theta} \partial \boldsymbol{\theta}^\top$ can be evaluated directly through auto-differentiation (AD) that is highly optimized for modern GPUs. For $\hat{\mathcal{I}}_1(\boldsymbol{\theta})$, we already have $\partial \ell_i / \partial \boldsymbol{\theta}$ to perform gradient descent. For $\hat{\mathcal{I}}_2(\boldsymbol{\theta})$, efficient methods to compute the Hessian are implemented in AD frameworks such as PyTorch [26]. Using these derivatives, the computational cost only scales with the number $N$ of samples but does not scale with $n_L$.

We rarely need the full FIM of size $\dim(\boldsymbol{\theta}) \times \dim(\boldsymbol{\theta})$. Most of the time, only its diagonal blocks are needed, where each block corresponds to a subset of parameters, *e.g.*, the neural network weights of a particular layer. Therefore the computation of both estimators can be further reduced.

If $p(\boldsymbol{y} \mid \boldsymbol{x}, \boldsymbol{\theta})$ has the parametric form in Eq. (3), from Eqs. (4) and (5), the FIM estimators become

$$\hat{\mathcal{I}}_1(\boldsymbol{\theta}) = \frac{\partial \boldsymbol{h}_L^a}{\partial \boldsymbol{\theta}} \cdot \frac{1}{N} \sum_{i=1}^N (\boldsymbol{t}_a(\boldsymbol{y}_i) - \boldsymbol{\eta}_a)(\boldsymbol{t}_b(\boldsymbol{y}_i) - \boldsymbol{\eta}_b) \cdot \frac{\partial \boldsymbol{h}_L^b}{\partial \boldsymbol{\theta}^\top}, \tag{7}$$

$$\hat{\mathcal{I}}_2(\boldsymbol{\theta}) = \left( \boldsymbol{\eta}_a - \frac{1}{N} \sum_{i=1}^N \boldsymbol{t}_a(\boldsymbol{y}_i) \right) \frac{\partial^2 \boldsymbol{h}_L^a}{\partial \boldsymbol{\theta} \partial \boldsymbol{\theta}^\top} + \frac{\partial \boldsymbol{h}_L^a}{\partial \boldsymbol{\theta}} \mathcal{I}_{ab}(\boldsymbol{h}_L) \frac{\partial \boldsymbol{h}_L^b}{\partial \boldsymbol{\theta}^\top}. \tag{8}$$

Recall that the notation of $\hat{\mathcal{I}}_1(\boldsymbol{\theta})$ and $\hat{\mathcal{I}}_2(\boldsymbol{\theta})$ is abused as they depend on $\boldsymbol{x}$ and $\boldsymbol{y}_1 \cdots \boldsymbol{y}_N$. Notably, in Eq. (7), $\hat{\mathcal{I}}_1(\boldsymbol{\theta})$ is expressed in terms of the Jacobian matrix of the mapping $\boldsymbol{\theta} \to \boldsymbol{h}_L$ and the empirical variance of the minimal sufficient statistic $\boldsymbol{t}(\boldsymbol{y}_i)$ of the output exponential family. In Eq. (8), $\hat{\mathcal{I}}_2(\boldsymbol{\theta})$ depends on both the Jacobian and Hessian of $\boldsymbol{\theta} \to \boldsymbol{h}_L$ and the empirical average of $\boldsymbol{t}(\boldsymbol{y}_i)$. The second term on the right-hand side (RHS) of Eq. (8) is exactly the FIM, and therefore the first term serves as a bias term. Eqs. (7) and (8) are only for the case with exponential family output. If the output units belong to non-exponential families, *e.g.*, a statistical mixture model, one falls back to the general formulae, *i.e.*, Eq. (2) for the FIM.

As an application of the Fisher information, the Cramér-Rao lower bound (CRLB) states that any unbiased estimator $\hat{\boldsymbol{\theta}}$ of the parameters $\boldsymbol{\theta}$ satisfies $\mathrm{Cov}(\hat{\boldsymbol{\theta}}) \succeq [\mathcal{I}(\boldsymbol{\theta})]^{-1}$. For example, in Lemma 2, the FIM is w.r.t. the output of the neural network. As such, $\mathcal{I}(\boldsymbol{h}_L)$ can be used to study the estimation covariance of $\boldsymbol{h}_L$ based on random samples $\boldsymbol{y}_1 \cdots \boldsymbol{y}_N$ drawn from $p(\boldsymbol{y} \mid \boldsymbol{x}, \boldsymbol{\theta})$. Similarly for Eq. (6), we can consider unbiased estimators of the weights of the neural network. In any case, to apply the CRLB, one needs an accurate estimation of $\mathcal{I}(\boldsymbol{\theta})$. If the scale of $\mathcal{I}(\boldsymbol{\theta})$ is relatively small when compared to its covariance, its estimation $\hat{\mathcal{I}}(\boldsymbol{\theta})$ is more likely to be a small positive value (or even worse, zero or negative). The empirical computation of the CRLB is not meaningful in this case.

## 3 The Variance of the FIM Estimators

Based on the deep learning architecture specified in Eq. (3), we measure the quality of the two estimators $\hat{\mathcal{I}}_1(\boldsymbol{\theta})$ and $\hat{\mathcal{I}}_2(\boldsymbol{\theta})$ given by their variances. Given the same sample size $N$, a smaller variance is preferred as the estimator is more accurate and likely to be closer to the true FIM $\mathcal{I}(\boldsymbol{\theta})$. We study how the structure of the exponential family has an impact on the variance.

### 3.1 Variance in closed form

We first consider $\hat{\mathcal{I}}_1(\boldsymbol{\theta})$ and $\hat{\mathcal{I}}_2(\boldsymbol{\theta})$ in Eq. (2) as real matrices of dimension $\dim(\boldsymbol{\theta}) \times \dim(\boldsymbol{\theta})$. As $\hat{\mathcal{I}}_1(\boldsymbol{\theta})$ is a square matrix, the corresponding covariance is a 4D tensor $\left[ \mathrm{Cov}\left( \hat{\mathcal{I}}_1(\boldsymbol{\theta}) \right) \right]^{ijkl}$ of dimension $\dim(\boldsymbol{\theta}) \times \dim(\boldsymbol{\theta}) \times \dim(\boldsymbol{\theta}) \times \dim(\boldsymbol{\theta})$, representing the covariance between the two elements

$\hat{\mathcal{I}}_1^{ij}(\boldsymbol{\theta})$ and $\hat{\mathcal{I}}_1^{kl}(\boldsymbol{\theta})$. The element-wise variance of $\hat{\mathcal{I}}_1(\boldsymbol{\theta})$ is a matrix with the same size of $\hat{\mathcal{I}}_1(\boldsymbol{\theta})$, which we denote as $\mathrm{Var}(\hat{\mathcal{I}}_1(\boldsymbol{\theta}))$. Thus,

$$\mathrm{Var}(\hat{\mathcal{I}}_1(\boldsymbol{\theta}))^{ij} = \left[\mathrm{Cov}\left(\hat{\mathcal{I}}_1(\boldsymbol{\theta})\right)\right]^{ijij}. \tag{9}$$

Similarly, the covariance and element-wise variance of $\hat{\mathcal{I}}_2(\boldsymbol{\theta})$ are denoted as $\left[\mathrm{Cov}\left(\hat{\mathcal{I}}_2(\boldsymbol{\theta})\right)\right]^{ijkl}$ and $\mathrm{Var}(\hat{\mathcal{I}}_2(\boldsymbol{\theta}))^{ij}$, respectively.

As the samples $\boldsymbol{y}_1, \ldots, \boldsymbol{y}_N$ are i.i.d., we have

$$\mathrm{Cov}(\hat{\mathcal{I}}_1(\boldsymbol{\theta})) = \frac{1}{N}\mathrm{Cov}\left(\frac{\partial\ell}{\partial\boldsymbol{\theta}}\frac{\partial\ell}{\partial\boldsymbol{\theta}^\top}\right) \quad \text{and} \quad \mathrm{Cov}(\hat{\mathcal{I}}_2(\boldsymbol{\theta})) = \frac{1}{N}\mathrm{Cov}\left(\frac{\partial^2\ell}{\partial\boldsymbol{\theta}\partial\boldsymbol{\theta}^\top}\right). \tag{10}$$

Both $\mathrm{Cov}(\hat{\mathcal{I}}_1(\boldsymbol{\theta}))$ and $\mathrm{Cov}(\hat{\mathcal{I}}_2(\boldsymbol{\theta}))$ have an order of $\mathcal{O}(1/N)$. For the neural network model in Eq. (3), we further have those covariance tensors in closed form.

**Theorem 4.**

$$\begin{aligned}\left[\mathrm{Cov}\left(\hat{\mathcal{I}}_1(\boldsymbol{\theta})\right)\right]^{ijkl} &= \frac{1}{N}\cdot\mathrm{Cov}\left(\frac{\partial\ell}{\partial\boldsymbol{\theta}_i}\frac{\partial\ell}{\partial\boldsymbol{\theta}_j}, \frac{\partial\ell}{\partial\boldsymbol{\theta}_k}\frac{\partial\ell}{\partial\boldsymbol{\theta}_l}\right)\\ &= \frac{1}{N}\cdot\partial_i\boldsymbol{h}_L^a(\boldsymbol{x})\partial_j\boldsymbol{h}_L^b(\boldsymbol{x})\partial_k\boldsymbol{h}_L^c(\boldsymbol{x})\partial_l\boldsymbol{h}_L^d(\boldsymbol{x})\cdot\left(\mathcal{K}_{abcd}(\boldsymbol{t}) - \mathcal{I}_{ab}(\boldsymbol{h}_L)\cdot\mathcal{I}_{cd}(\boldsymbol{h}_L)\right),\end{aligned}$$

*where the 4D tensor*

$$\mathcal{K}_{abcd}(\boldsymbol{t}) := \mathbb{E}\left[(\boldsymbol{t}_a - \boldsymbol{\eta}_a(\boldsymbol{h}_L(\boldsymbol{x})))(\boldsymbol{t}_b - \boldsymbol{\eta}_b(\boldsymbol{h}_L(\boldsymbol{x})))(\boldsymbol{t}_c - \boldsymbol{\eta}_c(\boldsymbol{h}_L(\boldsymbol{x})))(\boldsymbol{t}_d - \boldsymbol{\eta}_d(\boldsymbol{h}_L(\boldsymbol{x})))\right]$$

*is the 4th (unscaled) central moment* [1] *of* $\boldsymbol{t}(\boldsymbol{y})$ *and* $\partial_i\boldsymbol{h}_L(\boldsymbol{x}) := \partial\boldsymbol{h}_L(\boldsymbol{x})/\partial\boldsymbol{\theta}_i$.

**Remark 4.1.** *The 4D tensor* $(\mathcal{K}_{abcd}(\boldsymbol{t}) - \mathcal{I}_{ab}(\boldsymbol{h}_L)\cdot\mathcal{I}_{cd}(\boldsymbol{h}_L))$ *is the covariance of the random matrix*

$$\frac{\partial\ell}{\partial\boldsymbol{h}_L}\frac{\partial\ell}{\partial\boldsymbol{h}_L^\top} = (\boldsymbol{t}(\boldsymbol{y}) - \boldsymbol{\eta})(\boldsymbol{t}(\boldsymbol{y}) - \boldsymbol{\eta})^\top,$$

*where* $\boldsymbol{y} \sim p(\boldsymbol{y}\,|\,\boldsymbol{h}_L)$. *This random matrix is an estimator of* $\mathcal{I}(\boldsymbol{h}_L)$, *i.e. the FIM w.r.t. the natural parameters* $\boldsymbol{h}_L$. *Theorem 4 describes how the covariance tensor adapts w.r.t. the coordinate transformation* $\boldsymbol{h}_L \to \boldsymbol{\theta}$.

Notably, as $\boldsymbol{t}(\boldsymbol{y})$ is the sufficient statistics of an exponential family, the derivatives of the log-partition function $F(\boldsymbol{h})$ w.r.t. the natural parameters $\boldsymbol{h}$ are equivalent to the *cumulants* of $\boldsymbol{t}(\boldsymbol{y})$. The cumulants correspond to the coefficients of the Taylor expansion of the logarithm of the moment generating function [20]. Importantly, the cumulants of order 3 and below are equivalent to the central moments (see *e.g.* Lemma 2). However, this is not the case for the 4th central moment which must be expressed as a combination of the 2nd and 4th cumulants, as stated in the following Lemma.

**Lemma 5.**

$$\mathcal{K}_{abcd}(\boldsymbol{t}) = \kappa_{abcd} + \mathcal{I}_{ab}(\boldsymbol{h}_L)\cdot\mathcal{I}_{cd}(\boldsymbol{h}_L) + \mathcal{I}_{ac}(\boldsymbol{h}_L)\cdot\mathcal{I}_{bd}(\boldsymbol{h}_L) + \mathcal{I}_{ad}(\boldsymbol{h}_L)\cdot\mathcal{I}_{bc}(\boldsymbol{h}_L),$$

*where*

$$\kappa_{abcd} := \left.\frac{\partial^4 F(\boldsymbol{h})}{\partial\boldsymbol{h}_a\partial\boldsymbol{h}_b\partial\boldsymbol{h}_c\partial\boldsymbol{h}_d}\right|_{\boldsymbol{h}=\boldsymbol{h}_L(\boldsymbol{x})}.$$

**Remark 5.1.** *In the 1D case, the 4th central moment simplifies to* $\mathcal{K}(\boldsymbol{t}) = F''''(\boldsymbol{h}_L) + 3(F''(\boldsymbol{h}_L))^2$.

For the second estimator $\hat{\mathcal{I}}_2(\boldsymbol{\theta})$, the covariance only depends on the 2nd central moment of $\boldsymbol{t}(\boldsymbol{y})$.

**Theorem 6.**

$$\left[\mathrm{Cov}\left(\hat{\mathcal{I}}_2(\boldsymbol{\theta})\right)\right]^{ijkl} = \frac{1}{N}\cdot\mathrm{Cov}\left(-\frac{\partial^2\ell}{\partial\boldsymbol{\theta}_i\partial\boldsymbol{\theta}_j}, -\frac{\partial^2\ell}{\partial\boldsymbol{\theta}_k\partial\boldsymbol{\theta}_l}\right) = \frac{1}{N}\cdot\partial_{ij}^2\boldsymbol{h}_L^\alpha(\boldsymbol{x})\partial_{kl}^2\boldsymbol{h}_L^\beta(\boldsymbol{x})\mathcal{I}_{\alpha\beta}(\boldsymbol{h}_L),$$

*where* $\partial_{ij}^2\boldsymbol{h}_L(\boldsymbol{x}) := \partial^2\boldsymbol{h}_L(\boldsymbol{x})/\partial\boldsymbol{\theta}_i\partial\boldsymbol{\theta}_j$ [2].

---

[1] The kurtosis of a random variable is defined by its 4th standardized (both centered and normalized) moment. Here, $\mathcal{K}(\cdot)$ denotes the 4th central moment but *not* the kurtosis.

[2] In this paper, the derivatives are by default taken w.r.t. $\boldsymbol{\theta}$. Therefore, $\partial_i := \frac{\partial}{\partial\boldsymbol{\theta}_i}$ and $\partial_{ij}^2 := \frac{\partial^2}{\partial\boldsymbol{\theta}_i\partial\boldsymbol{\theta}_j}$.

**Remark 6.1.** *By Lemma 2, the matrix $\mathcal{I}_{\alpha\beta}(\boldsymbol{h}_L)$ is the covariance of the sufficient statistic $\boldsymbol{t}(\boldsymbol{y})$. Hence, the covariance of $\hat{\mathcal{I}}_2(\boldsymbol{\theta})$ scales with the covariance of $\boldsymbol{t}(\boldsymbol{y})$. If $\boldsymbol{t}(\boldsymbol{y})$ tends to be deterministic, then the covariance of $\hat{\mathcal{I}}_2(\boldsymbol{\theta})$ shrinks towards 0 and its estimation of the FIM becomes accurate.*

The covariance in Theorems 4 and 6 has two different components: ① the derivatives of the deep neural network; and ② the central (unscaled) moments of $\boldsymbol{t}(\boldsymbol{y})$. The 4D tensor $\mathcal{K}_{abcd}(\boldsymbol{t})$ and the 2D FIM $\mathcal{I}_{\alpha\beta}(\boldsymbol{h}_L)$ correspond to the 4th and 2nd central moments of $\boldsymbol{t}(\boldsymbol{y})$, respectively. Intuitively, the larger the scale of the Jacobian or the Hessian of the neural network mapping $\boldsymbol{\theta} \to \boldsymbol{h}_L$ and/or the larger the central moments of the exponential family, the lower the accuracy when estimating the FIM.

### 3.2 Variance Bounds

We aim to derive meaningful upper bounds of the covariances presented in Theorems 4 and 6. Using the Cauchy-Schwarz inequality, we can "decouple" the derivatives of the neural network mapping and the central moments of the exponential family into different terms. This provides various bounds on the scale of covariance quantities.

**Theorem 7.**

$$\left\| \mathrm{Cov}\left(\hat{\mathcal{I}}_1(\boldsymbol{\theta})\right) \right\|_F \leq \frac{1}{N} \cdot \left\| \frac{\partial \boldsymbol{h}_L}{\partial \boldsymbol{\theta}} \right\|_F^4 \cdot \left\| \mathcal{K}(\boldsymbol{t}) - \mathcal{I}(\boldsymbol{h}_L) \otimes \mathcal{I}(\boldsymbol{h}_L) \right\|_F,$$

*where $\otimes$ is the tensor-product: $(\mathcal{I}(\boldsymbol{h}_L) \otimes \mathcal{I}(\boldsymbol{h}_L))_{abcd} := \mathcal{I}_{ab}(\boldsymbol{h}_L) \cdot \mathcal{I}_{cd}(\boldsymbol{h}_L)$.*

The scale $\left\| \mathrm{Cov}\left(\hat{\mathcal{I}}_1(\boldsymbol{\theta})\right) \right\|_F$ measures how much the estimator $\hat{\mathcal{I}}_1(\boldsymbol{\theta})$ deviates from $\mathcal{I}(\boldsymbol{\theta})$. Theorem 7 says that this deviation is bounded by the scale of the Jacobian matrix $\partial \boldsymbol{h}_L / \partial \boldsymbol{\theta}$ as well as the scale of $(\mathcal{K}(\boldsymbol{t}) - \mathcal{I}(\boldsymbol{h}_L) \otimes \mathcal{I}(\boldsymbol{h}_L))$. Recall from Remark 4.1 the latter measures the variance when estimating the FIM $\mathcal{I}(\boldsymbol{h}_L)$ of the exponential family. Theorem 7 allows us to study these two different factors separately. Similarly, we have an upper bound on the scale of the covariance of $\hat{\mathcal{I}}_2(\boldsymbol{\theta})$.

**Theorem 8.**

$$\left\| \mathrm{Cov}\left(\hat{\mathcal{I}}_2(\boldsymbol{\theta})\right) \right\|_F \leq \frac{1}{N} \cdot \left\| \frac{\partial^2 \boldsymbol{h}_L(\boldsymbol{x})}{\partial \boldsymbol{\theta} \partial \boldsymbol{\theta}^\top} \right\|_F^2 \cdot \left\| \mathcal{I}(\boldsymbol{h}_L) \right\|_F.$$

On the RHS, the Hessian $\partial^2 \boldsymbol{h}_L(\boldsymbol{x}) / \partial \boldsymbol{\theta} \partial \boldsymbol{\theta}^\top$ is a 3D tensor of shape $n_L \times \dim(\boldsymbol{\theta}) \times \dim(\boldsymbol{\theta})$. Therefore, the variance of $\hat{\mathcal{I}}_2(\boldsymbol{\theta})$ is bounded by the scale of the Hessian, as well as the scale of the FIM $\mathcal{I}(\boldsymbol{h}_L)$ of the output exponential family.

We consider an upper bound to further simplify related terms in Theorems 7 and 8.

**Lemma 9.**

$$\left\| \mathcal{K}(\boldsymbol{t}) - \mathcal{I}(\boldsymbol{h}_L) \otimes \mathcal{I}(\boldsymbol{h}_L) \right\|_F \leq \sqrt{2} \left( \sum_{a=1}^{n_L} \left( \sqrt{\mathcal{K}_{aaaa}(\boldsymbol{t})} + \mathcal{I}_{aa}(\boldsymbol{h}_L) \right) \right)^2,$$

$$\left\| \mathcal{I}(\boldsymbol{h}_L) \right\|_F \leq \sum_{a=1}^{n_L} \mathcal{I}_{aa}(\boldsymbol{h}_L).$$

**Remark 9.1.** *Using Lemma 9, it is straightforward to bound the scale of the covariance tensors with the size of the Jacobian/Hessian, as well as the central moments $\mathcal{K}_{aaaa}(\boldsymbol{t})$ and $\mathcal{I}_{aa}(\boldsymbol{h}_L)$. These bounds are meaningful but omitted for brevity.*

**Remark 9.2.** *By Lemma 9, $\|\mathcal{K}(\boldsymbol{t}) - \mathcal{I}(\boldsymbol{h}_L) \otimes \mathcal{I}(\boldsymbol{h}_L)\|_F$ is in the order of $\mathcal{O}(n_L^2)$ and $\|\mathcal{I}(\boldsymbol{h}_L)\|_F$ is in the order of $\mathcal{O}(n_L)$.*

The scale of the tensors $\mathcal{K}(\boldsymbol{t}) - \mathcal{I}(\boldsymbol{h}_L) \otimes \mathcal{I}(\boldsymbol{h}_L)$ and $\mathcal{I}(\boldsymbol{h}_L)$ is bounded by the diagonal elements of $\mathcal{K}(\boldsymbol{t})$ and $\mathcal{I}(\boldsymbol{h}_L)$, or the element-wise central moments of $\boldsymbol{t}(\boldsymbol{y})$. Understanding the scale of these 1D central moments helps to understand the scale of the moment terms in our key statements.

Table 1: Cumulants of univariate exponential family distributions, given by derivatives of the log-partition function. $p$, $\mu$ and $\lambda$ denote the mean of the Bernoulli, normal, and Poisson distributions, respectively. † The normal distribution has unit standard deviation ($\sigma = 1$).

| DIST. | $F(\boldsymbol{h})$ | $\boldsymbol{h}$ | $\partial^2 F(\boldsymbol{h})$ | $\partial^4 F(\boldsymbol{h})$ |
|---|---|---|---|---|
| BERNOULLI | $\log(1 + \exp(\boldsymbol{h}))$ | $\log{p}/{1-p}$ | $p(1-p)$ | $p(1-p)(6p^2 - 6p + 1)$ |
| NORMAL† | $\boldsymbol{h}^2/2$ | $\mu$ | $1$ | $0$ |
| POISSON | $\exp(\boldsymbol{h})$ | $\log \lambda$ | $\lambda$ | $\lambda$ |

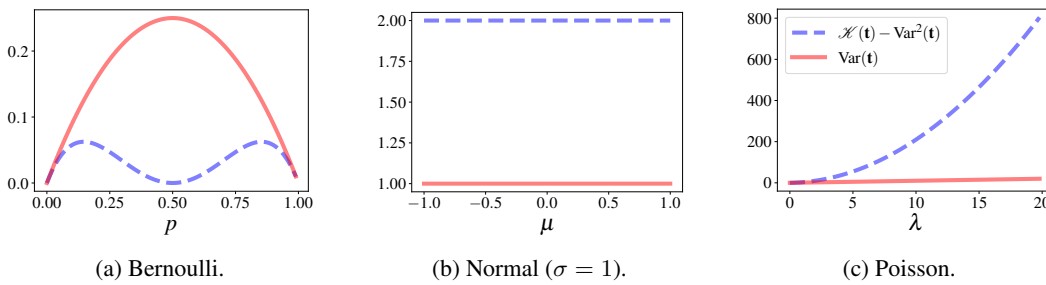

(a) Bernoulli.    (b) Normal ($\sigma = 1$).    (c) Poisson.

Figure 1: The scale of $\mathcal{K}(\boldsymbol{t}) - \mathrm{Var}^2(\boldsymbol{t})$ and $\mathrm{Var}(\boldsymbol{t})$ for the exponential family distributions in Table 1.

Table 1 presents some 1D exponential families and their cumulants. Figure 1 displays $\mathcal{K}(\boldsymbol{t}) - \mathrm{Var}^2(\boldsymbol{t})$ and $\mathrm{Var}(\boldsymbol{t})$ against the mean of these distributions. Based on Fig. 1a, if the neural network has Bernoulli output units, then the scale of $\mathcal{K}_{aaaa}(\boldsymbol{t}) - (\mathcal{I}_{aa}(\boldsymbol{h}_L))^2$ is smaller than $\mathcal{I}_{aa}(\boldsymbol{h}_L)$ regardless of $\boldsymbol{h}_L$. Notably, when $p = 0.5$, the variance of the first estimator $\hat{\mathcal{I}}_1(\boldsymbol{\theta})$ is 0 — regardless of $\boldsymbol{h}_L$. For normal distribution output units (corresponding to the mean squared error loss) in Fig. 1b, both central moment quantities are constant. For Poisson output units in Fig. 1c, $\mathcal{I}_{aa}(\boldsymbol{h}_L)$ increases linearly with the average number of events $\lambda$, while $\mathcal{K}_{aaaa}(\boldsymbol{t}) - (\mathcal{I}_{aa}(\boldsymbol{h}_L))^2$ increases quadratically. Thus, the upper bound of $\|\mathrm{Cov}(\hat{\mathcal{I}}_1(\boldsymbol{\theta}))\|_F$ increases faster than the upper bound of $\|\mathrm{Cov}(\hat{\mathcal{I}}_2(\boldsymbol{\theta}))\|_F$ as $\boldsymbol{h}_L$ enlarges. In this case, one may prefer $\hat{\mathcal{I}}_2(\boldsymbol{\theta})$ rather than $\hat{\mathcal{I}}_1(\boldsymbol{\theta})$ and/or control the scale of $\boldsymbol{h}_L$. In general, $\boldsymbol{h}_L$ is desired to be in certain regions in the parameter space of the exponential family to control the estimation variance of the FIM. Techniques to achieve this include regularization on the scale of $\boldsymbol{h}_L$; temperature scaling [12]; or normalization layers [4, 28]. Of course, they could inversely increase the scale of the derivatives of the neural network, which can be controlled by imposing additional constraints, *i.e.*, Lipschitz requirements.

### 3.3 Positive Definiteness

By definition, the FIM of any statistical model is positive semidefinite (p.s.d.). The first estimator $\hat{\mathcal{I}}_1(\boldsymbol{\theta})$ is naturally on the p.s.d. manifold (space of p.s.d. matrices). On the other hand, $\hat{\mathcal{I}}_2(\boldsymbol{\theta})$ can "fall off" the p.s.d. manifold. It is important to examine the likelihood for $\hat{\mathcal{I}}_2(\boldsymbol{\theta})$ having a negative spectrum and the corresponding scale, so that any algorithm (*e.g.* natural gradient) relying of the FIM being p.s.d. can be adapted.

Eq. (8) can be re-expressed as the sum of a p.s.d. matrix and a linear combination of $n_L$ symmetric matrices. We provide the likelihood for $\hat{\mathcal{I}}_2(\boldsymbol{\theta})$ staying on the p.s.d. manifold given conditions on the spectrum of the Hessian.

**Theorem 10.** *Let* $\lambda_{\min}(\cdot)$, $\lambda_{\max}(\cdot)$, *and* $\rho(\cdot)$ *denote the smallest eigenvalue, the largest eigenvalue, and the spectral radius (largest absolute value of the spectrum), respectively. Let* $\boldsymbol{\rho} := (\rho(\partial^2 \boldsymbol{h}_L^1), \cdots \rho(\partial^2 \boldsymbol{h}_L^{n_L}))$. *If* $\lambda_{\min}(\mathcal{I}(\boldsymbol{\theta})) > 0$, *then with probability at least*

$$1 - \frac{n_L \cdot \|\boldsymbol{\rho}\|_2^2 \cdot \lambda_{\max}(\mathcal{I}(\boldsymbol{h}_L))}{N \cdot \lambda_{\min}^2(\mathcal{I}(\boldsymbol{\theta}))},$$

*the estimator* $\hat{\mathcal{I}}_2(\boldsymbol{\theta})$ *with $N$ samples is a p.s.d. matrix.*

The bound becomes uninformative as the output layer size $n_L$ increases, as the spectrum of the Hessian of $\boldsymbol{h}_L$ scales up, or as the spectrum of the FIM $\mathcal{I}(\boldsymbol{h}_L)$ enlarges. On the other hand, as the

minimal eigenvalue of the FIM $\mathcal{I}(\boldsymbol{\theta})$ increases, Theorem 10 can give meaningful lower bounds. In particular, with sample rate $\mathcal{O}(N^{-1})$, estimator $\hat{\mathcal{I}}_2(\boldsymbol{\theta})$ will be a p.s.d. matrix. In practice for over-parametrized networks, $\lambda_{\min}(\mathcal{I}(\boldsymbol{\theta}))$ is close to or equals 0 and Theorem 10 is not meaningful. In any case, we need to consider the scale of the negative spectrum of $\hat{\mathcal{I}}_2(\boldsymbol{\theta})$.

**Theorem 11.**

$$\lambda_{\min}\left(\hat{\mathcal{I}}_2(\boldsymbol{\theta})\right) \geq -\rho(\partial^2 \boldsymbol{h}_L^a(\boldsymbol{x})) \left| \boldsymbol{\eta}_a - \frac{1}{N} \sum_{i=1}^{N} \boldsymbol{t}_a(\boldsymbol{y}_i) \right|.$$

Theorem 11 guarantees that in the worst case, the scale of the negative spectrum of $\hat{\mathcal{I}}_2(\boldsymbol{\theta})$ is controlled. By Lemma 2, $\mathrm{Var}(\boldsymbol{\eta}_a - \frac{1}{N}\sum_{i=1}^{N}\boldsymbol{t}_a(\boldsymbol{y}_i)) = \frac{1}{N}\mathcal{I}^{aa}(\boldsymbol{h}_L)$. Therefore, as $N$ increases or $\mathcal{I}^{aa}(\boldsymbol{h}_L)$ decreases, the negative spectrum of $\hat{\mathcal{I}}_2(\boldsymbol{\theta})$ will shrink. Further analysis on the spectrum of $\hat{\mathcal{I}}_1(\boldsymbol{\theta})$ and $\hat{\mathcal{I}}_2(\boldsymbol{\theta})$ can utilize the geometric structure of the p.s.d. manifold. This is left for future work.

### 3.4  Convergence Rate

The rate of convergence for each of the estimators is of particular interest when considering their practical viability. Through a generalized Chebyshev inequality [8], we can get a simple Frobenius norm convergence rate.

**Lemma 12.** *Let* $0 < \varepsilon < 1$. *Then*

$$\left\| \hat{\mathcal{I}}_1(\boldsymbol{\theta}) - \mathcal{I}(\boldsymbol{\theta}) \right\|_F \leq \frac{1}{\sqrt{\varepsilon N}} \cdot \sqrt{\sum_{i,j=1}^{\dim(\boldsymbol{\theta})} \mathrm{Var}\left( \frac{\partial \ell}{\partial \boldsymbol{\theta}_i} \frac{\partial \ell}{\partial \boldsymbol{\theta}_j} \right)}$$

*holds with probability at least* $1 - \varepsilon$; *and*

$$\left\| \hat{\mathcal{I}}_2(\boldsymbol{\theta}) - \mathcal{I}(\boldsymbol{\theta}) \right\|_F \leq \frac{1}{\sqrt{\varepsilon N}} \cdot \sqrt{\sum_{i,j=1}^{\dim(\boldsymbol{\theta})} \mathrm{Var}\left( -\frac{\partial^2 \ell}{\partial \boldsymbol{\theta}_i \partial \boldsymbol{\theta}_j} \right)}$$

*hold with probability at least* $1 - \varepsilon$.

Each of these convergence rates only depends on the element-wise variance of the estimator terms in Eq. (9). Moreover, each of the estimators has a convergence rate of $\mathcal{O}(N^{-1/2})$. The rate's constants are determined by the variance of the estimators given by Theorems 4 and 6, which are influenced by the derivatives of the neural network and the moments of the output exponential family.

## 4  Effect of Neural Network Derivatives

The derivatives of the deep learning network can affect the estimation variance of the FIM. By Theorem 4, the variance of the first estimator $\hat{\mathcal{I}}_1(\boldsymbol{\theta})$ scales with the Jacobian of the neural network mapping $\boldsymbol{\theta} \to \boldsymbol{h}_L(\boldsymbol{x})$. By Theorem 6, the variance of $\hat{\mathcal{I}}_2(\boldsymbol{\theta})$ scales with the Hessian of $\boldsymbol{\theta} \to \boldsymbol{h}_L(\boldsymbol{x})$. The larger the scale of the Jacobian or the Hessian, the larger the estimation variance. In this section, we examine these derivatives in more detail.

We give the closed form gradient of the log-likelihood $\ell$ and the last layer's output $\hat{\boldsymbol{h}}_L$ w.r.t. the neural network parameters.

**Lemma 13.**

$$\frac{\partial \ell}{\partial \boldsymbol{W}_l} = \boldsymbol{D}_l \frac{\partial \ell}{\partial \boldsymbol{h}_{l+1}} \bar{\boldsymbol{h}}_l^\top, \quad \frac{\partial \ell}{\partial \boldsymbol{h}_l} = \boldsymbol{B}_l^\top \left( \boldsymbol{t}(\boldsymbol{y}) - \boldsymbol{\eta}(\boldsymbol{h}_L) \right), \quad \frac{\partial \boldsymbol{h}_L^a}{\partial \boldsymbol{W}_l} = \boldsymbol{D}_l \boldsymbol{B}_{l+1}^\top \boldsymbol{e}_a \bar{\boldsymbol{h}}_l^\top,$$

*where* $\boldsymbol{e}_a$ *is the* $a^{th}$ *standard basis vector,* $\boldsymbol{B}_l$ *and* $\boldsymbol{D}_l$ *are recursively defined by*

$$\boldsymbol{B}_L = \boldsymbol{I}, \quad \boldsymbol{B}_l = \boldsymbol{B}_{l+1} \boldsymbol{D}_l \boldsymbol{W}_l^-,$$

$$\boldsymbol{D}_{L-1} = \boldsymbol{I}, \quad \boldsymbol{D}_l = \mathrm{diag}\left( \sigma'(\boldsymbol{W}_l \bar{\boldsymbol{h}}_l) \right),$$

$\boldsymbol{I}$ *is the identity matrix, and* $\mathrm{diag}(\cdot)$ *means a diagonal matrix with given diagonal entries.*

By Lemma 13, we can estimate the FIM w.r.t. the hidden representations $\boldsymbol{h}_l$ through

$$\hat{\mathcal{I}}_1(\boldsymbol{h}_l) = \frac{1}{N} \sum_{i=1}^{N} \frac{\partial \ell_i}{\partial \boldsymbol{h}_l} \frac{\partial \ell_i}{\partial \boldsymbol{h}_l^\top} = \boldsymbol{B}_l^\top \left( \frac{1}{N} \sum_{i=1}^{N} \left( \boldsymbol{t}(\boldsymbol{y}_i) - \boldsymbol{\eta}(\boldsymbol{h}_L) \right) \left( \boldsymbol{t}(\boldsymbol{y}_i) - \boldsymbol{\eta}(\boldsymbol{h}_L) \right)^\top \right) \boldsymbol{B}_l. \quad (11)$$

As $\boldsymbol{B}_l$ is recursively evaluated from the last layer to previous layers, the FIM can also be recursively estimated based on $\hat{\mathcal{I}}_1(\boldsymbol{\theta})$. It is similar to back-propagation, except that the FIMs are back-propagated instead of gradients of the network. This is similar to the backpropagated metric [23].

To investigate how the first estimator $\hat{\mathcal{I}}_1(\boldsymbol{\theta})$ is affected by the loss landscape, we bound the Frobenius norm of the parameter-output Jacobian $\partial \boldsymbol{h}_L / \partial \boldsymbol{\theta}$.

**Lemma 14.** *If the activation function has bounded gradient and $\forall z \in \Re$, $|\sigma'(z)| \leq 1$, then*

$$\left\| \frac{\partial \boldsymbol{h}_L}{\partial \boldsymbol{W}_l} \right\|_F = \| \boldsymbol{B}_{l+1} \boldsymbol{D}_l \|_F \cdot \| \bar{\boldsymbol{h}}_l \|_2 \leq \prod_{i=l+1}^{L-1} \| \boldsymbol{W}_i^- \|_F \cdot \| \bar{\boldsymbol{h}}_l \|_2, \quad (12)$$

where $\partial \boldsymbol{h}_L / \partial \boldsymbol{W}_l = \left[ \partial \boldsymbol{h}_L^1 / \partial \boldsymbol{W}_l, \cdots, \partial \boldsymbol{h}_L^{n_L} / \partial \boldsymbol{W}_l \right]$ is the derivative of a vector w.r.t. a matrix that is a 3D tensor.

Given Lemma 14, we see that the gradient $\partial \boldsymbol{h}_L / \partial \boldsymbol{W}_l$ scales with both the neural network weights $\boldsymbol{W}_i$ and the gradient of the activation function $\boldsymbol{D}_l$. Common activation functions have both bounded outputs and 1st-order derivatives; or at least are locally Lipschitz, *i.e.*, sigmoid and ReLU activation functions. During training, regularizing the scale of the neural network weights is a sufficient condition for bounding the variance of $\hat{\mathcal{I}}_1(\boldsymbol{\theta})$.

An alternative bound can be established which depends on the maximum singular values of the weight matrices.

**Lemma 15.** *Suppose that the activation function has bounded gradient $\forall z \in \Re$, $|\sigma'(z)| \leq 1$. Then*

$$\left\| \frac{\partial \boldsymbol{h}_L}{\partial \boldsymbol{W}_l} \right\|_{2_\sigma} \leq \left( \prod_{i=l+1}^{L-1} s_{\max}(\boldsymbol{W}_i^-) \right) \cdot \| \bar{\boldsymbol{h}}_l \|_2, \quad (13)$$

*where $s_{\max}(\cdot)$ denotes the maximum singular value and $\|\mathcal{T}\|_{2_\sigma}$ denotes the tensor spectral norm for a 3D tensor $\mathcal{T}$, defined by*

$$\|\mathcal{T}\|_{2_\sigma} = \max \left\{ \langle \mathcal{T}, \boldsymbol{\alpha} \otimes \boldsymbol{\beta} \otimes \boldsymbol{\gamma} \rangle : \|\boldsymbol{\alpha}\|_2 = \|\boldsymbol{\beta}\|_2 = \|\boldsymbol{\gamma}\|_2 = 1 \right\}.$$

Therefore, regularizing $s_{\max}(\boldsymbol{W}_i^-)$, or the spectral norm of the weight matrices, also helps to improve the estimation accuracy of the FIM.

We further reveal the relationship between the loss landscape and the FIM estimators. For a given target $\tilde{\boldsymbol{y}}$, the log-likelihood is denoted as $\tilde{l} := \log p(\tilde{\boldsymbol{y}} \,|\, \boldsymbol{x}, \boldsymbol{\theta})$. Furthermore, let us define $\Delta \hat{\mathcal{I}}_1(\boldsymbol{\theta}) := (\partial \tilde{l} / \partial \boldsymbol{\theta})(\partial \tilde{l} / \partial \boldsymbol{\theta}^\top) - \hat{\mathcal{I}}_1(\boldsymbol{\theta})$ and $\Delta \hat{\mathcal{I}}_2(\boldsymbol{\theta}) := -\partial^2 \tilde{l} / \partial \boldsymbol{\theta} \partial \boldsymbol{\theta}^\top - \hat{\mathcal{I}}_2(\boldsymbol{\theta})$. By Eqs. (4) and (5),

$$\Delta \hat{\mathcal{I}}_1(\boldsymbol{\theta}) = \frac{\partial \boldsymbol{h}_L^a}{\partial \boldsymbol{\theta}} \left[ (\boldsymbol{t}_a(\tilde{\boldsymbol{y}}) - \boldsymbol{\eta}_a)(\boldsymbol{t}_b(\tilde{\boldsymbol{y}}) - \boldsymbol{\eta}_b) - \frac{1}{N} \sum_{i=1}^{N} (\boldsymbol{t}_a(\boldsymbol{y}_i) - \boldsymbol{\eta}_a)(\boldsymbol{t}_b(\boldsymbol{y}_i) - \boldsymbol{\eta}_b) \right] \frac{\partial \boldsymbol{h}_L^b}{\partial \boldsymbol{\theta}^\top},$$

$$\Delta \hat{\mathcal{I}}_2(\boldsymbol{\theta}) = \left[ \frac{1}{N} \sum_{i=1}^{N} \boldsymbol{t}_a(\boldsymbol{y}_i) - \boldsymbol{t}_a(\tilde{\boldsymbol{y}}) \right] \frac{\partial^2 \boldsymbol{h}_L^a}{\partial \boldsymbol{\theta} \partial \boldsymbol{\theta}^\top}.$$

Hence, the difference between $\hat{\mathcal{I}}_1(\boldsymbol{\theta})$ (resp. $\hat{\mathcal{I}}_2(\boldsymbol{\theta})$) and the squared gradient (resp. Hessian) of the loss $-\tilde{\ell}$ depends on how $\boldsymbol{y}_i$ differs from $\tilde{\boldsymbol{y}}$. If the network $\boldsymbol{\theta}$ is trained, then the random samples $\boldsymbol{y}_i \sim p(\boldsymbol{y} \,|\, \boldsymbol{x}, \boldsymbol{\theta})$ are close to the given target $\tilde{\boldsymbol{y}}$. In this case, $\hat{\mathcal{I}}_1(\boldsymbol{\theta})$ corresponds to the squared gradient, and $\hat{\mathcal{I}}_2(\boldsymbol{\theta})$ corresponds to the Hessian. This is not true for untrained neural networks with random weights.

# 5  Related Work

The two estimators $\hat{\mathcal{I}}_1(\boldsymbol{\theta})$ and $\hat{\mathcal{I}}_2(\boldsymbol{\theta})$ are not new as one usually utilizes one of them to compute the FIM. Guo and Spall [11] analyzed their accuracy for univariate symmetric density functions based on the central limit theorem. Fisher information estimation is also examined in latent variable models [9]. Under the same topic, our work analyzes the factors affecting the variance of $\hat{\mathcal{I}}_1(\boldsymbol{\theta})$ and $\hat{\mathcal{I}}_2(\boldsymbol{\theta})$ for deep neural networks.

A large body of work tries to approximate the FIM or define similar curvature tensors for performing natural gradient descent [1, 18, 19, 14, 23, 31]. If the loss is an empirical expectation of $-\log p(\boldsymbol{y} \,|\, \boldsymbol{x}, \boldsymbol{\theta})$, its Hessian w.r.t. $\boldsymbol{h}_L$ is exactly $\mathcal{I}(\boldsymbol{h}_L)$. Then, the FIM in Eq. (6) is in the form of a Generalized Gauss-Newton matrix (GNN) [29, 18]. Ollivier [23] provided algorithm procedures to compute the unit-wise FIM and discusses Monte Carlo natural gradient. The FIM can be computed locally [31, 3] based on a joint distribution representation of the neural network. Efficient computational methods are developed to evaluate the FIM inverse [24].

The estimator $\hat{\mathcal{I}}_1(\boldsymbol{\theta})$ is *not* the "empirical Fisher" (see *e.g.* [18, Section 11]) as $\boldsymbol{y}_i$ is randomly sampled from $p(\boldsymbol{y} \,|\, \boldsymbol{x}, \boldsymbol{\theta})$, making $\hat{\mathcal{I}}_1(\boldsymbol{\theta})$ an unbiased estimator. The difference between the empirical Fisher and the FIM is clarified [15]. Similarly, the estimator $\hat{\mathcal{I}}_2(\boldsymbol{\theta})$ is *not* the Hessian of the loss, as $\boldsymbol{y}_i$ is randomly sampled rather than fixed to the given target.

Recently, the structure of the FIM (or its partial approximations) are examined in deep learning. The FIM of randomized networks is analyzed [2], where the weights of the neural network are assumed to be random. Often the analysis of randomized networks uses spectral analysis and random matrix theory [27]. An insight from this body of work is that most of the eigenvalues of the FIM are close to 0; while the high end of spectrum has large values [13]. In this paper, the estimators $\hat{\mathcal{I}}_1(\boldsymbol{\theta})$ and $\hat{\mathcal{I}}_2(\boldsymbol{\theta})$ are random matrices due to the sampling of $\boldsymbol{y}_i \sim p(\boldsymbol{y} \,|\, \boldsymbol{x})$ (the weights are considered fixed).

In information geometry [1, 22], the FIM serves as a Riemannian metric in the space of probability distributions. The FIM is a covariant tensor and is invariant to diffeomorphism on the sample space [22]. Higher order tensors are used to describe the intrinsic structure in this space. For example, the Riemannian curvature is a 4D tensor, while the Ricci curvature is 2D. The third cumulants of the sufficient statistics give an affine connection (belonging to the $\alpha$-connections or the Amari-Čensov tensor) of the exponential family. The FIM is generalized to a one-parameter family [21].

In statistics, our estimator $\hat{\mathcal{I}}_2(\boldsymbol{\theta})$ is Efron and Hinkley [10]'s "observed Fisher information", which is usually evaluated at the maximum likelihood estimation $\hat{\boldsymbol{\theta}}$. Higher order moments of the maximum likelihood estimator (MLE) were discussed (see *e.g.* Bowman and Shenton [5]). These moments are associated with parameter estimators and differ from the concept of the FIM estimators. Similar concepts are examined in higher-order asymptotic theory [1, Chapter 7].

# 6  Conclusion

The FIM is a covariant p.s.d. tensor revealing the intrinsic geometric structure of the parameter manifold. It yields useful practical methods such as the natural gradient. In practice, the true FIM $\mathcal{I}(\boldsymbol{\theta})$ is usually expensive or impossible to obtain. Estimators of the FIM based on empirical samples is used in the deep learning practice. We analyzed two different estimators $\hat{\mathcal{I}}_1(\boldsymbol{\theta})$ and $\hat{\mathcal{I}}_2(\boldsymbol{\theta})$ of the FIM of a deep neural network. These estimators are convenient to compute using auto-differentiation frameworks but randomly deviates from $\mathcal{I}(\boldsymbol{\theta})$. Our central results, Theorems 4 and 6, present the variance of $\hat{\mathcal{I}}_1(\boldsymbol{\theta})$ and $\hat{\mathcal{I}}_2(\boldsymbol{\theta})$ in closed form, which is further extended to upper bounds in simpler forms. Two factors affecting the estimation variance are ① the derivatives of neural network output $\boldsymbol{h}_L$ w.r.t. the weight parameters $\boldsymbol{\theta}$; and ② the property of $\boldsymbol{h}_L$ as an exponential family distribution. A large scale of the 1st- and/or 2nd-order derivatives leads to a large variance when estimating the FIM. Our analytical results can be useful to measure the quality of the estimated FIM and could lead to variance reduction techniques.

## Acknowledgments and Disclosure of Funding

We thank the anonymous NeurIPS reviewers for their constructive comments. We thank Frank Nielsen for the insightful feedback. We thank James C. Spall for pointing us to related work.

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
