# On the Variance of the Fisher Information
# for Deep Learning

# — Appendix —

**Alexander Soen**
The Australian National University
alexander.soen@anu.edu.au

**Ke Sun**
CSIRO's Data61, Australia
The Australian National University
sunk@ieee.org

This appendix contains a proofs of the results in the main text and further analysis on the two FIM estimators $\hat{\mathcal{I}}_1(\boldsymbol{\theta})$ and $\hat{\mathcal{I}}_2(\boldsymbol{\theta})$. In particular, Appendix C presents an analysis of how the FIM estimators and their covariance tensors change under reparametrization. Appendix D presents element-wise bound alternatives to those presented in Section 3.2. Appendix E explores various results using alternative norms to the Frobenius norm results of the main text. Appendix F presents an analysis on taking a linear combination of the two FIM estimators. Appendix G presents a numerical experiments of the FIM estimators on the MNIST dataset.

35th Conference on Neural Information Processing Systems (NeurIPS 2021).

# A  List of Symbols

Table A.1: Table of symbols.

| Symbol | Meaning | Defined |
|---|---|---|
| $\mathcal{I}(\boldsymbol{\theta})$ | Fisher information matrix (FIM) | Eq. (1) |
| $\hat{\mathcal{I}}_1(\boldsymbol{\theta})$ | An estimator of the FIM | Eq. (2) |
| $\hat{\mathcal{I}}_2(\boldsymbol{\theta})$ | Another estimator of the FIM | |
| $\mathrm{Var}(\cdot)$ | Element-wise variance of input; output is the same dimension as the input dimension | Eq. (9) |
| $\mathrm{Cov}(\cdot)$ | Pair-wise covariance of input; output is the dimension of an outer product on the input | Lemma 2 |
| $\boldsymbol{h}$ | Natural parameter of exponential family | Table 1 |
| $\boldsymbol{h}_l$ | Hidden layer output in our neural network model | Eq. (3) |
| $\boldsymbol{h}_L$ | Last layer's output in our neural network model and the natural parameter of the exponential family | Eq. (3) |
| $n_l$ | Size of layer $l$ | Eq. (3) |
| $F(\boldsymbol{h}_L)$ | Log-partition function of exponential family | Lemma 2 |
| $\boldsymbol{\eta} = \boldsymbol{\eta}(\boldsymbol{h}_L)$ | Dual parameterization of exponential family | Lemma 2 |
| $\mathcal{I}(\boldsymbol{h}_L) = \mathrm{Var}(\boldsymbol{t})$ | Covariance matrix of $\boldsymbol{t}$ | Lemma 2 |
| $\mathcal{K}(\boldsymbol{t})$ | 4th central moment of $\boldsymbol{t}$ | Theorem 4 |
| $\kappa_{a,b,c,d}$ | 4th order cumulant of exponential family | Lemma 5 |
| $\lambda_{\min}(M)$ | Smallest eigenvalue of $M$ | Theorem 10 |
| $\rho(M)$ | Spectral radius of $M$ (absolute value of spectrum) | Theorem 10 |
| $\partial\ell ; \partial^2\ell$ | Partial derivatives of likelihood w.r.t. weights | Eq. (2) |
| $\partial\boldsymbol{h}_L ; \partial^2\boldsymbol{h}_L$ | Partial derivatives of neural network w.r.t. weights | Eq. (4) |
| $\|\cdot\|_F$ | Frobenius norm / $L_2$-norm | Proposition 1 |
| $\|\cdot\|_2$ | 2-norm / $L_2$-norm | Proposition 1 |
| $\|\cdot\|_{2_\sigma}$ | Tensor spectral norm | Lemma 15 |
| $\|\cdot\|_1$ | $L_1$-norm | Theorem A.6 |
| $\|\cdot\|_\infty$ | $L_\infty$-norm | Theorem A.6 |
| $M^\top$ | Matrix transpose | Eq. (3) |
| $\otimes$ | Tensor product | Lemma A.4 |

# B  Variance of FIM estimators

## B.1  Proof of Proposition 1

*Proof.* The first statement holds as both $\hat{\mathcal{I}}_1(\boldsymbol{\theta})$ and $\hat{\mathcal{I}}_2(\boldsymbol{\theta})$ are point-wise estimators, they are unbiased (central limit theorem).

The second statement holds by the law of large numbers (and triangle inequality with $\varepsilon/2$ and a union bound). □

## B.2  Proof of Lemma 2

*Proof.* The statement follows as $p(\boldsymbol{y} \mid \boldsymbol{x}, \boldsymbol{\theta})$ is given by an exponential family. See [1]. □

## B.3 Proof of Theorem 3

*Proof.* Consider the alternative formulation of the FIM.

$$\mathbb{E}_{x,y}\left[-\frac{\partial^2}{\partial\boldsymbol{\theta}\partial\boldsymbol{\theta}^\top}\log p(\boldsymbol{y}\mid\boldsymbol{x})\right] = \mathbb{E}_{x,y}\left[\frac{\partial\ell}{\partial\boldsymbol{\theta}}\frac{\partial\ell}{\partial\boldsymbol{\theta}^\top}\right] - \mathbb{E}_{x,y}\left[\frac{\frac{\partial^2}{\partial\boldsymbol{\theta}\partial\boldsymbol{\theta}^\top}p(\boldsymbol{y}\mid\boldsymbol{x})}{p(\boldsymbol{y}\mid\boldsymbol{x})}\right]$$

$$= \mathcal{I}(\boldsymbol{\theta}) - \int p(\boldsymbol{x})\frac{\partial^2}{\partial\boldsymbol{\theta}\partial\boldsymbol{\theta}^\top}p(\boldsymbol{y}\mid\boldsymbol{x})\,\mathrm{d}\boldsymbol{y}\,\mathrm{d}\boldsymbol{x}.$$

Thus for this to be equivalent to the Jacobian definition, we need the residual term to be zero.

As $\sigma \in C^2(\Re)$ and thus $\sigma'$ is smooth, it follows by the composition of smooth functions that $p(\boldsymbol{y}\mid\boldsymbol{x})$ and $\partial p(\boldsymbol{y}\mid\boldsymbol{x})$ is also a smooth function. This provides sufficient conditions for the Leibniz integration rule to be used (switch order of integration and differentiation). As such,

$$\mathbb{E}_{x,y}\left[-\frac{\partial^2}{\partial\boldsymbol{\theta}\partial\boldsymbol{\theta}^\top}\log p(\boldsymbol{y}\mid\boldsymbol{x})\right] = \mathcal{I}(\boldsymbol{\theta}) - \int p(\boldsymbol{x})\frac{\partial^2}{\partial\boldsymbol{\theta}\partial\boldsymbol{\theta}^\top}p(\boldsymbol{y}\mid\boldsymbol{x})\,\mathrm{d}\boldsymbol{y}\,\mathrm{d}\boldsymbol{x}$$

$$= \mathcal{I}(\boldsymbol{\theta}) - \frac{\partial^2}{\partial\boldsymbol{\theta}\partial\boldsymbol{\theta}^\top}\int p(\boldsymbol{x})p(\boldsymbol{y}\mid\boldsymbol{x})\,\mathrm{d}\boldsymbol{y}\,\mathrm{d}\boldsymbol{x}$$

$$= \mathcal{I}(\boldsymbol{\theta}).$$

$\square$

## B.4 Proof of Theorem 4

*Proof.* First define $\boldsymbol{\delta} := \boldsymbol{\delta}(\boldsymbol{x},\boldsymbol{y};\boldsymbol{\theta}) = \boldsymbol{t}(\boldsymbol{y}) - \boldsymbol{\eta}(\boldsymbol{h}_L(\boldsymbol{x}))$.

$$\mathrm{Cov}\left(\partial_i\ell\cdot\partial_j\ell,\partial_k\ell\cdot\partial_l\ell\right) = \mathbb{E}_{\boldsymbol{y}\mid\boldsymbol{x};\boldsymbol{\theta}}[(\partial_i\ell\cdot\partial_j\ell)(\partial_k\ell\cdot\partial_l\ell)] - \mathbb{E}_{\boldsymbol{y}\mid\boldsymbol{x};\boldsymbol{\theta}}[\partial_i\ell\cdot\partial_j\ell]\cdot\mathbb{E}_{\boldsymbol{y}\mid\boldsymbol{x};\boldsymbol{\theta}}[\partial_k\ell\cdot\partial_l\ell].$$

We then calculate each components. For the first:

$$\mathbb{E}_{\boldsymbol{y}\mid\boldsymbol{x};\boldsymbol{\theta}}[(\partial_i\ell\cdot\partial_j\ell)(\partial_k\ell\cdot\partial_l\ell)] = \mathbb{E}_{\boldsymbol{y}\mid\boldsymbol{x};\boldsymbol{\theta}}\left[\partial_i\boldsymbol{h}_L^a(\boldsymbol{x})\cdot\partial_j\boldsymbol{h}_L^b(\boldsymbol{x})\cdot\partial_k\boldsymbol{h}_L^a(\boldsymbol{x})\cdot\partial_l\boldsymbol{h}_L^b(\boldsymbol{x})\cdot\delta_a\cdot\delta_b\cdot\delta_c\cdot\delta_d\right]$$

$$= \partial_i\boldsymbol{h}_L^a(\boldsymbol{x})\cdot\partial_j\boldsymbol{h}_L^b(\boldsymbol{x})\cdot\partial_k\boldsymbol{h}_L^a(\boldsymbol{x})\cdot\partial_l\boldsymbol{h}_L^b(\boldsymbol{x})\cdot\mathbb{E}_{\boldsymbol{y}\mid\boldsymbol{x};\boldsymbol{\theta}}\left[\delta_a\cdot\delta_b\cdot\delta_c\cdot\delta_d\right].$$

For the second term, we can first consider the expectation:

$$\mathbb{E}_{\boldsymbol{y}\mid\boldsymbol{x};\boldsymbol{\theta}}[\partial_i\ell\cdot\partial_j\ell]$$
$$= \mathbb{E}_{\boldsymbol{y}\mid\boldsymbol{x};\boldsymbol{\theta}}\left[\partial_i\boldsymbol{h}_L^a(\boldsymbol{x})\cdot\partial_j\boldsymbol{h}_L^b(\boldsymbol{x})\cdot\delta_a\cdot\delta_b\right]$$
$$= \partial_i\boldsymbol{h}_L^a(\boldsymbol{x})\cdot\partial_j\boldsymbol{h}_L^b(\boldsymbol{x})\cdot\mathbb{E}_{\boldsymbol{y}\mid\boldsymbol{x};\boldsymbol{\theta}}\left[\delta_a\cdot\delta_b\right].$$

Thus the product of this gives:

$$\mathbb{E}_{\boldsymbol{y}\mid\boldsymbol{x};\boldsymbol{\theta}}[\partial_i\ell\cdot\partial_j\ell]\cdot\mathbb{E}_{\boldsymbol{y}\mid\boldsymbol{x};\boldsymbol{\theta}}[\partial_k\ell\cdot\partial_l\ell]$$
$$= \partial_i\boldsymbol{h}_L^a(\boldsymbol{x})\cdot\partial_j\boldsymbol{h}_L^b(\boldsymbol{x})\cdot\partial_k\boldsymbol{h}_L^c(\boldsymbol{x})\cdot\partial_l\boldsymbol{h}_L^d(\boldsymbol{x})\cdot\mathbb{E}_{\boldsymbol{y}\mid\boldsymbol{x};\boldsymbol{\theta}}\left[\delta_a\cdot\delta_b\right]\cdot\mathbb{E}_{\boldsymbol{y}\mid\boldsymbol{x};\boldsymbol{\theta}}\left[\delta_c\cdot\delta_d\right].$$

This gives the total covariance:

$$\mathrm{Cov}\left(\partial_i\ell\cdot\partial_j\ell,\partial_k\ell\cdot\partial_l\ell\right) = \partial_i\boldsymbol{h}_L^a(\boldsymbol{x})\cdot\partial_j\boldsymbol{h}_L^b(\boldsymbol{x})\cdot\partial_k\boldsymbol{h}_L^c(\boldsymbol{x})\cdot\partial_l\boldsymbol{h}_L^d(\boldsymbol{x})\cdot$$
$$\left(\mathbb{E}_{\boldsymbol{y}\mid\boldsymbol{x};\boldsymbol{\theta}}\left[\delta_a\cdot\delta_b\cdot\delta_c\cdot\delta_d\right] - \mathbb{E}_{\boldsymbol{y}\mid\boldsymbol{x};\boldsymbol{\theta}}\left[\delta_a\cdot\delta_b\right]\cdot\mathbb{E}_{\boldsymbol{y}\mid\boldsymbol{x};\boldsymbol{\theta}}\left[\delta_c\cdot\delta_d\right]\right).$$

$\square$

## B.5 Proof of Lemma 5

*Proof.* We first consider the following notation, consistent with [20] except as subscripts, to denote the central moments and different multivariate cumulants. The *non-central moment* of a vector random variable $\boldsymbol{X}$ is denoted as

$$\kappa_{r_1\ldots r_m} = \mathbb{E}[\boldsymbol{X}_{r_1}\cdots\boldsymbol{X}_{r_m}],$$

where $r$ is an integer vector denoting the dimensions in which the non-central moment is taken as. The dimensions can be repeated, *i.e.*, $r_1 = r_2$ *etc*. Note that in the notation $r_1 \ldots r_m$ are not comma separated.

Similarly, for *central moments* we have:

$$\mathcal{K}_{r_1 \ldots r_m} = \mathbb{E}[(\boldsymbol{X}_{r_1} - \mathbb{E}[\boldsymbol{X}_{r_1}]) \cdots (\boldsymbol{X}_{r_m} - \mathbb{E}[\boldsymbol{X}_{r_m}])].$$

For cumulants, we use a comma separated subscripts to distinguish it from central moments. In our case ($X$ is from an exponential family), this just reduces to $m$-derivative of the log-partition function:

$$\kappa_{r_1, \ldots, r_m} = \frac{\partial^m F(\boldsymbol{h}_L)}{\partial \boldsymbol{h}_{r_1} \cdots \partial \boldsymbol{h}_{r_m}}$$

In addition to these moment-like quantities, we introduce the $[n]$ notation from [20] to denote the possible permutations in indices. For example:

$$\kappa_i \kappa_{j,k}[3] = \kappa_i \kappa_{j,k} + \kappa_j \kappa_{i,k} + \kappa_k \kappa_{i,j},$$

noting that the (central and non-central) moments and cumulants are index order invariant. The numbering also must be equal to the number of available permutations. Also note that $\kappa_i$ denotes a cumulant and not the moment.

[20] *only* provides a formulation from cumulants to non-central moments. Thus we need to rewrite $\mathcal{K}_{abcd}$ in terms of non-central moments. The following is given in [20, Equation (2.6)]:

$$\kappa_{ij} = \kappa_{i,j} + \kappa_i \kappa_j;$$
$$\kappa_{ijk} = \kappa_{i,j,k} + \kappa_i \kappa_{j,k}[3] + \kappa_i \kappa_j \kappa_k;$$
$$\kappa_{ijkl} = \kappa_{i,j,k,l} + \kappa_i \kappa_{j,k,l}[4] + \kappa_{i,j} \kappa_{k,l}[3] + \kappa_i \kappa_j \kappa_{k,l}[6] + \kappa_i \kappa_j \kappa_k \kappa_l.$$

Expanding $\mathcal{K}_{abcd}$, the 4th central moment, yields the following:

$$
\begin{aligned}
\mathcal{K}_{abcd} &= \kappa_{abcd} - \kappa_a \kappa_{bcd}[4] + \kappa_a \kappa_b \kappa_{cd}[6] - 3\kappa_a \kappa_b \kappa_c \kappa_d \\
&= \kappa_{abcd} - \kappa_a \kappa_{bcd}[4] + (\kappa_a \kappa_b (\kappa_{c,d} + \kappa_c \kappa_d))[6] - 3\kappa_a \kappa_b \kappa_c \kappa_d \\
&= \kappa_{abcd} - \kappa_a \kappa_{bcd}[4] + \kappa_a \kappa_b \kappa_{c,d}[6] + 3\kappa_a \kappa_b \kappa_c \kappa_d \\
&= \kappa_{abcd} - (\kappa_a (\kappa_{b,c,d} + \kappa_a \kappa_{c,d}[3] + \kappa_b \kappa_c \kappa_d))[4] + \kappa_a \kappa_b \kappa_{c,d}[6] + 3\kappa_a \kappa_b \kappa_c \kappa_d \\
&= \kappa_{abcd} - \kappa_a \kappa_{b,c,d}[4] - \kappa_a \kappa_b \kappa_{c,d}[6] - \kappa_a \kappa_b \kappa_c \kappa_d \\
&= \kappa_{a,b,c,d} + \kappa_{a,b} \kappa_{c,d}[3],
\end{aligned}
$$

which when substituting for derivatives proves the Lemma. $\square$

## B.6 Proof of Theorem 6

*Proof.*

$$\mathrm{Cov}\left(-\frac{\partial^2 \ell}{\partial \boldsymbol{\theta}_i \partial \boldsymbol{\theta}_j}, -\frac{\partial^2 \ell}{\partial \boldsymbol{\theta}_k \partial \boldsymbol{\theta}_l}\right)$$

$$= \mathrm{Cov}\left(\partial_i \boldsymbol{h}_L^a(\boldsymbol{x}) \mathcal{I}_{ab}(\boldsymbol{h}_L) \partial_j \boldsymbol{h}_L^b(\boldsymbol{x}) - [\boldsymbol{t}_\alpha(\boldsymbol{y}) - \boldsymbol{\eta}_\alpha] \frac{\partial^2 \boldsymbol{h}_L^\alpha(\boldsymbol{x})}{\partial \boldsymbol{\theta}_i \partial \boldsymbol{\theta}_j},\right.$$

$$\left. \partial_k \boldsymbol{h}_L^a(\boldsymbol{x}) \mathcal{I}_{ab}(\boldsymbol{h}_L) \partial_l \boldsymbol{h}_L^b(\boldsymbol{x}) - [\boldsymbol{t}_\beta(\boldsymbol{y}) - \boldsymbol{\eta}_\beta] \frac{\partial^2 \boldsymbol{h}_L^\beta(\boldsymbol{x})}{\partial \boldsymbol{\theta}_k \partial \boldsymbol{\theta}_l}\right)$$

$$= \mathrm{Cov}\left([\boldsymbol{t}_\alpha(\boldsymbol{y}) - \boldsymbol{\eta}_\alpha] \frac{\partial^2 \boldsymbol{h}_L^\alpha(\boldsymbol{x})}{\partial \boldsymbol{\theta}_i \partial \boldsymbol{\theta}_j}, [\boldsymbol{t}_\beta(\boldsymbol{y}) - \boldsymbol{\eta}_\beta] \frac{\partial^2 \boldsymbol{h}_L^\beta(\boldsymbol{x})}{\partial \boldsymbol{\theta}_k \partial \boldsymbol{\theta}_l}\right)$$

$$= E\left[[\boldsymbol{t}_\alpha(\boldsymbol{y}) - \boldsymbol{\eta}_\alpha] \frac{\partial^2 \boldsymbol{h}_L^\alpha(\boldsymbol{x})}{\partial \boldsymbol{\theta}_i \partial \boldsymbol{\theta}_j} \cdot [\boldsymbol{t}_\beta(\boldsymbol{y}) - \boldsymbol{\eta}_\beta] \frac{\partial^2 \boldsymbol{h}_L^\beta(\boldsymbol{x})}{\partial \boldsymbol{\theta}_k \partial \boldsymbol{\theta}_l}\right]$$

$$- E\left[[\boldsymbol{t}_\alpha(\boldsymbol{y}) - \boldsymbol{\eta}_\alpha] \frac{\partial^2 \boldsymbol{h}_L^\alpha(\boldsymbol{x})}{\partial \boldsymbol{\theta}_i \partial \boldsymbol{\theta}_j}\right] \cdot E\left[[\boldsymbol{t}_\beta(\boldsymbol{y}) - \boldsymbol{\eta}_\beta] \frac{\partial^2 \boldsymbol{h}_L^\beta(\boldsymbol{x})}{\partial \boldsymbol{\theta}_k \partial \boldsymbol{\theta}_l}\right]$$

$$= E\left[[\boldsymbol{t}_\alpha(\boldsymbol{y}) - \boldsymbol{\eta}_\alpha] \frac{\partial^2 \boldsymbol{h}_L^\alpha(\boldsymbol{x})}{\partial \boldsymbol{\theta}_i \partial \boldsymbol{\theta}_j} \cdot [\boldsymbol{t}_\beta(\boldsymbol{y}) - \boldsymbol{\eta}_\beta] \frac{\partial^2 \boldsymbol{h}_L^\beta(\boldsymbol{x})}{\partial \boldsymbol{\theta}_k \partial \boldsymbol{\theta}_l}\right]$$

$$= \frac{\partial^2 \boldsymbol{h}_L^\alpha(\boldsymbol{x})}{\partial \boldsymbol{\theta}_i \partial \boldsymbol{\theta}_j} \frac{\partial^2 \boldsymbol{h}_L^\beta(\boldsymbol{x})}{\partial \boldsymbol{\theta}_k \partial \boldsymbol{\theta}_l} E[\delta_\alpha \cdot \delta_\beta].$$

The theorem follows immediately. $\square$

## B.7 Proof of Theorem 7

*Proof.* Let $1 \le p, q \le \infty$ such that $1/p + 1/q = 1$. Let $\mathcal{T}_{abcd} = \mathcal{K}_{abcd}(\boldsymbol{t}) - \mathcal{I}_{ab}(\boldsymbol{h}_L) \cdot \mathcal{I}_{cd}(\boldsymbol{h}_L)$. From the last inequality in the proof of Lemma A.4 we have,

$$\left|\left[\mathrm{Cov}\left(\hat{\mathcal{I}}_1(\boldsymbol{\theta})\right)\right]^{ijkl}\right| \le \frac{1}{N} \cdot \|\partial_i \boldsymbol{h}_L(\boldsymbol{x})\|_p \cdot \|\partial_j \boldsymbol{h}_L(\boldsymbol{x})\|_p \cdot \|\partial_k \boldsymbol{h}_L(\boldsymbol{x})\|_p \cdot \|\partial_l \boldsymbol{h}_L(\boldsymbol{x})\|_p \cdot \|\mathcal{T}\|_q.$$

Thus for the $p$-norm,

$$\left\|\mathrm{Cov}\left(\hat{\mathcal{I}}_1(\boldsymbol{\theta})\right)\right\|_p$$

$$= \left(\sum_{i,j,k,l} \left|\left[\mathrm{Cov}\left(\hat{\mathcal{I}}_1(\boldsymbol{\theta})\right)\right]^{ijkl}\right|^p\right)^{1/p}$$

$$\le \frac{1}{N} \cdot \|\mathcal{T}\|_q \left(\sum_{i,j,k,l} |\|\partial_i \boldsymbol{h}_L(\boldsymbol{x})\|_p \cdot \|\partial_j \boldsymbol{h}_L(\boldsymbol{x})\|_p \cdot \|\partial_k \boldsymbol{h}_L(\boldsymbol{x})\|_p \cdot \|\partial_l \boldsymbol{h}_L(\boldsymbol{x})\|_p|^p\right)^{1/p}$$

$$= \frac{1}{N} \cdot \|\mathcal{T}\|_q \left(\left(\sum_i \|\partial_i \boldsymbol{h}_L(\boldsymbol{x})\|_p^p\right)^4\right)^{1/p}$$

$$= \frac{1}{N} \cdot \|\mathcal{T}\|_q \left(\left(\sum_i \|\partial_i \boldsymbol{h}_L(\boldsymbol{x})\|_p^p\right)^{1/p}\right)^4$$

$$= \frac{1}{N} \cdot \|\mathcal{T}\|_q \cdot \|\partial \boldsymbol{h}_L(\boldsymbol{x})\|_p^4.$$

Thus, by taking the $p = q = 2$ the Theorem holds. $\square$

## B.8 Proof of Theorem 8

*Proof.* Let $1 \leq p, q \leq \infty$ such that $1/p + 1/q = 1$. From the last inequality in the proof of Lemma A.5 we have,

$$\left| \left[ \text{Cov}\left( \hat{\mathcal{I}}_2(\boldsymbol{\theta}) \right) \right]^{ijkl} \right| \leq \frac{1}{N} \cdot \|\partial_{ij}^2 \boldsymbol{h}_L(\boldsymbol{x})\|_p \cdot \|\partial_{kl}^2 \boldsymbol{h}_L(\boldsymbol{x})\|_p \cdot \|\mathcal{I}(\boldsymbol{h}_L)\|_q$$

Thus for the $p$-norm,

$$
\begin{aligned}
\left\| \text{Cov}\left( \hat{\mathcal{I}}_2(\boldsymbol{\theta}) \right) \right\|_q &= \left( \sum_{i,j,k,l} \left| \left[ \text{Cov}\left( \hat{\mathcal{I}}_2(\boldsymbol{\theta}) \right) \right]^{ijkl} \right|^p \right)^{1/p} \\
&\leq \frac{1}{N} \cdot \|\mathcal{I}(\boldsymbol{h}_L)\|_q \cdot \left( \sum_{i,j,k,l} \left| \|\partial_{ij}^2 \boldsymbol{h}_L(\boldsymbol{x})\|_p \cdot \|\partial_{kl}^2 \boldsymbol{h}_L(\boldsymbol{x})\|_p \right|^p \right)^{1/p} \\
&= \frac{1}{N} \cdot \|\mathcal{I}(\boldsymbol{h}_L)\|_q \cdot \left( \left( \sum_{i,j} \left| \|\partial_{ij}^2 \boldsymbol{h}_L(\boldsymbol{x})\|_p \right|^p \right)^2 \right)^{1/p} \\
&= \frac{1}{N} \cdot \|\mathcal{I}(\boldsymbol{h}_L)\|_q \cdot \left( \left( \sum_{i,j} \left| \|\partial_{ij}^2 \boldsymbol{h}_L(\boldsymbol{x})\|_p \right|^p \right)^{1/p} \right)^2 \\
&= \frac{1}{N} \cdot \|\mathcal{I}(\boldsymbol{h}_L)\|_q \cdot \|\partial^2 \boldsymbol{h}_L(\boldsymbol{x})\|_p^2.
\end{aligned}
$$

Thus, by taking the $p = q = 2$ the Theorem holds. $\qquad\square$

## B.9 Proof of Lemma 9

*Proof.* We first prove the second part of the Lemma, which is simpler.

$$
\begin{aligned}
\|\mathcal{I}(\boldsymbol{h}_L)\|_F &= \sqrt{\sum_{a,b} \text{Cov}^2(\boldsymbol{t}_a, \boldsymbol{t}_b)} \\
&\leq \sqrt{\sum_{a,b} \text{Var}(\boldsymbol{t}_a)\text{Var}(\boldsymbol{t}_b)} \quad \text{(by Cauchy-Schwarz inequality)} \\
&= \sqrt{\sum_a \text{Var}(\boldsymbol{t}_a) \sum_b \text{Var}(\boldsymbol{t}_b)} \\
&= \sum_a \text{Var}(\boldsymbol{t}_a) \\
&= \sum_a \mathcal{I}_{aa}(\boldsymbol{h}_L). \quad\quad\quad\quad\quad\quad\quad\quad \text{(A.1)}
\end{aligned}
$$

We are left with the first part of the Lemma.

$$\|\mathcal{K}(\boldsymbol{t}) - \mathcal{I}(\boldsymbol{h}_L) \otimes \mathcal{I}(\boldsymbol{h}_L)\|_F^2$$

$$= \sum_{a,b,c,d} \Bigg( \mathbb{E}\left((\boldsymbol{t}_a - \boldsymbol{\eta}_a)(\boldsymbol{t}_b - \boldsymbol{\eta}_b)(\boldsymbol{t}_c - \boldsymbol{\eta}_c)(\boldsymbol{t}_d - \boldsymbol{\eta}_d)\right)$$

$$- \mathbb{E}\left((\boldsymbol{t}_a - \boldsymbol{\eta}_a)(\boldsymbol{t}_b - \boldsymbol{\eta}_b)\right) \mathbb{E}\left((\boldsymbol{t}_c - \boldsymbol{\eta}_c)(\boldsymbol{t}_d - \boldsymbol{\eta}_d)\right) \Bigg)^2$$

$$\leq 2 \sum_{a,b,c,d} \Bigg( \underbrace{\mathbb{E}^2\left((\boldsymbol{t}_a - \boldsymbol{\eta}_a)(\boldsymbol{t}_b - \boldsymbol{\eta}_b)(\boldsymbol{t}_c - \boldsymbol{\eta}_c)(\boldsymbol{t}_d - \boldsymbol{\eta}_d)\right)}_{S_{abcd}}$$

$$+ \underbrace{\mathbb{E}^2\left((\boldsymbol{t}_a - \boldsymbol{\eta}_a)(\boldsymbol{t}_b - \boldsymbol{\eta}_b)\right) \mathbb{E}^2\left((\boldsymbol{t}_c - \boldsymbol{\eta}_c)(\boldsymbol{t}_d - \boldsymbol{\eta}_d)\right)}_{T_{abcd}} \Bigg)$$

$$= 2 \sum_{a,b,c,d} (S_{abcd} + T_{abcd}).$$

We have

$$S_{abcd} = \mathbb{E}^2\left((\boldsymbol{t}_a - \boldsymbol{\eta}_a)(\boldsymbol{t}_b - \boldsymbol{\eta}_b)(\boldsymbol{t}_c - \boldsymbol{\eta}_c)(\boldsymbol{t}_d - \boldsymbol{\eta}_d)\right)$$

$$\leq \mathbb{E}\left((\boldsymbol{t}_a - \boldsymbol{\eta}_a)^2(\boldsymbol{t}_b - \boldsymbol{\eta}_b)^2\right) \mathbb{E}\left((\boldsymbol{t}_c - \boldsymbol{\eta}_c)^2(\boldsymbol{t}_d - \boldsymbol{\eta}_d)^2\right)$$

$$\leq \mathbb{E}^{1/2}(\boldsymbol{t}_a - \boldsymbol{\eta}_a)^4 \cdot \mathbb{E}^{1/2}(\boldsymbol{t}_b - \boldsymbol{\eta}_b)^4 \cdot \mathbb{E}^{1/2}(\boldsymbol{t}_c - \boldsymbol{\eta}_c)^4 \cdot \mathbb{E}^{1/2}(\boldsymbol{t}_d - \boldsymbol{\eta}_d)^4$$

$$= \sqrt{\mathcal{K}_{aaaa}(\boldsymbol{t})} \cdot \sqrt{\mathcal{K}_{bbbb}(\boldsymbol{t})} \cdot \sqrt{\mathcal{K}_{cccc}(\boldsymbol{t})} \cdot \sqrt{\mathcal{K}_{dddd}(\boldsymbol{t})}.$$

At the same time,

$$T_{abcd} = \mathbb{E}^2\left((\boldsymbol{t}_a - \boldsymbol{\eta}_a)(\boldsymbol{t}_b - \boldsymbol{\eta}_b)\right) \mathbb{E}^2\left((\boldsymbol{t}_c - \boldsymbol{\eta}_c)(\boldsymbol{t}_d - \boldsymbol{\eta}_d)\right)$$

$$\leq \mathbb{E}(\boldsymbol{t}_a - \boldsymbol{\eta}_a)^2 \cdot \mathbb{E}(\boldsymbol{t}_b - \boldsymbol{\eta}_b)^2 \cdot \mathbb{E}(\boldsymbol{t}_c - \boldsymbol{\eta}_c)^2 \cdot \mathbb{E}(\boldsymbol{t}_d - \boldsymbol{\eta}_d)^2$$

$$= \mathcal{I}_{aa}(\boldsymbol{h}_L) \cdot \mathcal{I}_{bb}(\boldsymbol{h}_L) \cdot \mathcal{I}_{cc}(\boldsymbol{h}_L) \cdot \mathcal{I}_{dd}(\boldsymbol{h}_L).$$

To sum up, we have

$$\|\mathcal{K}(\boldsymbol{t}) - \mathcal{I}(\boldsymbol{h}_L) \otimes \mathcal{I}(\boldsymbol{h}_L)\|_F^2 \leq 2 \sum_{a,b,c,d} (S_{abcd} + T_{abcd})$$

$$= 2 \left( \sum_a \sqrt{\mathcal{K}_{aaaa}(\boldsymbol{t})} \right)^4 + 2 \left( \sum_a \mathcal{I}_{aa}(\boldsymbol{h}_L) \right)^4$$

$$\leq 2 \left( \sum_a (\sqrt{\mathcal{K}_{aaaa}(\boldsymbol{t})} + \mathcal{I}_{aa}(\boldsymbol{h}_L)) \right)^4.$$

Taking the square root of both sides gives the result. $\qquad \square$

### B.10 Proof of Theorem 10

*Proof.* The estimator is a p.s.d. matrix subtracted by a linear combination of symmetric matrices.

$$\hat{\mathcal{I}}_2(\boldsymbol{x}; \boldsymbol{\theta}) = \mathcal{I}(\boldsymbol{\theta}) - \frac{1}{N} \sum_{i=1}^N [\boldsymbol{t}(\boldsymbol{y}_i) - \boldsymbol{\eta}(\boldsymbol{h}_L(\boldsymbol{x}))]^\top \frac{\partial^2 \boldsymbol{h}_L(\boldsymbol{x})}{\partial \boldsymbol{\theta} \partial \boldsymbol{\theta}^\top}$$

$$= \mathcal{I}(\boldsymbol{\theta}) - \sum_{a=1}^n (\bar{\boldsymbol{t}}_a - \mathbb{E}[\bar{\boldsymbol{t}}_a]) \frac{\partial^2 \boldsymbol{h}_L^a(\boldsymbol{x})}{\partial \boldsymbol{\theta} \partial \boldsymbol{\theta}^\top},$$

where $\bar{\boldsymbol{t}}_a = \frac{1}{N} \sum_{i=1}^N \boldsymbol{t}_a(\boldsymbol{y}_i)$.

We consider the Chebyshev inequalities for

$$k = \frac{\sqrt{N} \lambda_{\min}(\mathcal{I}(\boldsymbol{\theta}))}{\|\boldsymbol{\rho}\|_2 \sqrt{\lambda_{\max}(\mathcal{I}(\boldsymbol{h}_L))}} > 0.$$

$$\Pr\left(\|\bar{\boldsymbol{t}} - \mathbb{E}[\bar{\boldsymbol{t}}]\|_2 \leq k\sqrt{\frac{1}{N}\lambda_{\max}(\mathcal{I}(\boldsymbol{h}_L))}\right)$$

$$=\Pr\left(\frac{\|\bar{\boldsymbol{t}} - \mathbb{E}[\bar{\boldsymbol{t}}]\|_2^2}{\frac{1}{N}\lambda_{\max}(\mathcal{I}(\boldsymbol{h}_L))} \leq k^2\right)$$

$$\geq\Pr\left((\bar{\boldsymbol{t}} - \mathbb{E}[\bar{\boldsymbol{t}}])^\top\left(\frac{1}{N}\mathcal{I}(\boldsymbol{h}_L)\right)^{-1}(\bar{\boldsymbol{t}} - \mathbb{E}[\bar{\boldsymbol{t}}]) \leq k^2\right)$$

$$\geq 1 - \frac{n_L}{k^2}$$

$$= 1 - \frac{n_L\|\boldsymbol{\rho}\|_2^2\lambda_{\max}(\mathcal{I}(\boldsymbol{h}_L))}{N\lambda_{\min}^2(\mathcal{I}(\boldsymbol{\theta}))}.$$

Thus with probability at least

$$1 - \frac{n_L\|\boldsymbol{\rho}\|_2^2\lambda_{\max}(\mathcal{I}(\boldsymbol{h}_L))}{N\lambda_{\min}^2(\mathcal{I}(\boldsymbol{\theta}))},$$

the following statement is true: for all unit vector $\boldsymbol{v}$ we have

$$\boldsymbol{v}^\top\hat{\mathcal{I}}_2(\boldsymbol{\theta})\boldsymbol{v} = \boldsymbol{v}^\top\left(\mathcal{I}(\boldsymbol{\theta}) - \sum_{a=1}^n(\bar{\boldsymbol{t}}_a - \mathbb{E}[\bar{\boldsymbol{t}}_a])\frac{\partial^2\boldsymbol{h}_L^a(\boldsymbol{x})}{\partial\boldsymbol{\theta}\partial\boldsymbol{\theta}^\top}\right)\boldsymbol{v}$$

$$= \boldsymbol{v}^\top\mathcal{I}(\boldsymbol{\theta})\boldsymbol{v} - \sum_{a=1}^n(\bar{\boldsymbol{t}}_a - \mathbb{E}[\bar{\boldsymbol{t}}_a])\left(\boldsymbol{v}^\top\frac{\partial^2\boldsymbol{h}_L^a(\boldsymbol{x})}{\partial\boldsymbol{\theta}\partial\boldsymbol{\theta}^\top}\boldsymbol{v}\right)$$

$$\geq \lambda_{\min}(\mathcal{I}(\boldsymbol{\theta})) - \sum_{a=1}^n(\bar{\boldsymbol{t}}_a - \mathbb{E}[\bar{\boldsymbol{t}}_a])\left(\boldsymbol{v}^\top\frac{\partial^2\boldsymbol{h}_L^a(\boldsymbol{x})}{\partial\boldsymbol{\theta}\partial\boldsymbol{\theta}^\top}\boldsymbol{v}\right)$$

$$\geq \lambda_{\min}(\mathcal{I}(\boldsymbol{\theta})) - \sum_{a=1}^n|\bar{\boldsymbol{t}}_a - \mathbb{E}[\bar{\boldsymbol{t}}_a]| \cdot \rho(\partial^2\boldsymbol{h}_L^a)$$

$$\geq \lambda_{\min}(\mathcal{I}(\boldsymbol{\theta})) - \|\bar{\boldsymbol{t}} - \mathbb{E}[\bar{\boldsymbol{t}}]\|_2 \cdot \|\boldsymbol{\rho}\|_2$$

$$\geq \lambda_{\min}(\mathcal{I}(\boldsymbol{\theta})) - k \cdot \sqrt{\frac{1}{N}\lambda_{\max}(\mathcal{I}(\boldsymbol{h})_L)} \cdot \|\boldsymbol{\rho}\|_2$$

$$\geq \lambda_{\min}(\mathcal{I}(\boldsymbol{\theta})) - \lambda_{\min}(\mathcal{I}(\boldsymbol{\theta}))$$

$$= 0.$$

Thus with the specified probability, estimator $\hat{\mathcal{I}}_2(\boldsymbol{\theta})$ is a positive semidefinite matrix. $\qquad\square$

### B.11 Proof of Theorem 11

*Proof.* Let $\bar{\boldsymbol{t}}_a = \frac{1}{N}\sum_{i=1}^N\boldsymbol{t}_a(\boldsymbol{y}_i)$. As the FIM is a p.s.d. matrix, $\forall\boldsymbol{v}$ such that $\|\boldsymbol{v}\| = 1$, we have

$$\boldsymbol{v}^\top\hat{\mathcal{I}}_2(\boldsymbol{\theta})\boldsymbol{v} = \boldsymbol{v}^\top\left(\mathcal{I}(\boldsymbol{\theta}) - (\bar{\boldsymbol{t}}_a - \mathbb{E}[\bar{\boldsymbol{t}}_a])\frac{\partial^2\boldsymbol{h}_L^a(\boldsymbol{x})}{\partial\boldsymbol{\theta}\partial\boldsymbol{\theta}^\top}\right)\boldsymbol{v}$$

$$\geq -\boldsymbol{v}^\top\left((\bar{\boldsymbol{t}}_a - \mathbb{E}[\bar{\boldsymbol{t}}_a])\frac{\partial^2\boldsymbol{h}_L^a(\boldsymbol{x})}{\partial\boldsymbol{\theta}\partial\boldsymbol{\theta}^\top}\right)\boldsymbol{v}$$

$$= (\boldsymbol{\eta}_a - \bar{\boldsymbol{t}}_a) \cdot \boldsymbol{v}^\top\frac{\partial^2\boldsymbol{h}_L^a(\boldsymbol{x})}{\partial\boldsymbol{\theta}\partial\boldsymbol{\theta}^\top}\boldsymbol{v}$$

$$\geq -|\boldsymbol{\eta}_a - \bar{\boldsymbol{t}}_a| \cdot \rho(\partial^2\boldsymbol{h}_L^a) \cdot \boldsymbol{v}^\top\boldsymbol{v}$$

$$= -\rho(\partial^2\boldsymbol{h}_L^a)|\boldsymbol{\eta}_a - \bar{\boldsymbol{t}}_a|.$$

As $\hat{\mathcal{I}}_2(\boldsymbol{\theta})$ is a real symmetric matrix, we can write its spectrum decomposition as

$$\hat{\mathcal{I}}_2(\boldsymbol{\theta}) = \hat{\lambda}^\alpha\boldsymbol{v}_\alpha\boldsymbol{v}_\alpha^\top,$$

where $\{v_\alpha\}$ are orthonormal vectors and $\hat{\lambda}^\alpha$ are the corresponding eigenvalues of $\hat{\mathcal{I}}_2(\boldsymbol{\theta})$.

Therefore
$$\hat{\lambda}^\alpha = \boldsymbol{v}_\alpha^\top \hat{\mathcal{I}}_2(\boldsymbol{\theta})\boldsymbol{v}_\alpha \geq -\rho(\partial^2 \boldsymbol{h}_L^a)\,|\boldsymbol{\eta}_a - \bar{\boldsymbol{t}}_a|\,.$$

The statement in Theorem 11 follows immediately.

$\square$

## B.12 Proof of Lemma 12

To prove the Lemma, we first consider a variant of the result presented in Chen [8].

**Lemma A.1** (Variant of Chen [8]). *For any random vector $X \in \Re^n$ with variances* $\mathrm{Var}(X)$,

$$\Pr\left((X - \mathbb{E}X)^\top(X - \mathbb{E}X) \geq \varepsilon\right) \leq \frac{1}{\varepsilon} \cdot \sum_{i=1}^n \mathrm{Var}(X_i), \quad \forall \varepsilon > 0.$$

*Proof.* The proofs follows closely to Chen [8], with the slight change in set of variables considered. Let $\varepsilon > 0$ and $D_\varepsilon := \{V \in \Re^n : (V - \mathbb{E}X)^\top(V - \mathbb{E}X) \geq \varepsilon\}$. By definition we have that for all $V \in D_\varepsilon$,

$$(V - \mathbb{E}X)^\top(V - \mathbb{E}X) \cdot \frac{1}{\varepsilon} \geq 1$$

Thus for the probability of the set, we have:

$$\begin{aligned}
\Pr\left((X - \mathbb{E}X)^\top(X - \mathbb{E}X) \geq \varepsilon\right) &= \Pr\left(X \in D_\varepsilon\right) \\
&= \mathbb{E}\left[\mathbf{1}_{X \in D_\varepsilon}\right] \\
&\leq \frac{1}{\varepsilon} \cdot \mathbb{E}\left[(X - \mathbb{E}X)^\top(X - \mathbb{E}X) \cdot \mathbf{1}_{X \in D_\varepsilon}\right] \\
&\leq \frac{1}{\varepsilon} \cdot \mathbb{E}\left[(X - \mathbb{E}X)^\top(X - \mathbb{E}X)\right] \\
&= \frac{1}{\varepsilon} \cdot \sum_{i=1}^n \mathbb{E}\left[(X_i - \mathbb{E}X_i)(X_i - \mathbb{E}X_i)\right] \\
&= \frac{1}{\varepsilon} \cdot \sum_{i=1}^n \mathrm{Var}(X_i),
\end{aligned}$$

where $\mathbf{1}_{X \in S}$ denotes the indicator function for $X$ being in a set $S$.

$\square$

We can now prove the main Lemma through standard tricks:

*Proof.* From Lemma A.1 (and utilizing the vec operator) we have that for any $\delta > 0$ and each $z \in \{1, 2\}$:

$$\Pr(\|\hat{\mathcal{I}}_z(\boldsymbol{\theta}) - \mathcal{I}(\boldsymbol{\theta})\|_F^2 \leq \delta) \geq \frac{1}{\delta} \cdot \sum_{i,j=1}^{\dim(\boldsymbol{\theta})} \mathrm{Var}\left(\hat{\mathcal{I}}_z(\boldsymbol{\theta})\right)^{ij}.$$

Letting $\varepsilon := \frac{1}{\delta} \cdot \sum_{i,j=1}^{\dim(\boldsymbol{\theta})} \mathrm{Var}\left(\hat{\mathcal{I}}_z(\boldsymbol{\theta})\right)^{ij}$ and rearranging, we have:

$$\Pr\left(\|\hat{\mathcal{I}}_z(\boldsymbol{\theta}) - \mathcal{I}(\boldsymbol{\theta})\|_F^2 \geq \frac{1}{\epsilon} \cdot \sum_{i,j=1}^{\dim(\boldsymbol{\theta})} \mathrm{Var}\left(\hat{\mathcal{I}}_z(\boldsymbol{\theta})\right)^{ij}\right) \leq 1 - \varepsilon$$

$$\implies \Pr\left(\|\hat{\mathcal{I}}_z(\boldsymbol{\theta}) - \mathcal{I}(\boldsymbol{\theta})\|_F \geq \frac{1}{\sqrt{\epsilon}} \cdot \sqrt{\sum_{i,j=1}^{\dim(\boldsymbol{\theta})} \mathrm{Var}\left(\hat{\mathcal{I}}_z(\boldsymbol{\theta})\right)^{ij}}\right) \leq 1 - \varepsilon.$$

Substituting appropriately from Eq. (10) w.r.t. the FIM estimator considered completes the proof.

$\square$

### B.13 Proof of Lemma 13

*Proof.* We first show that $\boldsymbol{B}_l$ is the Jacobian of the mapping $\boldsymbol{h}_l \to \boldsymbol{h}_L$, that is,

$$d\boldsymbol{h}_L = \boldsymbol{B}_l d\boldsymbol{h}_l. \quad (l = 0, \cdots, L) \tag{A.2}$$

Obviously, this is true for $l = L$, as we have

$$d\boldsymbol{h}_L = \boldsymbol{I} \cdot d\boldsymbol{h}_L = \boldsymbol{B}_L \cdot d\boldsymbol{h}_L.$$

From Eq. (3), we have

$$\boldsymbol{h}_{l+1} = \sigma(\boldsymbol{W}_l \bar{\boldsymbol{h}}_l),$$

where $\sigma$ is abused to denote the non-linear activation function for $l = 0, \cdots, L - 2$ and $\sigma$ is the identity map for $l = L - 1$. Therefore

$$d\boldsymbol{h}_{l+1} = \boldsymbol{D}_l \cdot \boldsymbol{W}_l^- \cdot d\boldsymbol{h}_l,$$

where $\boldsymbol{D}_l = \operatorname{diag}\left(\sigma'(\boldsymbol{W}_l \bar{\boldsymbol{h}}_l)\right)$ for $l = 0, \cdots, L - 2$, and $\boldsymbol{D}_{L-1} = \boldsymbol{I}$. Assume Eq. (A.2) is true for $l + 1$, then

$$d\boldsymbol{h}_L = \boldsymbol{B}_{l+1} d\boldsymbol{h}_{l+1} = \boldsymbol{B}_{l+1} \cdot \boldsymbol{D}_l \cdot \boldsymbol{W}_l^- \cdot d\boldsymbol{h}_l = \boldsymbol{B}_l d\boldsymbol{h}_l.$$

Hence $\boldsymbol{B}_l$ is the Jacobian of $\boldsymbol{h}_l \to \boldsymbol{h}_L$ for $l = 0, \cdots, L$.

Now we are ready to derive the expression of $\frac{\partial \ell}{\partial \boldsymbol{h}_l}$.

$$
\begin{aligned}
d\ell &= d(\log p(\boldsymbol{y} \mid \boldsymbol{x})) \\
&= d(\boldsymbol{t}^\top(\boldsymbol{y})\boldsymbol{h}_L - F(\boldsymbol{h}_L)) \\
&= d(\operatorname{tr}\left\{\boldsymbol{t}^\top(\boldsymbol{y})\boldsymbol{h}_L - F(\boldsymbol{h}_L)\right\}) \\
&= \operatorname{tr}\left\{d(\boldsymbol{t}^\top(\boldsymbol{y})\boldsymbol{h}_L - F(\boldsymbol{h}_L))\right\} \\
&= \operatorname{tr}\left\{d(\boldsymbol{t}^\top(\boldsymbol{y})\boldsymbol{h}_L) - d(F(\boldsymbol{h}_L))\right\} \\
&= \operatorname{tr}\left\{\boldsymbol{t}^\top(\boldsymbol{y})d\boldsymbol{h}_L - \bigtriangledown F(\boldsymbol{h}_L)^\top d\boldsymbol{h}_L\right\} \\
&= \operatorname{tr}\left\{\boldsymbol{t}^\top(\boldsymbol{y})d\boldsymbol{h}_L - \boldsymbol{\eta}^\top(\boldsymbol{h}_L)d\boldsymbol{h}_L\right\} \\
&= \operatorname{tr}\left\{(\boldsymbol{t}(\boldsymbol{y}) - \boldsymbol{\eta}(\boldsymbol{h}_L))^\top d\boldsymbol{h}_L\right\} \\
&= \operatorname{tr}\left\{(\boldsymbol{t}(\boldsymbol{y}) - \boldsymbol{\eta}(\boldsymbol{h}_L))^\top \boldsymbol{B}_l d\boldsymbol{h}_l\right\}
\end{aligned}
$$

Therefore

$$\frac{\partial \ell}{\partial \boldsymbol{h}_l} = \left((\boldsymbol{t}(\boldsymbol{y}) - \boldsymbol{\eta}(\boldsymbol{h}_L))^\top \boldsymbol{B}_l\right)^\top = \boldsymbol{B}_l^\top (\boldsymbol{t}(\boldsymbol{y}) - \boldsymbol{\eta}(\boldsymbol{h}_L)).$$

Next, we show the gradient w.r.t. the neural network weights, *i.e.* $\frac{\partial \ell}{\partial \boldsymbol{W}_l}$. We have

$$
\begin{aligned}
d\ell &= \operatorname{tr}\left\{\left(\frac{\partial \ell}{\partial \boldsymbol{h}_{l+1}}\right)^\top d\boldsymbol{h}_{l+1}\right\} \\
&= \operatorname{tr}\left\{\left(\frac{\partial \ell}{\partial \boldsymbol{h}_{l+1}}\right)^\top \boldsymbol{D}_l \cdot d\boldsymbol{W}_l \cdot \bar{\boldsymbol{h}}_l\right\} \\
&= \operatorname{tr}\left\{\bar{\boldsymbol{h}}_l \left(\frac{\partial \ell}{\partial \boldsymbol{h}_{l+1}}\right)^\top \boldsymbol{D}_l \cdot d\boldsymbol{W}_l\right\}.
\end{aligned}
$$

Therefore

$$
\begin{aligned}
\frac{\partial \ell}{\partial \boldsymbol{W}_l} &= \left(\bar{\boldsymbol{h}}_l \left(\frac{\partial \ell}{\partial \boldsymbol{h}_{l+1}}\right)^\top \boldsymbol{D}_l\right)^\top \\
&= \boldsymbol{D}_l^\top \left(\frac{\partial \ell}{\partial \boldsymbol{h}_{l+1}}\right) \bar{\boldsymbol{h}}_l^\top \\
&= \boldsymbol{D}_l \left(\frac{\partial \ell}{\partial \boldsymbol{h}_{l+1}}\right) \bar{\boldsymbol{h}}_l^\top.
\end{aligned}
$$

Finally, we give the gradient of $\boldsymbol{h}_L$ w.r.t. the neural network parameters. By definition, $\boldsymbol{B}_{l+1}$ is the Jacobian of the mapping $\boldsymbol{h}_{l+1} \to \boldsymbol{h}_L$. Therefore

$$d\boldsymbol{h}_L = \boldsymbol{B}_{l+1} \cdot d\boldsymbol{h}_{l+1} = \boldsymbol{B}_{l+1} \cdot \boldsymbol{D}_l d\boldsymbol{W}_l \bar{\boldsymbol{h}}_l.$$

We rewrite the above equation in the element-wise form

$$
\begin{aligned}
d\boldsymbol{h}_L^a &= \mathrm{tr}\left\{\boldsymbol{B}_{l+1} \cdot \boldsymbol{D}_l d\boldsymbol{W}_l \bar{\boldsymbol{h}}_l \boldsymbol{e}_a^\top\right\} \\
&= \mathrm{tr}\left\{\bar{\boldsymbol{h}}_l \boldsymbol{e}_a^\top \boldsymbol{B}_{l+1} \boldsymbol{D}_l \cdot d\boldsymbol{W}_l\right\}.
\end{aligned}
$$

Therefore

$$
\begin{aligned}
\frac{\partial \boldsymbol{h}_L^a}{\partial \boldsymbol{W}_l} &= \left(\bar{\boldsymbol{h}}_l \boldsymbol{e}_a^\top \boldsymbol{B}_{l+1} \boldsymbol{D}_l\right)^\top \\
&= \boldsymbol{D}_l^\top \boldsymbol{B}_{l+1}^\top \boldsymbol{e}_a \bar{\boldsymbol{h}}_l^\top \\
&= \boldsymbol{D}_l \boldsymbol{B}_{l+1}^\top \boldsymbol{e}_a \bar{\boldsymbol{h}}_l^\top.
\end{aligned}
$$

$\square$

## B.14 Proof of Lemma 14

*Proof.* First, we notice that

$$\frac{\partial \boldsymbol{h}_L^a}{\partial \boldsymbol{W}_l} = \boldsymbol{D}_l \boldsymbol{B}_{l+1}^\top \boldsymbol{e}_a \cdot \bar{\boldsymbol{h}}_l^\top$$

has rank one. Therefore

$$\left\|\frac{\partial \boldsymbol{h}_L^a}{\partial \boldsymbol{W}_l}\right\|_F = \|\boldsymbol{D}_l \boldsymbol{B}_{l+1}^\top \boldsymbol{e}_a\|_2 \cdot \|\bar{\boldsymbol{h}}_l\|_2.$$

Therefore

$$
\begin{aligned}
\left\|\frac{\partial \boldsymbol{h}_L}{\partial \boldsymbol{W}_l}\right\|_F &= \sqrt{\sum_a \left\|\frac{\partial \boldsymbol{h}_L^a}{\partial \boldsymbol{W}_l}\right\|_F^2} \\
&= \sqrt{\sum_a \|\boldsymbol{D}_l \boldsymbol{B}_{l+1}^\top \boldsymbol{e}_a\|_2^2 \cdot \|\bar{\boldsymbol{h}}_l\|_2^2} \\
&= \sqrt{\sum_a \|\boldsymbol{D}_l \boldsymbol{B}_{l+1}^\top \boldsymbol{e}_a\|_2^2} \cdot \|\bar{\boldsymbol{h}}_l\|_2 \\
&= \|\boldsymbol{D}_l \boldsymbol{B}_{l+1}^\top\|_F \cdot \|\bar{\boldsymbol{h}}_l\|_2 \\
&= \|\boldsymbol{B}_{l+1} \boldsymbol{D}_l\|_F \cdot \|\bar{\boldsymbol{h}}_l\|_2.
\end{aligned}
$$

The matrix $\boldsymbol{D}_l$ is diagonal, with entries bounded in the range $[-1, 1]$. Therefore a left or right multiplication by $\boldsymbol{D}_l$ does not increase the Frobenius norm. Hence

$$\left\|\frac{\partial \boldsymbol{h}_L}{\partial \boldsymbol{W}_l}\right\|_F \le \|\boldsymbol{B}_{l+1}\|_F \cdot \|\bar{\boldsymbol{h}}_l\|_2.$$

By the recursive definition of $\boldsymbol{B}_l$, we have

$$\boldsymbol{B}_l = \boldsymbol{B}_{l+1} \boldsymbol{D}_l \boldsymbol{W}_l^-.$$

Therefore

$$\|\boldsymbol{B}_l\|_F \le \|\boldsymbol{B}_{l+1}\|_F \cdot \|\boldsymbol{D}_l \boldsymbol{W}_l^-\|_F \le \|\boldsymbol{B}_{l+1}\|_F \cdot \|\boldsymbol{W}_l^-\|_F.$$

Applying the above inequality repeatedly leads to

$$\|\boldsymbol{B}_l\|_F \leq \|\boldsymbol{B}_{L-1}\|_F \cdot \prod_{i=l}^{L-2} \|\boldsymbol{W}_i^-\|_F$$

$$= \|\boldsymbol{B}_L \boldsymbol{D}_{L-1} \boldsymbol{W}_{L-1}^-\|_F \cdot \prod_{i=l}^{L-2} \|\boldsymbol{W}_i^-\|_F$$

$$= \|\boldsymbol{W}_{L-1}^-\|_F \cdot \prod_{i=l}^{L-2} \|\boldsymbol{W}_i^-\|_F$$

$$= \prod_{i=l}^{L-1} \|\boldsymbol{W}_i^-\|_F.$$

Hence

$$\left\| \frac{\partial \boldsymbol{h}_L}{\partial \boldsymbol{W}_l} \right\|_F \leq \|\boldsymbol{B}_{l+1}\|_F \cdot \|\bar{\boldsymbol{h}}_l\|_2$$

$$\leq \prod_{i=l+1}^{L-1} \|\boldsymbol{W}_i^-\|_F \cdot \|\bar{\boldsymbol{h}}_l\|_2.$$

$\square$

## B.15  Proof of Lemma 15

*Proof.* Recall the 2-spectral norm for a $d$-dimensional tensor is defined by

$$\|\mathcal{T}\|_{2_\sigma} = \max \left\{ \langle \mathcal{T}, \boldsymbol{x}^1 \otimes \ldots \otimes \boldsymbol{x}^d \rangle : \|\boldsymbol{x}^k\|_2 = 1, \ \forall k \in [d] \right\}.$$

Let $\boldsymbol{\alpha}, \boldsymbol{\beta}, \boldsymbol{\gamma}$ be unit vectors, so that

$$\|\boldsymbol{\alpha}\|_2 = 1, \quad \|\boldsymbol{\beta}\|_2 = 1, \quad \|\boldsymbol{\gamma}\|_2 = 1.$$

Then

$$\max_{\boldsymbol{\alpha},\boldsymbol{\beta},\boldsymbol{\gamma}} \left\langle \frac{\partial \boldsymbol{h}_L}{\partial \boldsymbol{W}_l}, \boldsymbol{\alpha} \otimes \boldsymbol{\beta} \otimes \boldsymbol{\gamma} \right\rangle = \max_{\boldsymbol{\alpha},\boldsymbol{\beta},\boldsymbol{\gamma}} \left\langle \sum_a \frac{\partial h_L^a}{\partial \boldsymbol{W}_l} \gamma_a, \boldsymbol{\alpha} \otimes \boldsymbol{\beta} \right\rangle$$

$$= \max_{\boldsymbol{\alpha},\boldsymbol{\beta},\boldsymbol{\gamma}} \sum_a \boldsymbol{\alpha}^\top \boldsymbol{D}_l \boldsymbol{B}_{l+1}^\top \boldsymbol{e}_a \bar{\boldsymbol{h}}_l^\top \boldsymbol{\beta} \gamma_a$$

$$= \max_{\boldsymbol{\alpha},\boldsymbol{\beta},\boldsymbol{\gamma}} \boldsymbol{\alpha}^\top \boldsymbol{D}_l \boldsymbol{B}_{l+1}^\top (\sum_a \gamma_a \boldsymbol{e}_a)(\bar{\boldsymbol{h}}_l^\top \boldsymbol{\beta})$$

$$= \max_{\boldsymbol{\alpha},\boldsymbol{\beta},\boldsymbol{\gamma}} \boldsymbol{\alpha}^\top \boldsymbol{D}_l \boldsymbol{B}_{l+1}^\top \boldsymbol{\gamma} (\bar{\boldsymbol{h}}_l^\top \boldsymbol{\beta})$$

$$= \max_{\boldsymbol{\alpha},\boldsymbol{\beta},\boldsymbol{\gamma}} \langle \boldsymbol{B}_{l+1} \boldsymbol{D}_l \boldsymbol{\alpha}, \boldsymbol{\gamma} \rangle \cdot \langle \bar{\boldsymbol{h}}_l, \boldsymbol{\beta} \rangle$$

$$= \max_{\boldsymbol{\alpha}} \|\boldsymbol{B}_{l+1} \boldsymbol{D}_l \boldsymbol{\alpha}\|_2 \cdot \|\bar{\boldsymbol{h}}_l\|_2.$$

Recall that

$$\boldsymbol{B}_l = \boldsymbol{B}_{l+1} \boldsymbol{D}_l \boldsymbol{W}_l^-.$$

Therefore

$$\boldsymbol{B}_l \boldsymbol{D}_{l-1} \boldsymbol{\alpha} = \boldsymbol{B}_{l+1} \boldsymbol{D}_l \boldsymbol{W}_l^- \boldsymbol{D}_{l-1} \boldsymbol{\alpha}.$$

Moreover,

$$\max_{\boldsymbol{\alpha}} \|\boldsymbol{B}_l \boldsymbol{D}_{l-1} \boldsymbol{\alpha}\|_2 = \max_{\boldsymbol{\alpha}} \|\boldsymbol{B}_{l+1} \boldsymbol{D}_l \cdot \boldsymbol{W}_l^- \boldsymbol{D}_{l-1} \boldsymbol{\alpha}\|_2$$

$$\leq \max_{\boldsymbol{\alpha}'} s_{\max}(\boldsymbol{W}_l^-) \|\boldsymbol{B}_{l+1} \boldsymbol{D}_l \boldsymbol{\alpha}'\|_2$$

$$= s_{\max}(\boldsymbol{W}_l^-) \max_{\boldsymbol{\alpha}'} \|\boldsymbol{B}_{l+1} \boldsymbol{D}_l \boldsymbol{\alpha}'\|_2.$$

Hence,

$$\max_{\boldsymbol{\alpha},\boldsymbol{\beta},\boldsymbol{\gamma}} \left\langle \frac{\partial \boldsymbol{h}_L}{\partial \boldsymbol{W}_l}, \boldsymbol{\alpha} \otimes \boldsymbol{\beta} \otimes \boldsymbol{\gamma} \right\rangle = \max_{\boldsymbol{\alpha}} \|\boldsymbol{B}_{l+1}\boldsymbol{D}_l\boldsymbol{\alpha}\|_2 \cdot \|\bar{\boldsymbol{h}}_l\|_2$$

$$\leq \prod_{i=l+1}^{L-1} s_{\max}(\boldsymbol{W}_i^-) \max_{\boldsymbol{\alpha}'} \|\boldsymbol{B}_L\boldsymbol{D}_{L-1}\boldsymbol{\alpha}'\|_2 \cdot \|\bar{\boldsymbol{h}}_l\|_2$$

$$= \prod_{i=l+1}^{L-1} s_{\max}(\boldsymbol{W}_i^-) \max_{\boldsymbol{\alpha}} \|\boldsymbol{B}_L\boldsymbol{D}_{L-1}\boldsymbol{\alpha}\|_2 \cdot \|\bar{\boldsymbol{h}}_l\|_2$$

$$= \prod_{i=l+1}^{L-1} s_{\max}(\boldsymbol{W}_i^-) \max_{\boldsymbol{\alpha}} \|\boldsymbol{I}\boldsymbol{I}\boldsymbol{\alpha}\|_2 \cdot \|\bar{\boldsymbol{h}}_l\|_2$$

$$= \prod_{i=l+1}^{L-1} s_{\max}(\boldsymbol{W}_i^-) \max_{\boldsymbol{\alpha}} \|\boldsymbol{\alpha}\|_2 \cdot \|\bar{\boldsymbol{h}}_l\|_2$$

$$= \prod_{i=l+1}^{L-1} s_{\max}(\boldsymbol{W}_i^-) \cdot \|\bar{\boldsymbol{h}}_l\|_2.$$

$\square$

## C   Change of Coordinates and Covariance

Reparametrization is a common technique in deep learning. See *e.g.* weight normalization [28]. From a geometric perspective, reparametrization corresponds to change of coordinates. We consider how our results can be generalized in a new coordinate system $\{\boldsymbol{\xi}_i\}$ instead of the default coordinates $\{\boldsymbol{\theta}_\alpha\}$. By definition, the FIM w.r.t. to the new coordinates is

$$\mathcal{I}(\boldsymbol{\xi}) = \frac{\partial \boldsymbol{\theta}_a}{\partial \boldsymbol{\xi}} \frac{\partial \boldsymbol{h}_L^\alpha}{\partial \boldsymbol{\theta}^a} \mathcal{I}_{\alpha\beta}(\boldsymbol{h}_L) \frac{\partial \boldsymbol{h}_L^\beta}{\partial \boldsymbol{\theta}^b} \frac{\partial \boldsymbol{\theta}_b}{\partial \boldsymbol{\xi}^\top} = \frac{\partial \boldsymbol{\theta}_a}{\partial \boldsymbol{\xi}} \mathcal{I}^{ab}(\boldsymbol{\theta}) \frac{\partial \boldsymbol{\theta}_b}{\partial \boldsymbol{\xi}^\top}. \tag{A.3}$$

The FIM $\mathcal{I}(\boldsymbol{\xi})$ can be estimated by Eq. (2), where the estimators are denoted by $\hat{\mathcal{I}}_1(\boldsymbol{\xi})$ and $\hat{\mathcal{I}}_2(\boldsymbol{\xi})$.

Note that as per the main text, upper- and lower-indices in the Einstein are equivalent. Similarly, we take an upper index for derivatives of $\boldsymbol{h}_L$ w.r.t. $\boldsymbol{\theta}$ in this section — which we take as equivalent to the lower-index notation appearing in the main text. That is $\partial^\beta \boldsymbol{h}_L = \partial \boldsymbol{h}_L / \partial \boldsymbol{\theta}^\beta = \partial \boldsymbol{h}_L / \partial \boldsymbol{\theta}_\beta$.

**Theorem A.2.** *Consider the FIM estimators under the coordinate transformation $\boldsymbol{\theta} \to \boldsymbol{\xi}$ with the same samples size $N$.*

$$\hat{\mathcal{I}}_1(\boldsymbol{\xi}) = \frac{\partial \boldsymbol{\theta}_a}{\partial \boldsymbol{\xi}} \hat{\mathcal{I}}_1^{ab}(\boldsymbol{\theta}) \frac{\partial \boldsymbol{\theta}_b}{\partial \boldsymbol{\xi}^\top}, \tag{A.4}$$

$$\hat{\mathcal{I}}_2(\boldsymbol{\xi}) = \frac{\partial \boldsymbol{\theta}_a}{\partial \boldsymbol{\xi}} \hat{\mathcal{I}}_2^{ab}(\boldsymbol{\theta}) \frac{\partial \boldsymbol{\theta}_b}{\partial \boldsymbol{\xi}^\top} + \left( \eta_\alpha - \frac{1}{N}\sum_{i=1}^N \boldsymbol{t}_\alpha(\boldsymbol{y}_i) \right) \partial^\beta \boldsymbol{h}_L^\alpha \frac{\partial^2 \boldsymbol{\theta}_\beta}{\partial \boldsymbol{\xi}\partial \boldsymbol{\xi}^\top}. \tag{A.5}$$

*Proof.* Let $\hat{\mathbb{E}}[X]$ denote the empirical expectation of random variable $X$.

For the first estimator, consider the partial derivative of the log-likelihood:

$$\frac{\partial \ell}{\partial \boldsymbol{\xi}} = \left(\frac{\partial \boldsymbol{\theta}}{\partial \boldsymbol{\xi}}\right)^\top \left(\frac{\partial \ell}{\partial \boldsymbol{\theta}}\right)$$

$$= \frac{\partial \boldsymbol{\theta}_a}{\partial \boldsymbol{\xi}} \frac{\partial \boldsymbol{h}_L^\alpha}{\partial \boldsymbol{\theta}^a} (\boldsymbol{t}_\alpha(\boldsymbol{y}) - \eta_\alpha).$$

Thus the first FIM estimator follows immediately:

$$
\hat{\mathbb{E}}\left[\frac{\partial \ell}{\partial \boldsymbol{\xi}}\frac{\partial \ell}{\partial \boldsymbol{\xi}^T}\right] = \hat{\mathbb{E}}\left[\frac{\partial \boldsymbol{\theta}_a}{\partial \boldsymbol{\xi}}\frac{\partial \boldsymbol{h}_L^\alpha}{\partial \boldsymbol{\theta}^a}(\boldsymbol{t}_\alpha(\boldsymbol{y}) - \boldsymbol{\eta}_\alpha)\frac{\partial \boldsymbol{\theta}_b}{\partial \boldsymbol{\xi}^\top}\frac{\partial \boldsymbol{h}_L^\beta}{\partial \boldsymbol{\theta}^b}(\boldsymbol{t}_\beta(\boldsymbol{y}) - \boldsymbol{\eta}_\beta)\right]
$$

$$
= \frac{\partial \boldsymbol{\theta}_a}{\partial \boldsymbol{\xi}}\hat{\mathbb{E}}\left[\frac{\partial \boldsymbol{h}_L^\alpha}{\partial \boldsymbol{\theta}^a}(\boldsymbol{t}_\alpha(\boldsymbol{y}) - \boldsymbol{\eta}_\alpha)\frac{\partial \boldsymbol{h}_L^\beta}{\partial \boldsymbol{\theta}^b}(\boldsymbol{t}_\beta(\boldsymbol{y}) - \boldsymbol{\eta}_\beta)\right]\frac{\partial \boldsymbol{\theta}_b}{\partial \boldsymbol{\xi}^\top}
$$

$$
= \frac{\partial \boldsymbol{\theta}_a}{\partial \boldsymbol{\xi}}[\hat{\mathcal{I}}_1(\boldsymbol{\theta})]^{ab}\frac{\partial \boldsymbol{\theta}_b}{\partial \boldsymbol{\xi}^\top}.
$$

For the second estimator, we consider the second derivative:

$$
\frac{\partial^2 \ell}{\partial \boldsymbol{\xi}\partial \boldsymbol{\xi}^\top} = \frac{\partial}{\partial \boldsymbol{\xi}}\left[\frac{\partial \boldsymbol{\theta}_a}{\partial \boldsymbol{\xi}^\top}\frac{\partial \boldsymbol{h}_L^\alpha}{\partial \boldsymbol{\theta}^a}(\boldsymbol{t}_\alpha(\boldsymbol{y}) - \boldsymbol{\eta}_\alpha)\right]
$$

$$
= -\frac{\partial \boldsymbol{\eta}_\alpha}{\partial \boldsymbol{\xi}}\frac{\partial \boldsymbol{h}_L^\alpha}{\partial \boldsymbol{\theta}^a}\frac{\partial \boldsymbol{\theta}_a}{\partial \boldsymbol{\xi}^\top} + (\boldsymbol{t}_\alpha - \boldsymbol{\eta}_\alpha)\left[\frac{\partial}{\partial \boldsymbol{\xi}}\left(\frac{\partial \boldsymbol{h}_L^\alpha}{\partial \boldsymbol{\theta}^a}\right)\right]\frac{\partial \boldsymbol{\theta}_a}{\partial \boldsymbol{\xi}^\top} + (\boldsymbol{t}_\alpha - \boldsymbol{\eta}_\alpha)\frac{\partial \boldsymbol{h}_L^\alpha}{\partial \boldsymbol{\theta}^a}\frac{\partial^2 \boldsymbol{\theta}_a}{\partial \boldsymbol{\xi}\partial \boldsymbol{\xi}^\top}
$$

$$
= -\frac{\partial \boldsymbol{\theta}_b}{\partial \boldsymbol{\xi}}\left[\frac{\partial [\boldsymbol{h}_L]_\beta}{\partial \boldsymbol{\theta}^b}\frac{\partial \boldsymbol{\eta}_\alpha}{\partial \boldsymbol{h}_L^\beta}\frac{\partial \boldsymbol{h}_L^\alpha}{\partial \boldsymbol{\theta}^a} - (\boldsymbol{t}_\alpha - \boldsymbol{\eta}_\alpha)\frac{\partial^2 \boldsymbol{h}_L^\alpha}{\partial \boldsymbol{\theta}^b\partial \boldsymbol{\theta}^a}\right]\frac{\partial \boldsymbol{\theta}_a}{\partial \boldsymbol{\xi}^\top} + (\boldsymbol{t}_\alpha - \boldsymbol{\eta}_\alpha)\frac{\partial \boldsymbol{h}_L^\alpha}{\partial \boldsymbol{\theta}^a}\frac{\partial^2 \boldsymbol{\theta}_a}{\partial \boldsymbol{\xi}\partial \boldsymbol{\xi}^\top}.
$$

Taking the empirical expectation we have:

$$
\hat{\mathbb{E}}\left[-\frac{\partial^2 \ell}{\partial \boldsymbol{\xi}\partial \boldsymbol{\xi}^\top}\right]
$$

$$
= \hat{\mathbb{E}}\left[\frac{\partial \boldsymbol{\theta}_b}{\partial \boldsymbol{\xi}}\left[\frac{\partial [\boldsymbol{h}_L]_\beta}{\partial \boldsymbol{\theta}^b}\frac{\partial \boldsymbol{\eta}_\alpha}{\partial \boldsymbol{h}_L^\beta}\frac{\partial \boldsymbol{h}_L^\alpha}{\partial \boldsymbol{\theta}^a} - (\boldsymbol{t}_\alpha - \boldsymbol{\eta}_\alpha)\frac{\partial^2 \boldsymbol{h}_L^\alpha}{\partial \boldsymbol{\theta}^b\partial \boldsymbol{\theta}^a}\right]\frac{\partial \boldsymbol{\theta}_a}{\partial \boldsymbol{\xi}^\top} - (\boldsymbol{t}_\alpha - \boldsymbol{\eta}_\alpha)\frac{\partial \boldsymbol{h}_L^\alpha}{\partial \boldsymbol{\theta}^a}\frac{\partial^2 \boldsymbol{\theta}_a}{\partial \boldsymbol{\xi}\partial \boldsymbol{\xi}^\top}\right]
$$

$$
= \frac{\partial \boldsymbol{\theta}_b}{\partial \boldsymbol{\xi}}\hat{\mathbb{E}}\left[\frac{\partial [\boldsymbol{h}_L]_\beta}{\partial \boldsymbol{\theta}^b}\frac{\partial \boldsymbol{\eta}_\alpha}{\partial \boldsymbol{h}_L^\beta}\frac{\partial \boldsymbol{h}_L^\alpha}{\partial \boldsymbol{\theta}^a} - (\boldsymbol{t}_\alpha - \boldsymbol{\eta}_\alpha)\frac{\partial^2 \boldsymbol{h}_L^\alpha}{\partial \boldsymbol{\theta}^b\partial \boldsymbol{\theta}^a}\right]\frac{\partial \boldsymbol{\theta}_a}{\partial \boldsymbol{\xi}^\top} - \hat{\mathbb{E}}\left[(\boldsymbol{t}_\alpha - \boldsymbol{\eta}_\alpha)\frac{\partial \boldsymbol{h}_L^\alpha}{\partial \boldsymbol{\theta}^a}\frac{\partial^2 \boldsymbol{\theta}_a}{\partial \boldsymbol{\xi}\partial \boldsymbol{\xi}^\top}\right]
$$

$$
= \frac{\partial \boldsymbol{\theta}_b}{\partial \boldsymbol{\xi}}[\hat{\mathcal{I}}_2(\boldsymbol{\theta})]^{ba}\frac{\partial \boldsymbol{\theta}_a}{\partial \boldsymbol{\xi}^\top} - \hat{\mathbb{E}}\left[(\boldsymbol{t}_\alpha - \boldsymbol{\eta}_\alpha)\frac{\partial \boldsymbol{h}_L^\alpha}{\partial \boldsymbol{\theta}^a}\frac{\partial^2 \boldsymbol{\theta}_a}{\partial \boldsymbol{\xi}\partial \boldsymbol{\xi}^\top}\right]
$$

$$
= \frac{\partial \boldsymbol{\theta}_a}{\partial \boldsymbol{\xi}}[\hat{\mathcal{I}}_2(\boldsymbol{\theta})]^{ab}\frac{\partial \boldsymbol{\theta}_b}{\partial \boldsymbol{\xi}^\top} + \left(\boldsymbol{\eta}_\alpha - \frac{1}{N}\sum_{i=1}^N \boldsymbol{t}_\alpha(\boldsymbol{y}_i)\right)\frac{\partial \boldsymbol{h}_L^\alpha}{\partial \boldsymbol{\theta}^a}\frac{\partial^2 \boldsymbol{\theta}_a}{\partial \boldsymbol{\xi}\partial \boldsymbol{\xi}^\top}
$$

$$
= \frac{\partial \boldsymbol{\theta}_a}{\partial \boldsymbol{\xi}}[\hat{\mathcal{I}}_2(\boldsymbol{\theta})]^{ab}\frac{\partial \boldsymbol{\theta}_b}{\partial \boldsymbol{\xi}^\top} + \left(\boldsymbol{\eta}_\alpha - \frac{1}{N}\sum_{i=1}^N \boldsymbol{t}_\alpha(\boldsymbol{y}_i)\right)\frac{\partial \boldsymbol{h}_L^\alpha}{\partial \boldsymbol{\theta}^\beta}\frac{\partial^2 \boldsymbol{\theta}_\beta}{\partial \boldsymbol{\xi}\partial \boldsymbol{\xi}^\top}.
$$

Thus we have both estimators for the theorem. $\qquad\square$

Interestingly, the way to compute $\hat{\mathcal{I}}_1(\boldsymbol{\xi})$ under coordinate transformation follows the same rule to compute the FIM $\mathcal{I}(\boldsymbol{\xi})$ in the new coordinate system. See the similarity between Eq. (A.3) and Eq. (A.4). The transformation rule of $\hat{\mathcal{I}}_2(\boldsymbol{\xi})$ introduces an additional term, which depends on the Hessian of the coordinate transform $\partial^2 \boldsymbol{\theta}_\beta / \partial \boldsymbol{\xi}\partial \boldsymbol{\xi}^\top$. This term vanishes as the number of samples increases $N \to \infty$, or the transformation $\boldsymbol{\theta} \to \boldsymbol{\xi}$ is affine.

The results we need to generalize for coordinate transformation depend on the (co)variance of the estimators. As such, we present a $\{\boldsymbol{\xi}_i\}$ variant of Theorems 4 and 6.

**Theorem A.3.**

$$
\left[\mathrm{Cov}\left(\hat{\mathcal{I}}_1(\boldsymbol{\xi})\right)\right]^{ijkl} = \frac{\partial \boldsymbol{\theta}_a}{\partial \boldsymbol{\xi}_i}\frac{\partial \boldsymbol{\theta}_b}{\partial \boldsymbol{\xi}_j}\frac{\partial \boldsymbol{\theta}_c}{\partial \boldsymbol{\xi}_k}\frac{\partial \boldsymbol{\theta}_d}{\partial \boldsymbol{\xi}_l}\left[\mathrm{Cov}\left(\hat{\mathcal{I}}_1(\boldsymbol{\theta})\right)\right]^{abcd} \tag{A.6}
$$

$$
\left[\mathrm{Cov}\left(\hat{\mathcal{I}}_2(\boldsymbol{\xi})\right)\right]^{ijkl} = \frac{\partial \boldsymbol{\theta}_a}{\partial \boldsymbol{\xi}_i}\frac{\partial \boldsymbol{\theta}_b}{\partial \boldsymbol{\xi}_j}\frac{\partial \boldsymbol{\theta}_c}{\partial \boldsymbol{\xi}_k}\frac{\partial \boldsymbol{\theta}_d}{\partial \boldsymbol{\xi}_l}\left[\mathrm{Cov}\left(\hat{\mathcal{I}}_2(\boldsymbol{\theta})\right)\right]^{abcd} + \frac{1}{N}\mathcal{C}^{\alpha\beta}\mathcal{I}_{\alpha\beta}(\boldsymbol{h}_L), \tag{A.7}
$$

*where* $\mathcal{C}^{\alpha\beta} :=$

$$
\frac{\partial \boldsymbol{\theta}_a}{\partial \boldsymbol{\xi}_i}\frac{\partial \boldsymbol{\theta}_b}{\partial \boldsymbol{\xi}_j}\frac{\partial^2 \boldsymbol{\theta}_c}{\partial \boldsymbol{\xi}_k\partial \boldsymbol{\xi}_l}\partial^a\partial^b \boldsymbol{h}_L^\alpha \partial^c \boldsymbol{h}_L^\beta + \frac{\partial^2 \boldsymbol{\theta}_a}{\partial \boldsymbol{\xi}_i\partial \boldsymbol{\xi}_j}\frac{\partial \boldsymbol{\theta}_b}{\partial \boldsymbol{\xi}_k}\frac{\partial \boldsymbol{\theta}_c}{\partial \boldsymbol{\xi}_l}\partial^a \boldsymbol{h}_L^\alpha \partial^b\partial^c \boldsymbol{h}_L^\beta + \frac{\partial^2 \boldsymbol{\theta}_a}{\partial \boldsymbol{\xi}_i\partial \boldsymbol{\xi}_j}\frac{\partial^2 \boldsymbol{\theta}_b}{\partial \boldsymbol{\xi}_k\partial \boldsymbol{\xi}_l}\partial^a \boldsymbol{h}_L^\alpha \partial^b \boldsymbol{h}_L^\beta.
$$

*Proof.* For the first estimator's covariance, immediately get the result by the way covariance interacts with constant products:

$$\mathrm{Cov}\left([\hat{\mathcal{I}}_1(\boldsymbol{\xi})]^{ij},[\hat{\mathcal{I}}_1(\boldsymbol{\xi})]^{kl}\right) = \mathrm{Cov}\left(\frac{\partial\boldsymbol{\theta}_a}{\partial\boldsymbol{\xi}_i}[\hat{\mathcal{I}}_1(\boldsymbol{\theta})]^{ab}\frac{\partial\boldsymbol{\theta}_b}{\partial\boldsymbol{\xi}_j},\frac{\partial\boldsymbol{\theta}_c}{\partial\boldsymbol{\xi}_k}[\hat{\mathcal{I}}_1(\boldsymbol{\theta})]^{cd}\frac{\partial\boldsymbol{\theta}_d}{\partial\boldsymbol{\xi}_l}\right)$$

$$= \frac{\partial\boldsymbol{\theta}_a}{\partial\boldsymbol{\xi}_i}\frac{\partial\boldsymbol{\theta}_b}{\partial\boldsymbol{\xi}_j}\frac{\partial\boldsymbol{\theta}_c}{\partial\boldsymbol{\xi}_k}\frac{\partial\boldsymbol{\theta}_d}{\partial\boldsymbol{\xi}_l}\mathrm{Cov}\left([\hat{\mathcal{I}}_1(\boldsymbol{\theta})]^{ab},[\hat{\mathcal{I}}_1(\boldsymbol{\theta})]^{cd}\right)$$

For the second estimator, we must exploit the linear combination property of covariances as well:

$$\mathrm{Cov}\left([\hat{\mathcal{I}}_2(\boldsymbol{\xi})]^{ij},[\hat{\mathcal{I}}_2(\boldsymbol{\xi})]^{kl}\right)$$

$$= \mathrm{Cov}\left(\frac{\partial\boldsymbol{\theta}_a}{\partial\boldsymbol{\xi}_i}[\hat{\mathcal{I}}_2(\boldsymbol{\theta})]^{ab}\frac{\partial\boldsymbol{\theta}_b}{\partial\boldsymbol{\xi}_j} + \left(\boldsymbol{\eta}_\alpha - \frac{1}{N}\sum_{i=1}\boldsymbol{t}_\alpha(\boldsymbol{y}_i)\right)\frac{\partial h_L^\alpha}{\partial\boldsymbol{\theta}^a}\frac{\partial^2\boldsymbol{\theta}_a}{\partial\boldsymbol{\xi}_i\partial\boldsymbol{\xi}_j},\right.$$

$$\left.\frac{\partial\boldsymbol{\theta}_c}{\partial\boldsymbol{\xi}_k}[\hat{\mathcal{I}}_2(\boldsymbol{\theta})]^{cd}\frac{\partial\boldsymbol{\theta}_d}{\partial\boldsymbol{\xi}_l} + \left(\boldsymbol{\eta}_\beta - \frac{1}{N}\sum_{i=1}\boldsymbol{t}_\beta(\boldsymbol{y}_i)\right)\frac{\partial h_L^\beta}{\partial\boldsymbol{\theta}^b}\frac{\partial^2\boldsymbol{\theta}_b}{\partial\boldsymbol{\xi}_k\partial\boldsymbol{\xi}_l}\right)$$

$$= \mathrm{Cov}\left(\frac{\partial\boldsymbol{\theta}_a}{\partial\boldsymbol{\xi}_i}[\hat{\mathcal{I}}_2(\boldsymbol{\theta})]^{ab}\frac{\partial\boldsymbol{\theta}_b}{\partial\boldsymbol{\xi}_j},\frac{\partial\boldsymbol{\theta}_c}{\partial\boldsymbol{\xi}_k}[\hat{\mathcal{I}}_2(\boldsymbol{\theta})]^{cd}\frac{\partial\boldsymbol{\theta}_d}{\partial\boldsymbol{\xi}_l}\right)$$

$$+ \mathrm{Cov}\left(\frac{\partial\boldsymbol{\theta}_a}{\partial\boldsymbol{\xi}_i}[\hat{\mathcal{I}}_2(\boldsymbol{\theta})]^{ab}\frac{\partial\boldsymbol{\theta}_b}{\partial\boldsymbol{\xi}_j},\left(\boldsymbol{\eta}_\beta - \frac{1}{N}\sum_{i=1}\boldsymbol{t}_\beta(\boldsymbol{y}_i)\right)\frac{\partial h_L^\beta}{\partial\boldsymbol{\theta}^b}\frac{\partial^2\boldsymbol{\theta}_b}{\partial\boldsymbol{\xi}_k\partial\boldsymbol{\xi}_l}\right)$$

$$+ \mathrm{Cov}\left(\left(\boldsymbol{\eta}_\alpha - \frac{1}{N}\sum_{i=1}\boldsymbol{t}_\alpha(\boldsymbol{y}_i)\right)\frac{\partial h_L^\alpha}{\partial\boldsymbol{\theta}^a}\frac{\partial^2\boldsymbol{\theta}_a}{\partial\boldsymbol{\xi}_i\partial\boldsymbol{\xi}_j},\frac{\partial\boldsymbol{\theta}_c}{\partial\boldsymbol{\xi}_k}[\hat{\mathcal{I}}_2(\boldsymbol{\theta})]^{cd}\frac{\partial\boldsymbol{\theta}_d}{\partial\boldsymbol{\xi}_l}\right)$$

$$+ \mathrm{Cov}\left(\left(\boldsymbol{\eta}_\alpha - \frac{1}{N}\sum_{i=1}\boldsymbol{t}_\alpha(\boldsymbol{y}_i)\right)\frac{\partial h_L^\alpha}{\partial\boldsymbol{\theta}^a}\frac{\partial^2\boldsymbol{\theta}_a}{\partial\boldsymbol{\xi}_i\partial\boldsymbol{\xi}_j},\left(\boldsymbol{\eta}_\beta - \frac{1}{N}\sum_{i=1}\boldsymbol{t}_\beta(\boldsymbol{y}_i)\right)\frac{\partial h_L^\beta}{\partial\boldsymbol{\theta}^b}\frac{\partial^2\boldsymbol{\theta}_b}{\partial\boldsymbol{\xi}_k\partial\boldsymbol{\xi}_l}\right).$$

It follows that the first covariance term is exactly the coordinate transform of the original variance:

$$\mathrm{Cov}\left(\frac{\partial\boldsymbol{\theta}_a}{\partial\boldsymbol{\xi}_i}[\hat{\mathcal{I}}_2(\boldsymbol{\theta})]^{ab}\frac{\partial\boldsymbol{\theta}_b}{\partial\boldsymbol{\xi}_j},\frac{\partial\boldsymbol{\theta}_c}{\partial\boldsymbol{\xi}_k}[\hat{\mathcal{I}}_2(\boldsymbol{\theta})]^{cd}\frac{\partial\boldsymbol{\theta}_d}{\partial\boldsymbol{\xi}_l}\right) = \frac{\partial\boldsymbol{\theta}_a}{\partial\boldsymbol{\xi}_i}\frac{\partial\boldsymbol{\theta}_b}{\partial\boldsymbol{\xi}_j}\frac{\partial\boldsymbol{\theta}_c}{\partial\boldsymbol{\xi}_k}\frac{\partial\boldsymbol{\theta}_d}{\partial\boldsymbol{\xi}_l}\mathrm{Cov}\left([\hat{\mathcal{I}}_2(\boldsymbol{\theta})]^{ab},[\hat{\mathcal{I}}_2(\boldsymbol{\theta})]^{cd}\right).$$

For the final covariance term we have:

$$\mathrm{Cov}\left(\left(\boldsymbol{\eta}_\alpha - \frac{1}{N}\sum_{i=1}\boldsymbol{t}_\alpha(\boldsymbol{y}_i)\right)\frac{\partial h_L^\alpha}{\partial\boldsymbol{\theta}^a}\frac{\partial^2\boldsymbol{\theta}_a}{\partial\boldsymbol{\xi}_i\partial\boldsymbol{\xi}_j},\left(\boldsymbol{\eta}_\beta - \frac{1}{N}\sum_{i=1}\boldsymbol{t}_\beta(\boldsymbol{y}_i)\right)\frac{\partial h_L^\beta}{\partial\boldsymbol{\theta}^b}\frac{\partial^2\boldsymbol{\theta}_b}{\partial\boldsymbol{\xi}_k\partial\boldsymbol{\xi}_l}\right)$$

$$= \frac{\partial^2\boldsymbol{\theta}_a}{\partial\boldsymbol{\xi}_i\partial\boldsymbol{\xi}_j}\frac{\partial^2\boldsymbol{\theta}_b}{\partial\boldsymbol{\xi}_k\partial\boldsymbol{\xi}_l}\mathrm{Cov}\left(\left(\boldsymbol{\eta}_\alpha - \frac{1}{N}\sum_{i=1}\boldsymbol{t}_\alpha(\boldsymbol{y}_i)\right)\frac{\partial h_L^\alpha}{\partial\boldsymbol{\theta}^a},\left(\boldsymbol{\eta}_\beta - \frac{1}{N}\sum_{i=1}\boldsymbol{t}_\beta(\boldsymbol{y}_i)\right)\frac{\partial h_L^\beta}{\partial\boldsymbol{\theta}^b}\right)$$

$$= \frac{\partial^2\boldsymbol{\theta}_a}{\partial\boldsymbol{\xi}_i\partial\boldsymbol{\xi}_j}\frac{\partial^2\boldsymbol{\theta}_b}{\partial\boldsymbol{\xi}_k\partial\boldsymbol{\xi}_l}\frac{\partial h_L^\alpha}{\partial\boldsymbol{\theta}^a}\frac{\partial h_L^\beta}{\partial\boldsymbol{\theta}^b}\mathrm{Cov}\left(\left(\boldsymbol{\eta}_\alpha - \frac{1}{N}\sum_{i=1}\boldsymbol{t}_\alpha(\boldsymbol{y}_i)\right),\left(\boldsymbol{\eta}_\beta - \frac{1}{N}\sum_{i=1}\boldsymbol{t}_\beta(\boldsymbol{y}_i)\right)\right)$$

$$= \frac{\partial^2\boldsymbol{\theta}_a}{\partial\boldsymbol{\xi}_i\partial\boldsymbol{\xi}_j}\frac{\partial^2\boldsymbol{\theta}_b}{\partial\boldsymbol{\xi}_k\partial\boldsymbol{\xi}_l}\frac{\partial h_L^\alpha}{\partial\boldsymbol{\theta}^a}\frac{\partial h_L^\beta}{\partial\boldsymbol{\theta}^b}\frac{1}{N}\sum_{i=1}\mathrm{Cov}\left((\boldsymbol{\eta}_\alpha - \boldsymbol{t}_\alpha(\boldsymbol{y}_i)),(\boldsymbol{\eta}_\beta - \boldsymbol{t}_\beta(\boldsymbol{y}_i))\right)$$

$$= \frac{\partial^2\boldsymbol{\theta}_a}{\partial\boldsymbol{\xi}_i\partial\boldsymbol{\xi}_j}\frac{\partial^2\boldsymbol{\theta}_b}{\partial\boldsymbol{\xi}_k\partial\boldsymbol{\xi}_l}\frac{\partial h_L^\alpha}{\partial\boldsymbol{\theta}^a}\frac{\partial h_L^\beta}{\partial\boldsymbol{\theta}^b}\frac{1}{N}\mathcal{I}_{\alpha\beta}(\boldsymbol{h}_L),$$

where the second last line comes from the independence of samples.

Thus all there is left is to calculate the middle terms. Without loss of generality, we calculate:

$$\text{Cov}\left(\frac{\partial\boldsymbol{\theta}_a}{\partial\boldsymbol{\xi}_i}[\hat{\mathcal{I}}_2(\boldsymbol{\theta})]^{ab}\frac{\partial\boldsymbol{\theta}_b}{\partial\boldsymbol{\xi}_j}, \left(\boldsymbol{\eta}_\beta - \frac{1}{N}\sum_{i=1}^{N}\boldsymbol{t}_\beta(\boldsymbol{y}_i)\right)\frac{\partial\boldsymbol{h}_L^\beta}{\partial\boldsymbol{\theta}^b}\frac{\partial^2\boldsymbol{\theta}_b}{\partial\boldsymbol{\xi}_k\partial\boldsymbol{\xi}_l}\right)$$

$$= \frac{\partial\boldsymbol{\theta}_a}{\partial\boldsymbol{\xi}_i}\frac{\partial\boldsymbol{\theta}_b}{\partial\boldsymbol{\xi}_j}\frac{\partial^2\boldsymbol{\theta}_c}{\partial\boldsymbol{\xi}_k\partial\boldsymbol{\xi}_l}\text{Cov}\left([\hat{\mathcal{I}}_2(\boldsymbol{\theta})]^{ab}, \left(\boldsymbol{\eta}_\beta - \frac{1}{N}\sum_{i=1}^{N}\boldsymbol{t}_\beta(\boldsymbol{y}_i)\right)\frac{\partial\boldsymbol{h}_L^\beta}{\partial\boldsymbol{\theta}^c}\right)$$

$$= \frac{\partial\boldsymbol{\theta}_a}{\partial\boldsymbol{\xi}_i}\frac{\partial\boldsymbol{\theta}_b}{\partial\boldsymbol{\xi}_j}\frac{\partial^2\boldsymbol{\theta}_c}{\partial\boldsymbol{\xi}_k\partial\boldsymbol{\xi}_l}$$
$$\text{Cov}\left(\mathcal{I}_{ab}(\boldsymbol{\theta}) + \left(\boldsymbol{\eta}_\beta - \frac{1}{N}\sum_{i=1}^{N}\boldsymbol{t}_\beta(\boldsymbol{y}_i)\right)\frac{\partial^2\boldsymbol{h}_L^\alpha}{\partial\boldsymbol{\theta}^a\boldsymbol{\theta}^b}, \left(\boldsymbol{\eta}_\beta - \frac{1}{N}\sum_{i=1}^{N}\boldsymbol{t}_\beta(\boldsymbol{y}_i)\right)\frac{\partial\boldsymbol{h}_L^\beta}{\partial\boldsymbol{\theta}^c}\right)$$

$$= \frac{\partial\boldsymbol{\theta}_a}{\partial\boldsymbol{\xi}_i}\frac{\partial\boldsymbol{\theta}_b}{\partial\boldsymbol{\xi}_j}\frac{\partial^2\boldsymbol{\theta}_c}{\partial\boldsymbol{\xi}_k\partial\boldsymbol{\xi}_l}\frac{\partial^2\boldsymbol{h}_L^\alpha}{\partial\boldsymbol{\theta}^a\boldsymbol{\theta}^b}\frac{\partial\boldsymbol{h}_L^\beta}{\partial\boldsymbol{\theta}^c}\text{Cov}\left(\left(\boldsymbol{\eta}_\beta - \frac{1}{N}\sum_{i=1}^{N}\boldsymbol{t}_\beta(\boldsymbol{y}_i)\right), \left(\boldsymbol{\eta}_\beta - \frac{1}{N}\sum_{i=1}^{N}\boldsymbol{t}_\beta(\boldsymbol{y}_i)\right)\right)$$

$$= \frac{\partial\boldsymbol{\theta}_a}{\partial\boldsymbol{\xi}_i}\frac{\partial\boldsymbol{\theta}_b}{\partial\boldsymbol{\xi}_j}\frac{\partial^2\boldsymbol{\theta}_c}{\partial\boldsymbol{\xi}_k\partial\boldsymbol{\xi}_l}\frac{\partial^2\boldsymbol{h}_L^\alpha}{\partial\boldsymbol{\theta}^a\boldsymbol{\theta}^b}\frac{\partial\boldsymbol{h}_L^\beta}{\partial\boldsymbol{\theta}^c}\frac{1}{N}\mathcal{I}_{\alpha\beta}(\boldsymbol{h}_L).$$

Combining these 3 covariance results gives us the covariance presented in the theorem. $\square$

As per the estimators themselves in Theorem A.2, the 4D covariances obey similar rules under coordinate transformations. Equation (A.7) has an additional term $\frac{1}{N}\mathcal{C}^{\alpha\beta}\mathcal{I}_{\alpha\beta}(\boldsymbol{h}_L)$ which depends on the Hessian of the coordinate transformation. Notice that each of the coordinate transformed covariance values depend on the same central moments of the exponential family – even the second estimator with the additional term only depends on the covariance/FIM w.r.t. $\boldsymbol{h}_L$. Each of these covariances include a weighted sums of the original covariance matrices in Theorems 4 and 6. As such, our initial element-wise considerations of the covariance will still be useful in the computation of the covariance in the new coordinates. In-fact, our upper bounds in Lemmas A.4 and A.5 and Theorem A.6 can be simply adapted by adding the appropriate norms of the coordinate transformation (and its Hessian component for the second estimator).

## D  Element-wise Covariance Bounds

The covariance tensor $\left[\text{Cov}\left(\hat{\mathcal{I}}_1(\boldsymbol{\theta})\right)\right]^{ijkl}$ has the following element-wise bound.

**Lemma A.4.**

$$\left|\left[\text{Cov}\left(\hat{\mathcal{I}}_1(\boldsymbol{\theta})\right)\right]^{ijkl}\right| \leq \frac{1}{N}\cdot\|\partial_i\boldsymbol{h}_L(\boldsymbol{x})\|_2\cdot\|\partial_j\boldsymbol{h}_L(\boldsymbol{x})\|_2\cdot\|\partial_k\boldsymbol{h}_L(\boldsymbol{x})\|_2\cdot\|\partial_l\boldsymbol{h}_L(\boldsymbol{x})\|_2$$
$$\cdot\|\mathcal{K}(\boldsymbol{t}) - \mathcal{I}(\boldsymbol{h}_L)\otimes\mathcal{I}(\boldsymbol{h}_L)\|_F,$$

*where $\|\cdot\|_F$ is the Frobenius norm of a tensor (square root of the sum of the squares of the elements / the $L_2$-norm) and $\otimes$ is the tensor-product: $(\mathcal{I}(\boldsymbol{h}_L)\otimes\mathcal{I}(\boldsymbol{h}_L))_{abcd} := \mathcal{I}_{ab}(\boldsymbol{h}_L)\cdot\mathcal{I}_{cd}(\boldsymbol{h}_L)$.*

*Proof.* Corollary holds immediately from the use of Hölder's inequality / Cauchy-Schwarz. Let $1 \leq p,q \leq \infty$ such that $1/p + 1/q = 1$. Let $\mathcal{T}_{abcd} = \mathcal{K}_{abcd}(\boldsymbol{t}) - \mathcal{I}_{ab}(\boldsymbol{h}_L)\cdot\mathcal{I}_{cd}(\boldsymbol{h}_L)$. From Theorem 4 we have:

$$\left|\left[\text{Cov}\left(\hat{\mathcal{I}}_1(\boldsymbol{\theta})\right)\right]^{ijkl}\right|$$
$$= \frac{1}{N}\cdot\left|\partial_i\boldsymbol{h}_L^a(\boldsymbol{x})\partial_j\boldsymbol{h}_L^b(\boldsymbol{x})\partial_k\boldsymbol{h}_L^c(\boldsymbol{x})\partial_l\boldsymbol{h}_L^d(\boldsymbol{x})\cdot\mathcal{T}_{abcd}\right|$$
$$\leq \frac{1}{N}\cdot\|\partial_i\boldsymbol{h}_L(\boldsymbol{x})\otimes\partial_j\boldsymbol{h}_L(\boldsymbol{x})\otimes\partial_k\boldsymbol{h}_L(\boldsymbol{x})\otimes\partial_l\boldsymbol{h}_L(\boldsymbol{x})\|_p\cdot\|\mathcal{T}\|_q$$
$$= \frac{1}{N}\cdot\|\partial_i\boldsymbol{h}_L(\boldsymbol{x})\|_p\cdot\|\partial_j\boldsymbol{h}_L(\boldsymbol{x})\|_p\cdot\|\partial_k\boldsymbol{h}_L(\boldsymbol{x})\|_p\cdot\|\partial_l\boldsymbol{h}_L(\boldsymbol{x})\|_p\cdot\|\mathcal{T}\|_q.$$

Thus, by taking the $p = q = 2$ the Lemma holds. $\square$

We have similar element-wise and global upper bounds on the covariance of $\hat{\mathcal{I}}_2(\boldsymbol{\theta})$.

**Lemma A.5.**

$$\left| \left[ \mathrm{Cov}\left( \hat{\mathcal{I}}_2(\boldsymbol{\theta}) \right) \right]^{ijkl} \right| \leq \frac{1}{N} \cdot \|\partial^2_{ij}\boldsymbol{h}_L(\boldsymbol{x})\|_2 \cdot \|\partial^2_{kl}\boldsymbol{h}_L(\boldsymbol{x})\|_2 \cdot \|\mathcal{I}(\boldsymbol{h}_L)\|_F.$$

*Proof.* Corollary holds immediately from the use of Hölder's inequality / Cauchy-Schwarz. Let $1 \leq p, q \leq \infty$ such that $1/p + 1/q = 1$. From Theorem 6 we have:

$$
\begin{aligned}
\left| \left[ \mathrm{Cov}\left( \hat{\mathcal{I}}_2(\boldsymbol{\theta}) \right) \right]^{ijkl} \right| &= \frac{1}{N} \cdot \left| \partial^2_{ij}\boldsymbol{h}_L^\alpha(\boldsymbol{x})\partial^2_{kl}\boldsymbol{h}_L^\beta(\boldsymbol{x})\mathcal{I}_{\alpha\beta}(\boldsymbol{h}_L) \right| \\
&\leq \frac{1}{N} \cdot \|\partial^2_{ij}\boldsymbol{h}_L(\boldsymbol{x}) \otimes \partial^2_{kl}\boldsymbol{h}_L(\boldsymbol{x})\| \cdot \|\mathcal{I}(\boldsymbol{h}_L)\| \\
&= \frac{1}{N} \cdot \|\partial^2_{ij}\boldsymbol{h}_L(\boldsymbol{x})\| \cdot \|\partial^2_{kl}\boldsymbol{h}_L(\boldsymbol{x})\| \cdot \|\mathcal{I}(\boldsymbol{h}_L)\|.
\end{aligned}
$$

Thus, by taking the $p = q = 2$ the Lemma holds. $\qquad\square$

# E  Alternative Norm Results

An alternative bound to Theorem 7 and Theorem 8 can be established by utilizing Hölder's inequality for $L_p$-norms.

**Theorem A.6.**

$$\left\| \mathrm{Cov}\left( \hat{\mathcal{I}}_1(\boldsymbol{\theta}) \right) \right\|_\infty \leq \frac{1}{N} \cdot \|\partial\boldsymbol{h}_L(\boldsymbol{x})\|_\infty^4 \cdot \|\mathcal{K}(\boldsymbol{t}) - \mathcal{I}(\boldsymbol{h}_L) \otimes \mathcal{I}(\boldsymbol{h}_L)\|_1 \tag{A.8}$$

$$\left\| \mathrm{Cov}\left( \hat{\mathcal{I}}_2(\boldsymbol{\theta}) \right) \right\|_\infty \leq \frac{1}{N} \cdot \|\partial^2\boldsymbol{h}_L(\boldsymbol{x})\|_\infty^2 \cdot \|\mathcal{I}(\boldsymbol{h}_L)\|_1, \tag{A.9}$$

where $\| \cdot \|_\infty$ is the $L_\infty$-norm and $\| \cdot \|_1$ is the $L_1$-norm.

*Proof.* The Corollary holds directly from the inequalities given in the proof of Theorem 7 and Theorem 8. Let $p = \infty$ and $q = 1$.

Thus we have the inequalities for $L_p$-norms:

$$\left\| \mathrm{Cov}\left( \hat{\mathcal{I}}_1(\boldsymbol{\theta}) \right) \right\|_\infty \leq \frac{1}{N} \cdot \|\partial\boldsymbol{h}_L(\boldsymbol{x})\|_\infty^4 \cdot \|\mathcal{K}(\boldsymbol{t}) - \mathcal{I}(\boldsymbol{h}_L) \otimes \mathcal{I}(\boldsymbol{h}_L)\|_1$$

$$\left\| \mathrm{Cov}\left( \hat{\mathcal{I}}_2(\boldsymbol{\theta}) \right) \right\|_\infty \leq \frac{1}{N} \cdot \|\partial^2\boldsymbol{h}_L(\boldsymbol{x})\|_\infty^2 \cdot \|\mathcal{I}(\boldsymbol{h}_L)\|_1.$$

**Remark A.6.1.** *Note that these are exactly equivalent to certain spectral and nuclear norms. However these have slight differences in their definition [6].*

*The $p$-spectral norm for a $d$-dimensional tensor is given by:*

$$\|\mathcal{T}\|_{p_\sigma} = \max \left\{ \langle \mathcal{T}, \boldsymbol{x}^1 \otimes \ldots \otimes \boldsymbol{x}^d \rangle : \|\boldsymbol{x}^k\|_p = 1, \ \forall k \in [d] \right\}.$$

*The standard tensor spectral norm is only equivalent when $p = 2$, i.e., $\| \cdot \|_{2_\sigma}$.*

*The $p$-nuclear norm for a $d$-dimensional tensor is given by:*

$$\|\mathcal{T}\|_{p_*} = \min \left\{ \sum_{i=1}^r |\lambda_i| : \mathcal{T} = \sum_{i=1}^r \lambda_i \boldsymbol{x}_i^1 \otimes \ldots \otimes \boldsymbol{x}_i^d : \|\boldsymbol{x}_i^k\|_p = 1, \ \forall k \in [d] \ and \ i, r \in \mathbb{N} \right\}.$$

*The standard tensor nuclear norm is only equivalent when $p = 2$, i.e., $\| \cdot \|_{2_*}$.*

*It follows that for the $L_p$ norms we have established, the $L_1$-norm is equivalent to the 1-nuclear norm and the $L_\infty$-norm is equal to the 1-spectral norm [6, Proposition 2.6]. However, this equality is not true for the standard tensor nuclear and spectral norms.*

*Instead, we can upper bound the $L_\infty$-norm by the standard tensor spectral norm by the definition of the $p$-spectral norm, i.e., $\| \cdot \|_\infty = \| \cdot \|_{1_\sigma} \leq \| \cdot \|_{2_\sigma}$. For the $L_1$-norm, we can upper bound it by the corresponding $L_2$-norm through Cauchy-Schwarz, i.e., $\| \cdot \|_1 \leq \| \cdot \|_2 \cdot \sqrt{\mathcal{D}}$, where $\mathcal{D}(\cdot)$ is the product of the dimension size of the tensor. One should note that $L_2$ is the Frobenius norm.*

$\square$

Notably, the $L_1$-norm can be upper-bounded by the Frobenius norm trivially through the Cauchy-Schwarz inequality, with $\| \cdot \|_1 \le \| \cdot \|_F \cdot \sqrt{\mathcal{D}}$, where $\mathcal{D}$ is the product of the dimension size of the tensor. Thus the analysis in Lemma 9 can be useful to extend the bounds in Theorem A.6. On the other hand, the $L_\infty$-norm $\| \cdot \|_\infty$ is upper bounded by the largest singular value of the tensor [17]. As such, the $\| \cdot \|_\infty$ quantities on the RHS can be interpreted as functions of the maximum singular value of the Jacobian $\partial \boldsymbol{h}_L(\boldsymbol{x})$ or the Hessian $\partial^2 \boldsymbol{h}_L(\boldsymbol{x})$. Similar corollaries for Lemma A.4 and Lemma A.5 can be established — which we omit for brevity.

## F   Combination of estimators

We consider a convex combination of estimators, i.e., Eqs. (7) and (8).

In particular, for $0 \le \alpha \le 1$ we have:

$$\hat{\mathcal{I}}_\alpha(\boldsymbol{\theta}) = \alpha \hat{\mathcal{I}}_1(\boldsymbol{\theta}) + (1-\alpha)\hat{\mathcal{I}}_2(\boldsymbol{\theta}). \tag{A.10}$$

Clearly, this is also a point-wise estimator. Thus, Proposition 1 holds for this estimator.

For the variance, we use the following linear relation:

$$\text{Var}(\alpha \hat{\mathcal{I}}_1(\boldsymbol{\theta}) + (1-\alpha)\hat{\mathcal{I}}_2(\boldsymbol{\theta})) = \alpha^2 \text{Var}(\hat{\mathcal{I}}_1(\boldsymbol{\theta})) + (1-\alpha)^2 \text{Var}(\hat{\mathcal{I}}_2(\boldsymbol{\theta})) + 2\alpha(1-\alpha)\text{Cov}(\hat{\mathcal{I}}_1(\boldsymbol{\theta}), \hat{\mathcal{I}}_2(\boldsymbol{\theta})).$$

Or as we have previously discussed we consider:

$$\alpha^2 \text{Var}\left( \frac{\partial \ell}{\partial \boldsymbol{\theta}_i} \frac{\partial \ell}{\partial \boldsymbol{\theta}_j} \right) + (1-\alpha)^2 \text{Var}\left( -\frac{\partial^2 \ell}{\partial \boldsymbol{\theta}_i \partial \boldsymbol{\theta}_j} \right) + 2\alpha(1-\alpha)\text{Cov}\left( \frac{\partial \ell}{\partial \boldsymbol{\theta}_i} \frac{\partial \ell}{\partial \boldsymbol{\theta}_j}, -\frac{\partial^2 \ell}{\partial \boldsymbol{\theta}_i \partial \boldsymbol{\theta}_j} \right)$$

We already have the variance values from Theorems 4 and 6. Thus all we have to calculate is the covariance term.

$$\text{Cov}\left( \partial_i \ell \cdot \partial_j \ell, -\partial^2_{ij}\ell \right) = -\mathbb{E}[\partial_i \ell \cdot \partial_j \ell \cdot \partial^2_{ij}\ell] + \mathbb{E}[\partial_i \ell \cdot \partial_j \ell]\mathbb{E}[\partial^2_{ij}\ell].$$

The first term of the covariance can be given by:

$$-\mathbb{E}[\partial_i \ell \cdot \partial_j \ell \cdot \partial^2_{ij}\ell]$$
$$= E\left[ \partial_i \ell \cdot \partial_j \ell \cdot \left( \partial^c_i \boldsymbol{h}_L \partial^d_j \boldsymbol{h}_L \mathcal{I}_{cd}(\boldsymbol{h}_L) - \delta_c \partial^c_{ij}\boldsymbol{h}_L \right) \right]$$
$$= E\left[ \partial_i \ell \cdot \partial_j \ell \cdot \partial^c_i \boldsymbol{h}_L \partial^d_j \boldsymbol{h}_L \mathcal{I}_{cd}(\boldsymbol{h}_L) \right] - E\left[ \partial_i \ell \cdot \partial_j \ell \cdot \delta_c \partial^c_{ij}\boldsymbol{h}_L \right]$$
$$= \partial^c_i \boldsymbol{h}_L \partial^d_j \boldsymbol{h}_L \mathcal{I}_{cd}(\boldsymbol{h}_L) \cdot E\left[ \partial_i \ell \cdot \partial_j \ell \right] - \partial^c_{ij}\boldsymbol{h}_L \cdot E\left[ \partial_i \ell \cdot \partial_j \ell \cdot \delta_c \right]$$
$$= \partial^c_i \boldsymbol{h}_L \partial^d_j \boldsymbol{h}_L \mathcal{I}_{cd}(\boldsymbol{h}_L) \cdot E\left[ \partial_i \ell \cdot \partial_j \ell \right] - \partial^c_{ij}\boldsymbol{h}_L \cdot E\left[ \partial_i \ell \cdot \partial_j \ell \cdot \delta_c \right]$$
$$= \partial_i \boldsymbol{h}^a_L \cdot \partial_j \boldsymbol{h}^b_L \cdot \partial^c_i \boldsymbol{h}_L \cdot \partial^d_j \boldsymbol{h}_L \cdot \mathcal{I}_{cd}(\boldsymbol{h}_L) \cdot E\left[ \delta_a \cdot \delta_b \right] - \partial_i \boldsymbol{h}^a_L(\boldsymbol{x}) \cdot \partial_j \boldsymbol{h}^b_L(\boldsymbol{x}) \cdot \partial^c_{ij}\boldsymbol{h}_L \cdot E\left[ \delta_a \cdot \delta_b \cdot \delta_c \right]$$
$$= \partial_i \boldsymbol{h}^a_L \cdot \partial_j \boldsymbol{h}^b_L \cdot \partial^c_i \boldsymbol{h}_L \cdot \partial^d_j \boldsymbol{h}_L \cdot \mathcal{I}_{cd}(\boldsymbol{h}_L) \cdot \mathcal{I}_{ab}(\boldsymbol{h}_L) - \partial_i \boldsymbol{h}^a_L(\boldsymbol{x}) \cdot \partial_j \boldsymbol{h}^b_L(\boldsymbol{x}) \cdot \partial^c_{ij}\boldsymbol{h}_L \cdot E\left[ \delta_a \cdot \delta_b \cdot \delta_c \right]$$

The second term is given by:

$$\mathbb{E}[\partial_i \ell \cdot \partial_j \ell]\mathbb{E}[\partial^2_{ij}\ell] = -\partial_i \boldsymbol{h}^a_L \cdot \partial_j \boldsymbol{h}^b_L \cdot \partial^c_i \boldsymbol{h}_L \cdot \partial^d_j \boldsymbol{h}_L \cdot \mathcal{I}_{cd}(\boldsymbol{h}_L) \cdot \mathcal{I}_{ab}(\boldsymbol{h}_L)$$

Thus, together we have the covariance:

$$\text{Cov}\left( \partial_i \ell \cdot \partial_j \ell, -\partial^2_{ij}\ell \right) = -\partial_i \boldsymbol{h}^a_L \cdot \partial_j \boldsymbol{h}^b_L \cdot \partial^c_{ij}\boldsymbol{h}_L \cdot E\left[ \delta_a \cdot \delta_b \cdot \delta_c \right].$$

This gives us the variance of the combined estimator:

$$\alpha^2 \partial_i \boldsymbol{h}^a_L \partial_j \boldsymbol{h}^b_L \partial_i \boldsymbol{h}^c_L \partial_j \boldsymbol{h}^d_L \cdot \left( \mathcal{K}_{abcd}(\boldsymbol{t}) - \mathcal{I}_{ab}(\boldsymbol{h}_L) \cdot \mathcal{I}_{cd}(\boldsymbol{h}_L) \right) + (1-\alpha)^2 \partial^2_{ij}\boldsymbol{h}^\alpha_L \partial^2_{ij}\boldsymbol{h}^\beta_L \cdot \mathcal{I}_{\alpha\beta}(\boldsymbol{h}_L)$$
$$- 2\alpha(1-\alpha)\partial_i \boldsymbol{h}^a_L \partial_j \boldsymbol{h}^b_L \partial^c_{ij}\boldsymbol{h}_L \cdot E\left[ \delta_a \cdot \delta_b \cdot \delta_c \right]. \tag{A.11}$$

The covariance (as per Theorems 4 and 6) can similarly be calculated by changing the variables of the partial derivatives of $\boldsymbol{h}_L$. Notably, the largest differentiating factor from the original estimator is that the combined estimator is dependent on the third central moment of $\boldsymbol{t}$. This third central moment is equivalent to the third-order cumulant of $\boldsymbol{t}$. Thus it can be directly calculated via the derivatives of the log-partition function $F(\boldsymbol{h}_L)$.

## G    Experimental Verification of Bounds

The bounds of the variance of the FIM estimators $\hat{\mathcal{I}}_1(\boldsymbol{\theta})$ and $\hat{\mathcal{I}}_2(\boldsymbol{\theta})$ (as presented in Section 3) can be experimentally verified. We train a simple convolutional neural network trained on the standard MNIST dataset. By leveraging the Jacobian and Hessian in-built functions in PyTorch, we calculate the bounds in Theorems 7 and 8 applied to the variance of the estimators (as described in Section 3.1).

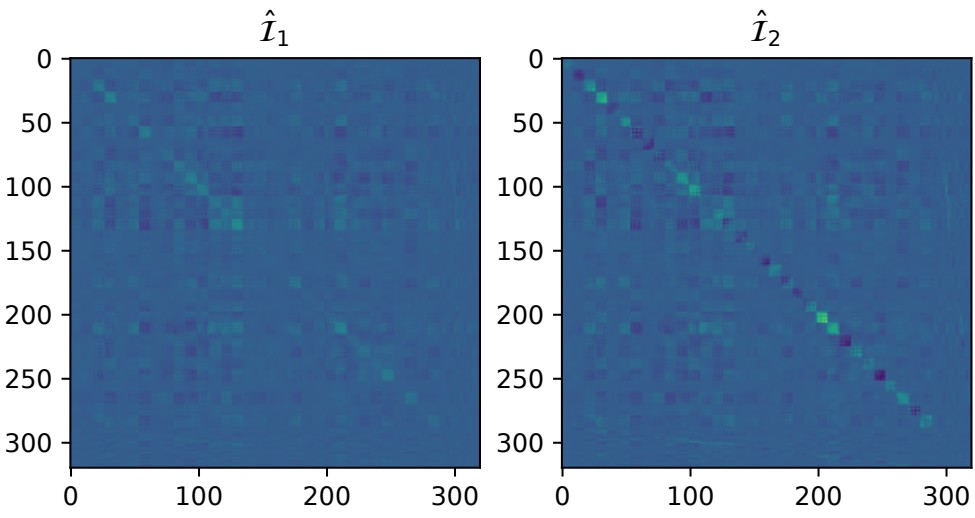

(a) $\hat{\mathcal{I}}_1(\boldsymbol{\theta})$ and $\hat{\mathcal{I}}_2(\boldsymbol{\theta})$, where $\boldsymbol{\theta}$ is a random model. Color values are shared.

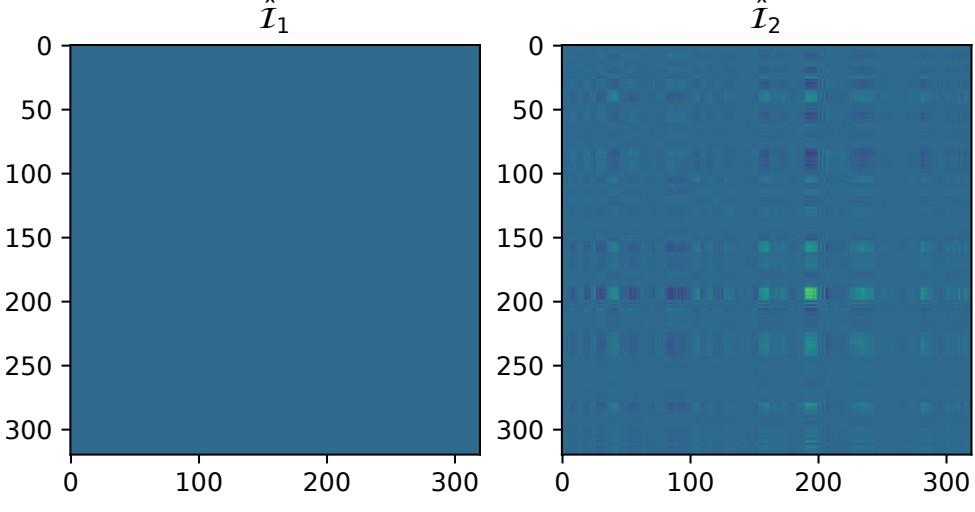

(b) $\hat{\mathcal{I}}_1(\boldsymbol{\theta})$ and $\hat{\mathcal{I}}_2(\boldsymbol{\theta})$, where $\boldsymbol{\theta}$ is a trained model. Color values are shared.

Figure A.1: The estimated FIM presented in heatmaps corresponding to the first layer of a CNN .

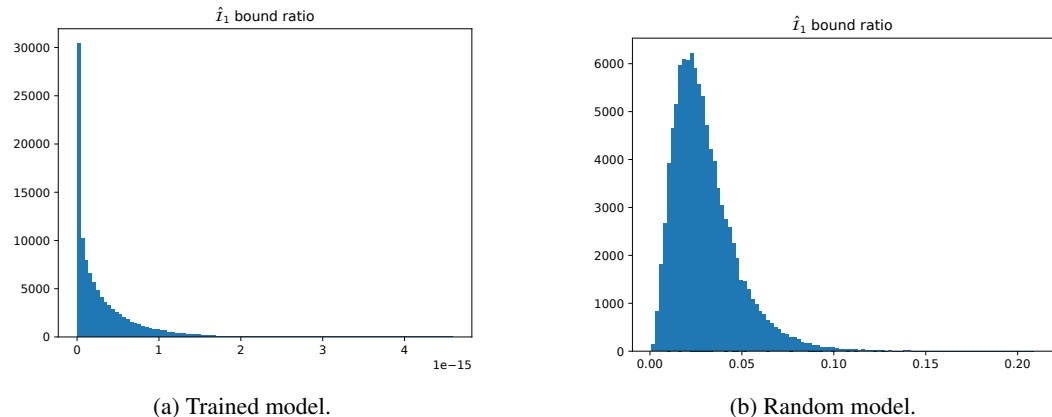

(a) Trained model.

(b) Random model.

Figure A.2: Ratio of the bound of the variance in Theorem 7 over the true estimated variance.

## G.1 Dataset

We consider the MNIST dataset of $28 \times 28$ pixel grayscale images after normalization. The training set consists of 60,000 examples and the test set consists of 10,000 examples.

## G.2 Model Setup

The convolutional neural network considered in our experiments are given by the following layers (in order):

- $\rightarrow$ Conv(in_channel=1, out_channel=32, kernel_size=(3, 3), stride=(1, 1))
- $\rightarrow$ Softplus()
- $\rightarrow$ Conv(in_channel=32, out_channel=64, kernel_size=(3, 3), stride=(1, 1))
- $\rightarrow$ Softplus()
- $\rightarrow$ MaxPool2D()
- $\rightarrow$ Dropout(p=0.25)
- $\rightarrow$ Linear(in_features=9216, out_features=128)
- $\rightarrow$ Softplus()
- $\rightarrow$ Dropout(p=0.5)
- $\rightarrow$ Linear(in_features=128, out_features=10)
- $\rightarrow$ LogSoftMax()

After training, the final model has a 99% test accuracy. For most of the training samples, the predicted probabilities have a low entropy and are close to a one-hot vector. Consequently, the overall variance of the estimated FIM is very close to zero.

## G.3 Evaluation

To compute the FIM, We only consider the weight and bias parameters in the first layer for simplicity. We randomly choose a fixed $x$ with multiple sampled $y_i$ for calculation, as per Eqs. (7) and (8). Recall that the randomness of our estimators comes from the sampling of $y_i \sim p(y \mid x)$. For all related computation, we use double-precision floating point (64 bits). The FIM estimations for both a trained model and a random model are given in Figs. A.1a and A.1b.

To calculate the "true" variance to compare against the bounds, we use Monte-Carlo estimation using a large number (1,000 for the presented results) of samples. Then, the variance of the estimator is approximated by the sample variance.

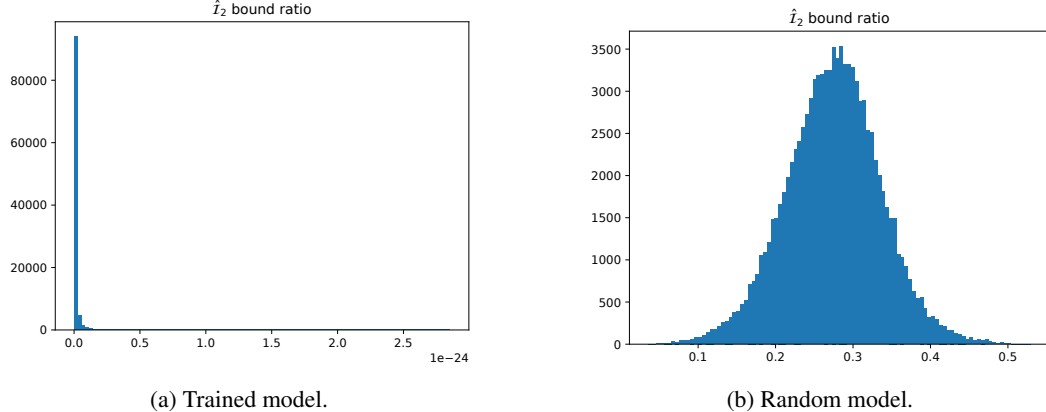

(a) Trained model.  (b) Random model.

Figure A.3: Ratio of the bound of the variance in Theorem 8 over the true estimated variance.

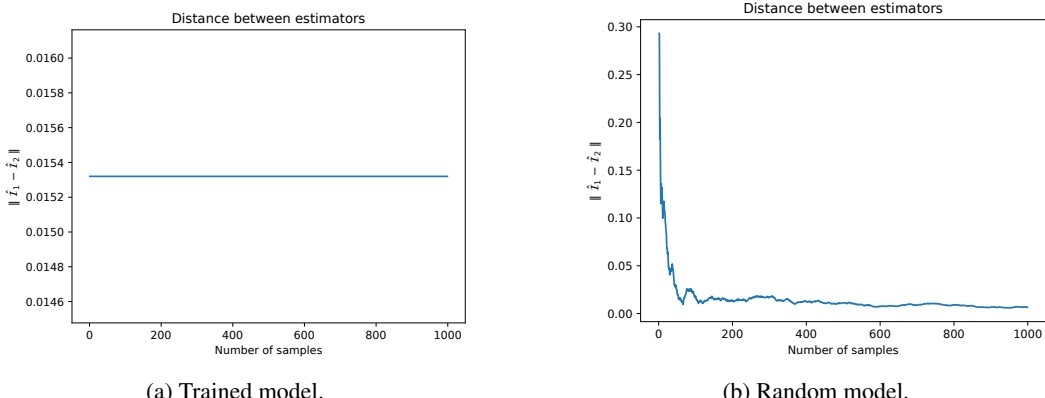

(a) Trained model.  (b) Random model.

Figure A.4: Distance between estimator $\hat{\mathcal{I}}_1(\boldsymbol{\theta})$ and $\hat{\mathcal{I}}_2(\boldsymbol{\theta})$.

### G.4 Results

We plot the histograms of the ratio of the estimated variance of $\hat{\mathcal{I}}_1(\boldsymbol{\theta})$ ($\hat{\mathcal{I}}_2(\boldsymbol{\theta})$) over the variance bound given by Theorem 7 (Theorem 8). For the trained network, the plot is given by Fig. A.2a (Fig. A.3a); for the random network, the plot is by Fig. A.2b (Fig. A.3b). In both the trained and random models, the bounds are empirically verified (the ratio is always smaller than 1). When comparing the ratio histograms, a smaller ratio value corresponds to a looser bound. We can immediately see that the trained network's bounds are looser than that of the randomized network.

We also plot of the (Frobenius) distance $\|\hat{\mathcal{I}}_1(\boldsymbol{\theta}) - \hat{\mathcal{I}}_2(\boldsymbol{\theta})\|_F$ between the two estimators over the number of samples for calculating the estimators. See Figs. A.4a and A.4b for the cases of trained and random networks, respectively. As the trained model has a very small variance of $\boldsymbol{y}_i$, it is hard to observe in Fig. A.4a any change of the distance between $\hat{\mathcal{I}}_1(\boldsymbol{\theta})$ and $\hat{\mathcal{I}}_2(\boldsymbol{\theta})$ as the samples increase. For the randomized model, we do observe the decrease in estimator distance as the number of samples increase, as expected, Fig. A.4b.

## H  Univariant Gaussian

We consider the case where we parameterize a univariant Gaussian distribution and consider the FIM and the corresponding estimators quantities.

Firstly, we specify Eq. (3) for a univariant Gaussian by setting:

$$\boldsymbol{t}(y) = (y, y^2); \quad F(\boldsymbol{h}) = -\frac{\boldsymbol{h}_1^2}{4\boldsymbol{h}_2} + \frac{1}{2}\ln\left(\frac{-\pi}{\boldsymbol{h}_2}\right),$$

where $\boldsymbol{y} = y \in \Re$ and $\boldsymbol{h} = \boldsymbol{h}_L$ for readability.

In particular, the 2 dimensional output neural network parametrizes the mean $\mu$ and standard deviation s by:

$$\boldsymbol{h} = \left( \frac{\mu}{\mathsf{s}^2}, -\frac{1}{2\mathsf{s}^2} \right).$$

Furthermore, we have dual coordinates:

$$\boldsymbol{\eta} = \left( -\frac{\boldsymbol{h}_1}{2\boldsymbol{h}_2}, \frac{\boldsymbol{h}_1^2 - 2\boldsymbol{h}_2}{4\boldsymbol{h}_2^2} \right) = (\mu, \mu^2 + \mathsf{s}^2).$$

The closed form for the FIM/covariance matrix is given by:

$$\mathcal{I}(\boldsymbol{h}) = \text{Var}(\boldsymbol{t}) = \begin{bmatrix} -\frac{1}{2\boldsymbol{h}_2} & \frac{\boldsymbol{h}_1}{2\boldsymbol{h}_2^2} \\ \frac{\boldsymbol{h}_1}{2\boldsymbol{h}_2^2} & -\frac{\boldsymbol{h}_1^2}{2\boldsymbol{h}_2^3} + \frac{1}{2\boldsymbol{h}_2^2} \end{bmatrix}.$$

Notably, we have that $\boldsymbol{h}_1 \in \Re$ and $\boldsymbol{h}_2 \in (-\infty, 0)$.

We present the following element-wise plots of FIM in Fig. A.5.

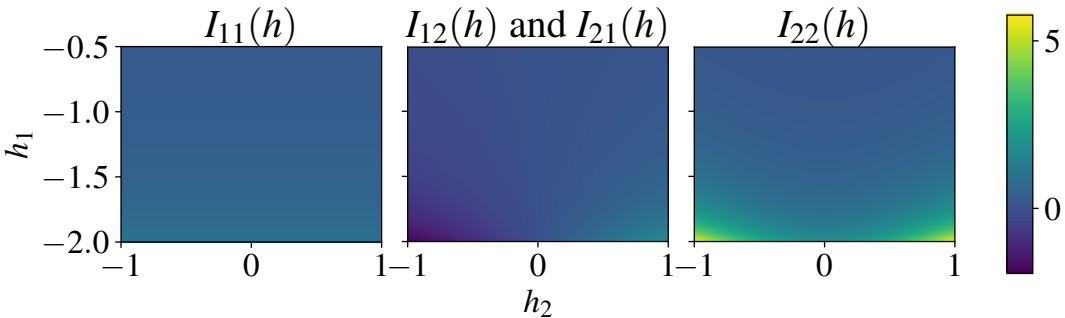

Figure A.5: Normal distribution $\mathcal{I}(\boldsymbol{h})$.

## I   Residual simple cases

We look into the residual term which is present in the proof of Theorem 3. Specifically, the quantity:

$$\mathcal{R}(\boldsymbol{\theta}) = \int p(\boldsymbol{x}) \frac{\partial^2}{\partial\boldsymbol{\theta}\partial\boldsymbol{\theta}^\top} p(\boldsymbol{y} \mid \boldsymbol{x}) \, \mathrm{d}\boldsymbol{y} \, \mathrm{d}\boldsymbol{x}.$$

Consider the simple case where we only have a single weight and neuron,

$$p(y \mid x) = \exp\left( t(y)h - F(h) \right),$$
$$h = \sigma(wx).$$

First consider the first derivative:

$$\frac{\partial}{\partial w} p(y \mid x) = p(y \mid x) \cdot \frac{\partial}{\partial w} \left( t(y)h - F(h) \right)$$
$$= p(y \mid x) \cdot (t(y) - \eta(h)) \cdot \frac{\partial h}{\partial w}$$
$$= p(y \mid x) \cdot (t(y) - \eta(h)) \cdot \sigma'(wx) \cdot x$$

## I.1 Assume that activation is identity

As $\sigma'(z) = 1$, we have the second derivative is the following:

$$\frac{\partial^2}{\partial^2 w} p(y \mid x) = \frac{\partial}{\partial w} \left( p(y \mid x) \cdot (t(y) - \eta(h)) \cdot x \right)$$
$$= p(y \mid x) \left[ (t(y) - \eta(h))^2 \cdot x^2 - \triangledown \eta(h) \cdot x^2 \right]$$

Integrating the first term for the residual we have

$$\int \int p(x) p(y \mid x) \cdot (t(y) - \eta(h))^2 \cdot x^2 \, \mathrm{d}y \, \mathrm{d}x = \int p(x) \cdot \triangledown \eta(h) \cdot x^2 \, \mathrm{d}x.$$

Integrating the second term for the residual we have

$$\int \int p(x) p(y \mid x) \cdot \triangledown \eta(h) \cdot x^2 \, \mathrm{d}y \, \mathrm{d}x = \int p(x) \cdot \triangledown \eta(h) \cdot x^2 \, \mathrm{d}x.$$

Thus by taking the difference, the residual is zero.

## I.2 Assume that activation is not identity

The second derivative is the following:

$$\frac{\partial^2}{\partial^2 w} p(y \mid x) = \frac{\partial}{\partial w} \left( p(y \mid x) \cdot (t(y) - \eta(h)) \cdot \sigma'(wx) \cdot x \right)$$
$$= \frac{\partial}{\partial w} \left( p(y \mid x) \right) \cdot (t(y) - \eta(h)) \cdot \sigma'(wx) \cdot x$$
$$+ p(y \mid x) \cdot \frac{\partial}{\partial w} \left( t(y) - \eta(h) \right) \cdot \sigma'(wx) \cdot x$$
$$+ p(y \mid x) \cdot (t(y) - \eta(h)) \cdot \frac{\partial}{\partial w} \left( \sigma'(wx) \right) \cdot x$$
$$= p(y \mid x) \cdot (t(y) - \eta(h))^2 \cdot \sigma'(wx)^2 \cdot x^2$$
$$+ p(y \mid x) \cdot \frac{\partial}{\partial w} \left( t(y) - \eta(h) \right) \cdot \sigma'(wx) \cdot x$$
$$+ p(y \mid x) \cdot (t(y) - \eta(h)) \cdot \frac{\partial}{\partial w} \left( \sigma'(wx) \right) \cdot x$$
$$= p(y \mid x) \cdot (t(y) - \eta(h))^2 \cdot \sigma'(wx)^2 \cdot x^2$$
$$- p(y \mid x) \cdot \triangledown \eta(h) \cdot \sigma'(wx)^2 \cdot x^2$$
$$+ p(y \mid x) \cdot (t(y) - \eta(h)) \cdot \frac{\partial}{\partial w} \left( \sigma'(wx) \right) \cdot x.$$

By the linearity of the integral, we calculate each term of the residual. For the first term:

$$\int \int p(x) \cdot p(y \mid x) \cdot (t(y) - \eta(h))^2 \cdot \sigma'(wx)^2 \cdot x^2 \, \mathrm{d}y \, \mathrm{d}x$$
$$= \int p(x) \cdot \sigma'(wx)^2 \cdot x^2 \int p(y \mid x) \cdot (t(y) - \eta(h))^2 \, \mathrm{d}y \, \mathrm{d}x$$
$$= \int p(x) \cdot \sigma'(wx)^2 \cdot x^2 \cdot \triangledown \eta(h) \, \mathrm{d}x.$$

Given that the second term only has $p(y \mid x)$ which is dependent on $y$, the first and second term of the residual cancel out. Thus we only have:

$$
\begin{aligned}
\mathcal{R}(w) &= \int \int p(x)p(y \mid x) \cdot (t(y) - \eta(h)) \cdot \frac{\partial}{\partial w}\left(\sigma'(wx)\right) \cdot x \,\mathrm{d}y \,\mathrm{d}x \\
&= \int \int p(x)p(y \mid x) \cdot t(y) \cdot \frac{\partial}{\partial w}\left(\sigma'(wx)\right) \cdot x \,\mathrm{d}y \,\mathrm{d}x \\
&\quad - \int \int p(x)p(y \mid x) \cdot \eta(h) \cdot \frac{\partial}{\partial w}\left(\sigma'(wx)\right) \cdot x \,\mathrm{d}y \,\mathrm{d}x \\
&= \int p(x) \cdot \frac{\partial}{\partial w}\left(\sigma'(wx)\right) \cdot x \int p(y \mid x) \cdot t(y) \,\mathrm{d}y \,\mathrm{d}x - \int p(x) \cdot \eta(h) \cdot \frac{\partial}{\partial w}\left(\sigma'(wx)\right) \cdot x \,\mathrm{d}x.
\end{aligned}
$$

Given that

$$
\frac{\partial}{\partial w}\left(\sigma'(wx)\right) = \delta(wx)
$$

and

$$
\int f(x)\delta(x) \,\mathrm{d}x = f(0), \tag{A.12}
$$

the residual will result in zero for this case.