# OpenReview forum: "On the Variance of the Fisher Information for Deep Learning"
_NeurIPS.cc/2021/Conference — NeurIPS 2021 Poster_

### Official Review · Reviewer_Q3bA · 2021-07-15

**Rating:** 6
**Confidence:** 2

**Summary:**

This paper investigates the Fisher information matrix (FIM) theoretically. Two equivalent expressions of FIM and their estimators are discussed (rediscovered) for deep neural network with exponential family output units. The closed-form variance (and simpler-form upper bounds) for the two estimators is proposed; the convergence rate of the positiveness definiteness are also analyzed. Finally, the variance is connected with the structure of the corresponding deep neural networks.

**Ethics Review Area:**

["I don’t know"]

**Main Review:**

**Originality**

While the two estimators are not proposed originally, their variance has not been analyzed theoretically in machine learning, claimed by the authors. Besides, positive definiteness, convergence rates of the two estimators and the connection with the structure of the deep neural networks are also analyzed in details.

**Clarity**

This paper is clearly presented and readable. Although I am not familiar with this research area, I could keep up with the storyline without much effort.

**Significance**

The discussed estimators are very intuitive and make sense to me. In fact, I feel a little surprised about the "fact" that the two estimators have not been analyzed.

**Concerns**

In spite of some mentioned related work, more comparison of existing estimators with the two estimators should be proposed. It is confusing about the position in literature.

Moreover, since the two esitmators are quite simple, it is suggested to add some application cases (maybe toy cases) about them.

**Time Spent Reviewing:**

3

---

> ### Author Response · Authors · 2021-08-10
> **Reply to Reviewer Q3bA**
>
> Thank you for your reviewing efforts and positive remarks on our originality, clarity, and significance.
>
> ---
>
> **Position in literature**
>
> The observed Fisher information (Efron & Hinkley 1978) is defined using the second estimator $\hat{\mathcal{I}}_2(\mathbf{\theta})$. In the deep learning community, most of the time, the FIM or its inverse is based on an approximation of our first estimator $\hat{\mathcal{I}}_1(\mathbf{\theta})$, such as the empirical FIM, or the unit-wise FIM, or the K-FAC.  Ollivier (2015)'s Monte Carlo natural gradient is also based on $\hat{\mathcal{I}}_1(\mathbf{\theta})$.  The two estimators discussed in this paper differ from the FIM approximations, as they naturally arise from the definition of the FIM and are both consistent and unbiased, as stated in Proposition 1.  These estimators are not new, and we are not the first to realize their difference.  See e.g. the cited (Delattre & Kuhn 2019).  Nonetheless, to the best of our knowledge, their closed-form variance and bounds for deep neural networks were not carefully analyzed as in the current paper.
>
> ---
>
> **Some application cases**
>
> This paper focuses on the problem to estimate the FIM. Our results are mostly theoretical. From an application perspective, we will include in the supplementary material a separate section to showcase our FIM estimators. For details, please see our reply "Empirical Results" to all reviewers.
>
> ---
>
> References:
>
> Efron & Hinkley. Assessing the accuracy of the maximum likelihood estimator: Observed versus expected fisher information. 1978.
>
> Ollivier. Riemannian metrics for neural networks I: feedforward networks. 2015.
>
> Delattre & Kuhn. Estimating Fisher information matrix in latent variable models based on the score function. 2019.

---

### Official Review · Reviewer_muyd · 2021-07-16

**Rating:** 8
**Confidence:** 4

**Summary:**

For two commonly used approximations to the Fisher Information Matrix, this paper derives bounds on the variance of these estimates. They give examples of how these bounds change for different types of exponential family neural network output distributions.

**Limitations And Societal Impact:**

I agree with the authors that I do not believe there are relevant societal impacts of this work that are useful to discuss at this time.

**Main Review:**

After looking at the author responses, I am raising my score to an 8 to reflect that I already believed the paper was good enough for acceptance, but is now stronger.

----------------------------

Overall this is a well-done theory paper. I believe that these bounds will be useful references for practitioners deriving optimization preconditioners or other methods to compute local loss surface information via the FIM, which in my opinion is an important ongoing area of research. Additionally, the possibility of negative eigenvalues is underappreciated in practice, and so section 3.3 is a nice consideration of this situation.

Correctness:
I briefly reviewed the proofs in the appendix, and while they appear correct I could have missed a detail due to the sometimes complicated algebra.

Writing:
The paper is well written, with arguments and theorems clearly set up and presented. Minor nits:
-line 52, exponentially -> exponential
-line 189 “neural network” -> “the neural network”

Prior work:
To my knowledge no such bounds have been previously derived, and thus this is a sufficiently novel paper.

Additionally, I’m not sure if there is such a relation (the forms look similar enough I wanted to ask), but is there any relationship between your equation (8) and the Generalized Gauss-Newton Matrix (see sec. 8 of https://arxiv.org/abs/1412.1193)? If so that would be a nice connection to make.

Additional feedback:
This is by no means a requirement, as theory-only papers are perfectly acceptable, but while the examples in Table/Figure 1 are useful and compelling, the paper would be further strengthened by small-scale feed-forward neural net experiments where you could analytically compute the FIM and the approximations presented in this paper.

At the end of section 2.3, you say “control the scale of neural network output h_L”. Could this be possible by using temperature scaling or adding a normalization layer to the top of the network (although the latter may introduce complexities in your assumptions about the NN structure h_i)? It could be nice to at least mention that in practice it is easily possible/common to control the scale of h_L.

**Time Spent Reviewing:**

5

---

> ### Author Response · Authors · 2021-08-10
> **Reply to Reviewer muyd**
>
> Thank you for recognizing the value of this work, and highlighting our potential connections with various areas of ongoing research. Please see below for our response to the main concerns. Other comments will be addressed in the revision.
>
> ---
>
> **Eq.(8) and Generalized Gauss-Newton Matrix (GGN)**
>
> There is indeed a strong relationship here. The last term on the RHS of Eq. (8) is the GGN. The GGN can be obtained from the Hessian at a local optimum of the loss function. See (Martens, 2020) Eq. (5) and (6). In our case, the loss function is the log-likelihood of an exponential family, giving (as per Martin 2020's notation) $\bigtriangledown_{z} L(y, z) = \bigtriangledown_{z} (t^{\top}(y) z - F(z) ) = t(y) - \eta(z)$. We will include this discussion near Eq.(8) and the "Related work" section in the next iteration of the paper.
>
> ---
>
> **Small-scale feed-forward neural net experiments**
>
> We agree to have some experiments in the supplementary material to provide more intuitions. Please see our response "Empirical Results" to all reviewers.
>
> ---
>
> **Control the scale of neural network output** $\mathbf{h}_L$
>
> By section 3.2, the analysis of the variance of the estimators can be broken down into
> - solely reasoning about the first and second order derivatives of $ \mathbf{h}_{L} $; and
> - solely reasoning about the "position" of $ \mathbf{h}_{L} $ in the exponential family.
>
> The reviewer's suggested "temperature scaling or adding a normalization layer" are useful practical methods which can affect the variance of the FIM from the latter aspect. Alternative methods include to perform regularization on the scale of $\mathbf{h}_L$.  Surely, modifying the neural network structure could change the behavior of the gradients, but this can be controlled by imposing additional Lipschitz requirements. We will include the techniques mentioned by the reviewer as well as the additional discussion.
>
> ---
>
> References:
>
> Martens. New insights and perspectives on the natural gradient method. 2020 (update).

---

> > ### Comment · Reviewer_muyd · 2021-08-22
> > **Response**
> >
> > Thank you very much for the follow-up work and edits to the paper! I will be raising my score to an 8 to reflect that I already believed the paper was good enough for acceptance, but is now stronger.

---

### Official Review · Reviewer_Wuqv · 2021-07-16

**Rating:** 7
**Confidence:** 3

**Summary:**

Update: I thank the authors for their response, and I will keep the score as is. It would be nice to have a simple empirical analysis in the final supplemental material.


The paper discusses the covariance of two FIM estimators, which is useful in deep learning where the exact FIM over the full dataset can be hard to compute. One is based on the variance of the score (type 1) and the other is based on the Hessian of the relative entropy (type 2). Comparing these covariances can depend on the exponential family used in the model, type 1 has a higher variance for Normal and Poisson whereas type 2 has a higher variance for Bernoulli. Convergence rate results are given which is $O(N^{-0.5))$ with constant determined by the variances. These results provide some theoretical justification as to which estimator of FIM to use when using it to perform natural gradient in deep learning.

**Ethical Concerns:**

No.

**Limitations And Societal Impact:**

Limitations are discussed.
Societal impact is N/A because the paper is theoretical.

**Main Review:**

The paper is theoretical but is highly related to deep learning practice, especially when natural gradients are used. The paper compared two variants of FIM estimators, computed their covariances (Theorem 4, and Theorem 6), upper bounds (Corollaries 7 and 8), when these estimators are PSD (theorem 9, theorem 10), convergence rate (Lemma 11), and the structure of the deep learning network (Lemma 13).

I think the paper is clearly written, the results are solid, and the problem investigated is important to machine learning and natural gradient methods. The variance results clearly indicate what type of FIM estimator is preferred in NG in principle.

My only concern is that there are not any empirical results that corroborate with the theory. One very simple example is to take a very small neural network (just a two-layer one would suffice) on a small task, and evaluate their FIM with the same number of samples using the two FIM estimators. I also think it is interesting to see if these choices affect natural gradient optimization (although I do not think this is necessary to the paper).

Additional comment: can we gain insights regarding the spectrum of the estimated FIM, not just the smallest eigenvalue? This also seems to be related to stochastic optimization, because a small condition number for the true FIM does not necessarily indicate a small condition number of the estimated FIM?


**Time Spent Reviewing:**

3 hours

---

> ### Author Response · Authors · 2021-08-10
> **Reply to Reviewer Wuqv**
>
> Thank you for your detailed review and recognizing our analytical results and their value in the deep learning practice. Please see below for our response to the concerns. Other comments will be addressed in our next version.
>
> ---
>
> **Empirical results that corroborate with the theory**
>
> Please see our response to all reviewers regarding "Empirical Results". We would like to use the main text to summarize our theoretical discoveries (which is already dense), and therefore this exploration will be put in the supplementary material.
>
> ---
>
> **Insights regarding the spectrum of the estimated FIM**
> - Our bounds of the Frobenius norm of the variance matrix (based on Corollary 7 and 8; see "Further Analytical Results" above) give an upper bound of the scale or the "energy" of the spectrum variations, as the Frobenius norm coincides with the Euclidean norm of the spectrum. We will add a remark on this.
> - In the above empirical study (to be added to the supplementary), we plot the spectrum of the estimated FIM, and the spectrum of the variance of the FIM.  Both spectra have a low-rank structure, meaning most eigenvalues are close to zero.
> - This paper mainly tackles the variance of FIM estimators as random matrices in $\Re^{\dim(\mathbf{\theta})\times\dim(\mathbf{\theta})}$. A more careful analysis on the variance of the spectrum can consider geometric structure and divergence on the manifold of positive semi-definite matrices.  This is left to future study.

---

### Author Response · Authors · 2021-08-10
**General remarks**

We would like to thank the reviewers for their time in reviewing the submission and the valuable comments. We would like to note some recent progress and address some shared reviewer comments. Under each individual review, we also include a per-reviewer discussion to their points.

---

**Further Analytical Results**

We made one round of revision after the submission. We list our progress as below.
- Theorem 4 & Theorem 6 are re-stated to give the closed-form covariance between any two elements in the corresponding random matrix. They include our previous statements on the variance as special cases.
- Corollary 7 & Corollary 8 state the elementwise bound on the variance of the corresponding random matrix. They lead to two other corollaries, which give global bounds on the Frobenius norm of the corresponding random matrix.  These new corollaries can be used to bound the spectrum of the variance.
- A new section to state how to transfer our results after a change of coordinates (reparametrization of the neural network). We found $\hat{\mathcal{I}}_1(\mathbf{\theta})$ (but not $\hat{\mathcal{I}}_2(\mathbf{\theta})$) is a covariant tensor, just like the FIM.

---

**Empirical Results**

After receiving the review, we performed the following experiment, which will be added as a section in the supplementary material. Let us remark that the reviewers agreed such an experiment study (although can further strengthen the contribution by providing intuitions) is not strictly necessary.

We take the official example of PyTorch convolutional neural network (CNN) for MNIST classification.  We train the network in its default settings (test accuracy $\approx$ 99%) and obtain $\mathbf{\theta}$. We use a subset of the training set to evaluate the diagonal blocks (corresponding to the four layers) of $\hat{\mathcal{I}}_1(\mathbf{\theta})$ and $\hat{\mathcal{I}}_2(\mathbf{\theta})$.  This is computed efficiently based on methods in "torch.autograd.functional".  We show in colormaps the mean and variance of $\hat{\mathcal{I}}_1(\mathbf{\theta})$ and $\hat{\mathcal{I}}_2(\mathbf{\theta})$ for $M$ independent runs of these estimators based on different random seeds.  We also evaluate the spectrum of the variance matrices.
We observe
- these FIM estimators are indeed random (the variance does not vanish);
- their variance in the colormaps has a non-trivial structure;
- for both estimators, the variance matrix has a low-rank structure, meaning that most of its eigenvalues are close to zero. This agrees with the low-rank structure of the FIM (Karakida et al. 2019).

We further evaluated our variance upper bounds in Corollary 7 and Corollary 8.  Again, this is computed efficiently based on "torch.autograd.functional".  We show the histogram of the bound gap. We observe
- Our upper bounds can effectively bound the sample variance when $M$ enlarges (sample variance becomes close to the true variance).  This verifies Corollary 7 & 8 empirically.
- The histogram of the bound gap looks like an exponential distribution, with a large portion of the gap values close to zero.

---

References:

Karakida, Akaho, & Amari. Universal statistics of fisher information in deep neural networks: Mean field approach. 2019.

---

### Decision · Program_Chairs · 2021-09-27

**Decision:**

Accept (Poster)

**Comment:**

The reviewers agreed that this is a solid theoretical paper that makes a real contribution. The paper is well-written relative to the complexity of what it's presenting.